# NEW RECIPES FOR GRAPH ANOMALY DETECTION: FORWARD DIFFUSION DYNAMICS AND GRAPH GENERATION

## ABSTRACT

Distinguishing atypical nodes in a graph, which is known as graph anomaly detection, is more crucial than the generic node classification in real applications, such as fraud and spam detection. However, the lack of prior knowledge about anomalies and the extremely class-imbalanced data pose formidable challenges in learning the distributions of normal nodes and anomalies, which serves as the foundation of the state of the arts. We introduce a novel paradigm (first recipe) for detecting graph anomalies, stemming from our empirical and rigorous analysis of the significantly distinct evolving patterns between anomalies and normal nodes when scheduled noise is injected into the node attributes, referred to as the forward diffusion process. Rather than modeling the data distribution, we present three non-GNN methods to capture the evolving patterns and achieve promising results on nine widely-used datasets, while mitigating the oversmoothing limitation and shallow architecture of GNN methods. We further investigate the generative power of denoising diffusion models to synthesize training samples that align with the original graph semantics (second recipe). In particular, we derive two principles for designing the denoising neural network and generating graphs. With our proposed graph generation method, we attain record-breaking performance while our generated graphs are also capable of enhancing the results of existing methods. All the code and data are available at `https://github.com/DiffAD/DiffAD`.

## 1 INTRODUCTION

Learning the graph data distribution with regard to its structure and node attributes serves as the foundation of static graph analysis (Chami et al., 2022; Cui et al., 2019), especially for detecting anomalous graph entities (Akoglu et al., 2015; Ma et al., 2021). Given the learned data distributions, anomalies are significantly divergent from normal entities due to the deviating mechanisms that generate them. Though anomalies are far

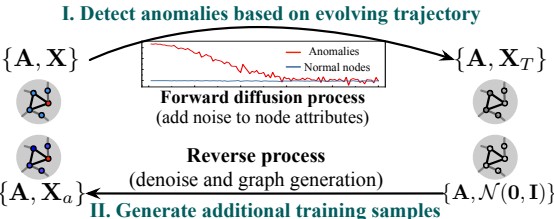

Figure 1: Two recipes for graph anomaly detection.

rarer and much less dominant than the majority, recognizing their presence and understanding their significant impacts are even more crucial for real-world applications. To list a few, fraud detection in online social networks (Dou et al., 2020; Wang et al., 2023b; Gao et al., 2023b), fake news detection in social media (Wang et al., 2023a), rare molecule detection for drug discovery, malware detection in computing systems, and brain health monitoring (Xu et al., 2022a; Ma et al., 2023).

More recently, graph neural networks (GNNs) have greatly advanced the frontier of graph machine learning, but learning the graph data distribution remains an open problem (Chen et al., 2022; Sun et al., 2023). Instead of solely learning either the structure distribution $p(\mathbf{A})$ or attribute distribution $p(\mathbf{X})$, graph learning techniques expect to learn the joint distribution $p(\mathbf{A}, \mathbf{X})$ considering the complex relation between the graph structure and node attributes. In the realm of graph anomaly detection, effectively capturing the distributions of anomalies is even more challenging due to the lack of prior knowledge about them and the tremendous cost of acquiring labeled anomalies. Nevertheless, since the number of anomalies is far less than normal entities, vanilla models will bias on learning normal entities giving such extremely imbalanced data (Johnson & Khoshgoftaar, 2019).

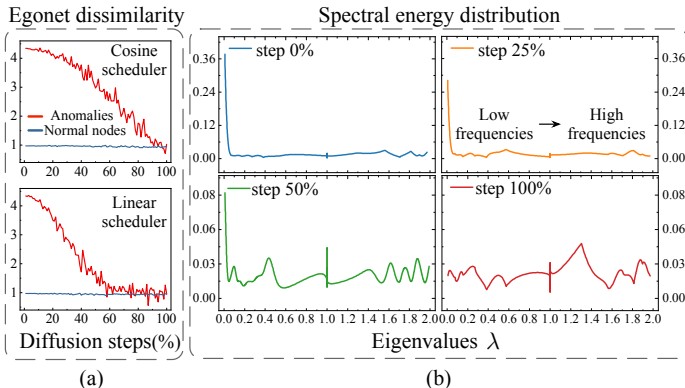

(a)             (b)

Figure 2: On Cora dataset, by gradually injecting scheduled noise to node attributes as the forward diffusion process. **(a).** Anomalies' egonet dissimilarities change more dramatically than normal nodes. **(b).** The ratios of lower frequency energy decreases while higher frequencies rise (shift from low to higher frequencies). Details are presented in §4.

Besides the challenges posed by the data for anomaly detection, GNNs also encounter intrinsic technical limitations (Azabou et al., 2023). Their fundamental message-passing (MP) mechanism undermines the effectiveness in learning the distributions of anomalies and normal nodes since vanilla MP aggregates anomalous and normal nodes' features with each other. Such schema blends anomalies and normal nodes in the representation space (Liu et al., 2020) and prevents a deeper architecture of GNNs since nodes will collapse together when staking more MP layers (Keriven, 2022).

In this paper, we first explore **a novel paradigm** to detect anomalies on a static graph without the need to explicitly learn its data distribution and to employ MP-GNNs. Our analysis on the differences between anomalies and their egonet neighbors, called *egonet dissimilarity*, illustrate that anomalies and normal nodes can be separated by gradually injecting $T$ scales of scheduled Gaussian noise into node attributes, referring to as the forward diffusion process. Scrutinizing the egonet dissimilarity changes in this process (Fig. 2(a)), we find that anomalies experience a more dramatic drop in egonet dissimilarity compared to normal nodes. We recognize this unexplored anomaly detection paradigm as classifying nodes in terms of their evolving trajectories in the forward diffusion.

Suffering from the shortage of knowledge about anomalies, we further investigate graph generation, assuming that high quality synthesized data could facilitate anomaly detectors learning a better decision boundary. Inspired by denoising-based generative models, we have delved into the reverse diffusion process for graph generation, which is to denoise the data, and introduce **two fundamental principles** to design the denoising neural network such that the generated graphs adhere to the original graph semantics. By inspecting the forward diffusion process via the lens of graph's spectrum, we discover a progressive shift in the graph's spectral energy from lower frequencies to higher frequencies as the diffusion forwards, which is depicted in Fig. 2(b) and Fig. 3 that the energy ratios of lower frequency signals (corresponding to smaller eigenvalues) decrease continuously from diffusion step 0 to 100% while higher frequencies are becoming more dominant. These observations from the forward diffusion process underscore the need for the denoising network to *1). Explore each node's egonet dissimilarity (capturing the local graph semantics)* and *2). Recover the low frequency graph energy*.

Upon these findings, we offer two fresh recipes for graph anomaly detection: 1) Distinguishing anomalies based on the distinctive forward diffusion dynamics and 2) Synthesizing additional samples to complement detectors (as depicted in Fig. 1). For learning the divergent dynamics of anomalies, we design three innovative non-GNN methods (§5) and for the purpose of graph generation, we follow the principles and devise a novel generative graph diffusion framework for synthesizing graphs, particularly for anomaly detection (§6). The main contributions of this paper are: **1) A novel paradigm to detect graph anomalies.** We, for the first time, propose to shift the focus of graph anomaly detection from learning a static data distribution to the exploration of dynamics in the forward diffusion process. This new paradigm enables us to promptly apply non-GNN techniques for investigating graph data, well-tackling the oversmoothing issues of MP-GNNs in anomaly detection. The promising results of our proposed non-GNN methods empirically underpin this as a potential research direction. **2) Two principles for designing the denoising-based graph diffusion model.** Adhering to the principles, our model can generate supplementary and effective training samples to mitigate the shortage of labeled anomalies. Extensive experiments on real-world datasets demonstrate that these generated samples can significantly improve the detection performance. **3)**, we rigorously review and prove our observations on the diffusion process, which will serve as foundations for future works in graph anomaly detection.

## 2 PRELIMINARIES

**Static attributed graph.** A static attributed graph $\mathcal{G} = \{\mathbf{A}, \mathbf{X}\}$ comprises $n$ nodes with attributes. $A_{i,j}$ in the adjacency matrix $\mathbf{A}$ is 1 if nodes $v_i$ and $v_j$ in $\mathcal{G}$ are directly connected; otherwise, 0. The attribute matrix $\mathbf{X} = [\boldsymbol{x}_i]_{n \times k}$ contains each node $v_i$'s $k$-dimensional attribute vector.

**Egonet dissimilarity.** The egonet dissimilarity $\boldsymbol{\Omega} = [\boldsymbol{\omega}_i]_{n \times k} = \mathbf{LX}$ quantifies how each node's attributes are different from its egonet neighbors. $\mathbf{L} = \mathbf{I} - \mathbf{D}^{-\frac{1}{2}} \mathbf{A} \mathbf{D}^{-\frac{1}{2}}$ is the normalized graph Laplacian corresponding to $\mathcal{G}$ and $\mathbf{D}$ is the diagonal degree matrix.

**Forward graph diffusion (process).** The forward graph diffusion is referred to as injecting $T$ scales of scheduled noise to node attributes with the fixed graph structure. For each diffusion step $\{t\}_0^T$, the corrupted graph $\mathcal{G}_{t+1} = \{\mathbf{A}, \mathbf{X}_{t+1}\}$ are derived from $\mathcal{G}_t = \{\mathbf{A}, \mathbf{X}_t\}$, with $\mathcal{G}_0 = \mathcal{G}$.

**Generative graph data augmentation for graph anomaly detection.** We define generative graph data augmentation as to synthesize additional graphs $\mathbb{G}_a = \{\mathcal{G}_a^1, \ldots, \mathcal{G}_a^{|\mathbb{G}_a|}\}$ for enhancing the graph anomaly detection performance. Each $\mathcal{G}_a^i = \{\mathbf{A}, \mathbf{X}_a^i\}$ has the same structure as the original graph $\mathcal{G}$ and the attribute distributions $p(\mathbf{X}_a^i | \mathbf{A}) \sim p(\mathbf{X} | \mathbf{A})$.

**Contextual anomalies and graph anomaly detection.** In this paper, we aim to detect contextual anomalies, which are defined as nodes exhibiting significantly different attributes compared to their neighbors (Liu et al., 2022b). The detection is conducted with access to a small proportion of labeled data so as to simulate real scenarios. Given an attributed graph $\mathcal{G}$ containing both anomalies in $\mathbb{V}_1$, and normal nodes in $\mathbb{V}_0$, we aim to learn a classification function that maps each node $v_i \in \mathbb{V}$, $\mathbb{V} = \mathbb{V}_1 \cup \mathbb{V}_0$ to its class label, which is 1 for anomalies and 0 for normal nodes, i.e., $f : \{\mathbf{A}, \mathbf{X}\} \to \boldsymbol{y} \in \{0, 1\}^n$. Practically, anomalies are far rarer than normal nodes, which means the cardinalities $|\mathbb{V}_1| \ll |\mathbb{V}_0|$.[1]

## 3 RELATED WORK

### 3.1 SEMI-/SUPERVISED GRAPH ANOMALY DETECTION

Anomalous node detection, particularly contextual anomaly detection, is a key topic in graph anomaly detection (Akoglu et al., 2015; Ma et al., 2021; Gavrilev & Burnaev, 2023). These anomalies are rare compared to the majority, but are pervasive in real-world scenarios. Concrete examples include fake news in social media, business fraudsters in financial systems, and malfunctioning cortices in brain networks. Tremendous effort has been committed to learn the graph distribution for detecting the violating anomalies, and most recent approaches have shifted to adopting MP-GNNs to investigate the abnormal patterns of anomalies (Dou et al., 2020; Tang et al., 2022; Liu et al., 2022b). However, due to the oversmoothing issue of MP-GNNs, straightforwardly applying them to anomaly detection is non-trivial, spurring Dou et al. (2020), Liu et al. (2020) and others to mitigate the negative impact of MP or seek band-pass filters to capture the signals of anomalies (Tang et al., 2022). These approaches consider the technical challenges associated with GNNs, but solutions to the shortage of labeled anomalies still lack sufficient exploration. Others, such as Ding et al. (2019), Zheng et al. (2021), and Liu et al. (2021) expect to address this with unsupervised/contrastive learning techniques, but they only predict the irregularity (a continuous score) of each node and cannot explicitly classify anomalies. Although a human defined threshold can be utilized to label anomalies, determining an effective threshold under the unsupervised setting is non-trivial in practice (Ma et al., 2021; Akoglu, 2021). Following (Tang et al., 2022), we do not cover structural anomalies that form more densely links with other nodes (Liu et al., 2022b) as this type of anomalies only exhibit deviating structural patterns and can be effectively detected using structure statistics like node degree and centralities.

### 3.2 DENOISING DIFFUSION PROBABILISTIC MODEL (DDPM)

Denoising diffusion probabilistic models (DDPMs) is a class of generative models built upon the *forward diffusion process* (diffusion process) and *reverse process* (denoising process) (Ho et al.,

---

[1]We use the terms 'graph anomalies' and 'anomalous nodes', as well as 'node features' and 'node attributes' interchangeably. We use anomalies to specifically represent contextual anomalies. The italic $T$ specifically denotes noise scales while the superscript '⊤' stands for the transpose of a matrix. To eliminate confusion, we use 'egonet dissimilarity' to specifically denote how each node's attributes differ from its egonet neighbors, distinct from the embedding method used in previous works.

2020; Nichol & Dhariwal, 2021). Typically, the diffusion process is defined as a Markov chain that progressively adds a sequence of scheduled Gaussian noise, to corrupt the original data $\boldsymbol{x}_0$ to standard Gaussian noise as follows:

$$q(\boldsymbol{x}_T|\boldsymbol{x}_0) = \prod_{t=1}^{T} q(\boldsymbol{x}_t|\boldsymbol{x}_{t-1}) = \mathcal{N}\big(\boldsymbol{x}_T; \sqrt{\bar{\alpha}_T}\boldsymbol{x}_0, (1-\bar{\alpha}_T)\mathbf{I}\big), \tag{1}$$

where $\bar{\alpha}_T = \prod_{t=1}^{T}(1-\beta_t)$ and $\beta_t$ is the variance of the noise at step $t$. The denoising process is a reverse Markov chain that attempts to recover the original data from noise by

$$p_{\boldsymbol{\theta}}(\boldsymbol{x}_{0:T}) := p(\boldsymbol{x}_T)\prod_{t=1}^{T} p_{\boldsymbol{\theta}}(\boldsymbol{x}_{t-1}|\boldsymbol{x}_t), \quad p_{\boldsymbol{\theta}}(\boldsymbol{x}_{t-1}|\boldsymbol{x}_t) := \mathcal{N}\big(\boldsymbol{x}_{t-1}; \boldsymbol{\mu}_{\boldsymbol{\theta}}(\boldsymbol{x}_t, t), \boldsymbol{\Sigma}_{\boldsymbol{\theta}}(\boldsymbol{x}_t, t)\big), \tag{2}$$

where the mean $\boldsymbol{\mu}_{\boldsymbol{\theta}}$ and variance $\boldsymbol{\Sigma}_{\boldsymbol{\theta}}$ of distribution $p_{\boldsymbol{\theta}}(\boldsymbol{x}_{t-1}|\boldsymbol{x}_t)$ are learned using a deep neural network with parameters $\boldsymbol{\theta}$. By minimizing the Kullback-Leibler (KL) divergence between $q(\boldsymbol{x}_{t-1}|\boldsymbol{x}_t, \boldsymbol{x}_0)$ and $p_{\boldsymbol{\theta}}(\boldsymbol{x}_{t-1}|\boldsymbol{x}_t)$, which is

$$\arg\min_{\boldsymbol{\theta}} D_{\text{KL}}(q(\boldsymbol{x}_{t-1}|\boldsymbol{x}_t, \boldsymbol{x}_0) \parallel p_{\boldsymbol{\theta}}(\boldsymbol{x}_{t-1}|\boldsymbol{x}_t)), \tag{3}$$

the neural network will capture the original data distributions. Consequently, new samples (i.e., $\boldsymbol{x}_0^a$) adhering to the original data distributions can be firmly generated by simply sampling $\boldsymbol{x}_T \sim \mathcal{N}(\mathbf{0}, \mathbf{I})$ and running the denoising process. Due to space limitations, we provide additional related works in graph anomaly detection and explanations of DDPM (including Eq. (3)) in Appendix A.

## 4 PRELIMINARY STUDY: ANOMALIES' DYNAMICS AND GRAPH ENERGY SHIFTS IN THE FORWARD DIFFUSION PROCESS

The power of diffusion models in discerning different modes of the data stems from the forward and reverse diffusion processes. Our work investigates both processes and unveils two significant observations, particularly in the context of graph anomaly detection.

### 4.1 PRELIMINARY STUDY SETUP

Our preliminary study aims at exploring: *The deviating dynamics of graph anomalies* and *Changes in the graph spectral energy distributions* during the forward diffusion. We conduct the study on Cora by fixing the graph structure and gradually injecting noise into node attributes. Specifically, we generate the noise employing two variance schedulers, i.e., linear scheduler and cosine scheduler, while the node attributes are randomly drawn from two Gaussian distributions, $\mathcal{N}(1, 1)$ for the normal class and $\mathcal{N}(1, 5)$ for anomalies, following Tang et al. (2022). Our observations are as follows.

### 4.2 OBSERVATION I - MORE DRAMATIC CHANGES IN ANOMALIES' EGONET SIMILARITIES

Since contextual anomalies have markedly distinct features compared to their egonet neighbors, we target the problem of what deviating patterns anomalies will manifest during the forward diffusion process. When measuring the average egonet dissimilarity $\frac{1}{|\mathbb{V}_y|}\sum_{v_j \in \mathbb{V}_y} \boldsymbol{\omega}_j$ for anomalies ($y=1$) and normal nodes ($y=0$) at each forward step, we surprisingly find that anomalies (the red line) undergo more substantial changes than normal nodes (the blue line), as depicted in Fig. 2(a).

**Proposition 1** *Given $T$ scales of noise from linear or cosine scheduler, when injecting them gradually into node attributes through the forward diffusion process, the egonet dissimilarities of anomalies change more dramatically than normal nodes.*

By this, we propose to detect anomalies by investigating the dynamics related to egonet dissimilarity $\boldsymbol{\Omega}_t$ in the diffusion process, as an alternation of learning the graph distribution. We recognize this as a novel paradigm for graph anomaly detection. Since learning $p(\mathbf{A}, \mathbf{X})$ is no longer mandatory, other powerful algorithms can be promptly adopted for detecting graph anomalies and breaking the limitations of MP-GNNs. As the pioneer in this line, we hereafter present three non-GNN models to capture such dynamics, namely FSC, TRC-TRANS, and TRC-MLP, built upon LSTM (Hochreiter & Schmidhuber, 1997), Transformer (Vaswani et al., 2017), and MLP (Goodfellow et al., 2016), respectively. Proof of this proposition is provided in Appendix B.

### 4.3 OBSERVATION II - RECOVER THE LOW FREQUENCY ENERGY FOR GRAPH GENERATION

Considering the recent success of DDPMs in generating high-quality data samples (Ho et al., 2020; Nichol & Dhariwal, 2021; Song et al., 2021), we are motivated to synthesize additional training samples to enrich the training set and complement anomaly detectors. However, the critical capabilities that the denoising neural network should possess for synthesizing training samples, especially in the context of graph anomaly detection, remain unexplored. We inspect particular changes in the graph spectrum across forward diffusion. Given a original graph signal $\boldsymbol{x}$ (a column of $\mathbf{X}$), a sequence of corrupted signal $(\boldsymbol{x}_1, \ldots, \boldsymbol{x}_T)^\top$ in the forward diffusion process (by Eq. (1)) and the eigenvectors $\mathbf{U} = (\boldsymbol{u}_1, \ldots, \boldsymbol{u}_n)^\top$ of the graph Laplacian $\mathbf{L}$, in which $\boldsymbol{u}_l$ is the eigenvector corresponding to the $l$-the smallest eigenvalue. We quantify the energy ratio of a particular frequency (rank $l$) at diffusion step $t$ as $\gamma_l(\boldsymbol{x}_t, \mathbf{L}) = (\hat{x}_l^t)^2 / \sum_{i=1}^n (\hat{x}_i^t)^2$,

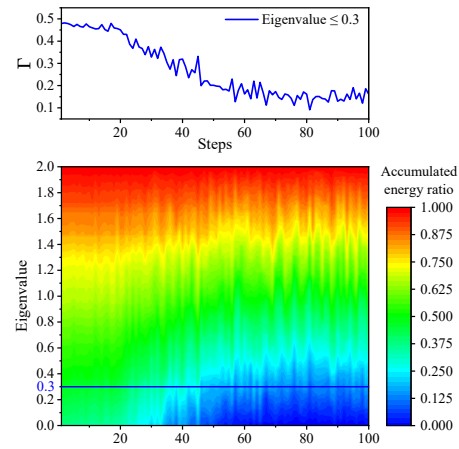

Figure 3: $\Gamma$ across diffusion steps.

where $\hat{\boldsymbol{x}}^t = \mathbf{U}^\top \boldsymbol{x}_t = (\hat{x}_1^t, \ldots, \hat{x}_l^t, \ldots, \hat{x}_n^t)^\top$ is the graph Fourier transformed signal. Taking $l$ as a threshold, we identify signals $(\hat{x}_1^t, \ldots, \hat{x}_l^t)$ as low frequency signals and the rest are the high frequency. As depicted in Fig. 2(b), the ratios of low frequency signals are gradually decreasing while higher frequencies are becoming more significant along the forward diffusion process. We further delve into the overall ratios of low and high frequency signals with regard to different thresholds at each step $t$ by measuring the accumulated energy ratio at rank $l$ following Tang et al. (2022) as:

$$\Gamma_l(\boldsymbol{x}_t, \mathbf{L}) = \sum_{i=1}^l \gamma_i(\boldsymbol{x}_t, \mathbf{L}). \tag{4}$$

We find that the accumulated energy is shifting to higher frequencies (as diffusion proceeds, the accumulated energy ratio with eigenvalues below a low threshold (e.g., 0.3) decreases continuously, as depicted Fig. 3) and we have the following expectation regarding $\Gamma_l(\boldsymbol{x}_t, \mathbf{L})$.

**Proposition 2** *The expectation of low frequency energy ratio $\mathbb{E}_{\boldsymbol{x} \sim \mathcal{N}(|\mu|, \sigma^2)}[\Gamma_l(\boldsymbol{x}_t, \mathbf{L})]$ is monotonically decreasing during the forward diffusion process.*

This indicates that the graph spectral energy distribution weights less on low frequency eigenvalues at step $t$ than $t-1$, and to denoise $\boldsymbol{x}_t$ for obtaining $\boldsymbol{x}_{t-1}$, the denoising network needs to recover the lower frequency energy. The proof is in Appendix C.

Ultimately, we identify two principles for designing denoising networks for DDPM-based graph generation. **Principle I** (from Observation I). *The denoising neural network should be capable of capturing the local information in egonets (the graph's local semantics)*, such that the prior distribution $p(\mathbf{X}_a | \mathbf{A})$ of the generated graph aligns with the original distribution $p(\mathbf{X} | \mathbf{A})$, enabling the classifier to learn a more effective decision boundary to distinguish anomalies. **Principle II** (from Observation II). *The denoising neural network needs to recover the low frequency energy.*

## 5 OUR APPROACH I – LEARNING DIFFUSION DYNAMICS FOR GRAPH ANOMALY DETECTION

Conforming to our novel paradigm for discerning anomalies upon the diffusion dynamics, in this section, we first introduce a new data structure for storing $\boldsymbol{\Omega}_t$ across forward diffusion, followed by our proposed sequence and trajectory learning based detection methods. Our algorithms are summarized in Appendix I.

### 5.1 STORING THE GRAPH INFORMATION IN FORWARD DIFFUSION

The forward diffusion process corrupts the original node attributes with respect to $T$ scales of scheduled noise. Eventually, the node attributes become standard Gaussian noise $\mathbf{X}_T \sim q(\mathbf{X}_T | \mathbf{A}) :=$

$\mathcal{N}(\mathbf{X}_T; \mathbf{0}, \mathbf{I})$ and for any discrete time step $t \in \{0, \ldots, T\}$, we can promptly infer the corrupted graph $\mathcal{G}_t = \{\mathbf{A}, \mathbf{X}_t\}$ based on Eq. (1) as follows:

$$\mathbf{X}_t \sim q(\mathbf{X}_t | \mathbf{X}, \mathbf{A}) = \mathcal{N}\big(\mathbf{X}_t; \sqrt{\bar{\alpha}_t}\mathbf{X}, (1 - \bar{\alpha}_t)\mathbf{I}\big). \tag{5}$$

We then employ a tensor $\mathbf{G} \in \mathbb{R}^{n \times T \times k}$ to store $\{\mathbf{\Omega}_t = \mathbf{L}\mathbf{X}_t\}_{t=0}^T$ at all diffusion steps. Specifically, the 2-D slice $\mathbf{G}_{i,:,:} = (\boldsymbol{\omega}_i^0, \ldots, \boldsymbol{\omega}_i^T)^\top$ encapsulates node $v_i$'s egonet dissimilarity from step 0 to step $T$. The 1-D slice $\mathbf{G}_{i,t,:} = \boldsymbol{\omega}_i^t \in \mathbb{R}^k$ denotes $v_i$'s egonet dissimilarity at a particular step $t$. Apparently, the memory cost of $\mathbf{G}$ is proportional to $T$, thus we present a skip-step algorithm to reduce its size, and provide a batch implementation to facilitate training (in Appendix D). Given $\mathbf{G}$, we then reformulate graph anomaly detection as to classify nodes with regard to their corresponding 2-D slices and propose the following methods.

## 5.2 FORWARD SEQUENCE CLASSIFICATION (FSC)

Each 2-D slice $\mathbf{G}_{i,:,:}$ is typically a sequence of multivariate features derived from $v_i$'s egonet dissimilarity in the diffusion process. Our goal is to encode the long- and short-term evolving patterns of the forward sequence into a hidden state for classifying nodes. Consider node $v_i$, we propose FSC to revisit $\boldsymbol{\omega}_i^t$ at each diffusion step and generates its hidden state $\boldsymbol{h}_0^i \in \mathbb{R}^d$ using a LSTM:

$$\boldsymbol{\tau}_t^i = tanh(\boldsymbol{W}_\tau \mathbf{G}_{i,t,:} + \boldsymbol{U}_\tau \boldsymbol{h}_{t+1}^i + \boldsymbol{b}_\tau), \qquad \boldsymbol{f}_t^i = tanh(\boldsymbol{W}_f \mathbf{G}_{i,t,:} + \boldsymbol{U}_f \boldsymbol{h}_{t+1}^i + \boldsymbol{b}_f) \tag{6}$$

$$\boldsymbol{g}_t^i = tanh(\boldsymbol{W}_g \mathbf{G}_{i,t,:} + \boldsymbol{U}_g \boldsymbol{h}_{t+1}^i + \boldsymbol{b}_g), \qquad \boldsymbol{o}_t^i = tanh(\boldsymbol{W}_o \mathbf{G}_{i,t,:} + \boldsymbol{U}_o \boldsymbol{h}_{t+1}^i + \boldsymbol{b}_o) \tag{7}$$

$$\boldsymbol{c}_t^i = \boldsymbol{f}_t^i \odot \boldsymbol{c}_{t+1}^i + \boldsymbol{\tau}_t^i \odot \boldsymbol{g}_t^i, \quad \text{and} \quad \boldsymbol{h}_t^i = \boldsymbol{o}_t^i \odot tanh(\boldsymbol{f}_t^i \odot \boldsymbol{c}_{t+1}^i + \boldsymbol{\tau}_t^i \odot \boldsymbol{g}_t^i), \tag{8}$$

where $\boldsymbol{h}_t^i, \boldsymbol{c}_t^i, \boldsymbol{\tau}_t^i, \boldsymbol{f}_t^i, \boldsymbol{g}_t^i$ and $\boldsymbol{o}_t^i$ are the hidden state, cell state, input gate, forget gate, cell gate, and output gate at step $t$, respectively. $\odot$ denotes the Hadamard product, and the initial $\boldsymbol{h}_T^i \sim \mathcal{N}(\mathbf{0}, \mathbf{I})$. We then predict its probabilities of being normal or anomalous through a fully connected layer:

$$f(\boldsymbol{h}_0^i; \boldsymbol{W}_c, \boldsymbol{b}_c) := \text{SOFTMAX}(\boldsymbol{h}_0^i \boldsymbol{W}_c + \boldsymbol{b}_c), \tag{9}$$

where $\boldsymbol{W}_c \in \mathbb{R}^{d \times 2}$ and $\boldsymbol{b}_c \in \mathbb{R}^2$ are trainable parameters. Regarding the extremely imbalanced data, we adjust the weights of anomalies and normal nodes in the training objective so as to regularize the model focuses equally on both classes. This class-wise training objective is to minimize:

$$\mathcal{L} = - \sum_{y \in \{0,1\}} \sum_{v_i \in \mathbb{V}_y} \frac{1}{|\mathbb{V}_y|} \log \psi(v_i \mid y), \tag{10}$$

where $\psi(v_i \mid y)$ is the predicted possibility of $v_i$ being anomalous ($y = 1$) or normal ($y = 0$).

## 5.3 TRAJECTORY REPRESENTATION-BASED CLASSIFICATION (TRC)

While FSC predicts a hidden state from the forward sequence for distinguishing anomalies, we propose TRC to learn a representation for each node's trajectory and detect anomalies upon it. As to capture the whole trajectory, for each node $v_i$, TRC first encodes $\mathbf{G}_{i,t,:}$ at each diffusion step $t$ into a latent space and then reads the trajectory representation out from all the steps following:

$$\boldsymbol{h}_i^{tr} = \text{READOUT}\left[\cup_{t \in [0,T]} g(\mathbf{G}_{i,t,:}; \boldsymbol{\theta}_g)\right], \quad \text{and} \quad \boldsymbol{h}_i^{tr} \in \mathbb{R}^d, \tag{11}$$

where $g(\cdot; \boldsymbol{\theta}_g)$ is the function for encoding $\mathbf{G}_{i,t,:}$, and READOUT$(\cdot)$ is to fuse the information at each diffusion step and extract trajectory representation $\boldsymbol{h}_i^{tr}$. We hereafter propose TRC-TRANS, and TRC-MLP to implement both functions. In TRC-TRANS, we adopt the raw Transformer architecture and generate the trajectory representation by passing $\mathbf{G}_{i,:,:}$ through the self-attention module (ATTN) and position-wise feed-forward neural network (FFN), which follows:

$$\text{ATTN}(\bar{\mathbf{G}}_{i,:,:}) = \text{SOFTMAX}(\frac{\mathbf{Q}_i \mathbf{K}_i^\top}{\sqrt{k}})\mathbf{V}_i, \quad \boldsymbol{h}_i^{tr} = \text{READOUT}\{\text{FFN}[\text{ATTN}(\bar{\mathbf{G}}_{i,:,:})]\}, \tag{12}$$

$$\text{with} \quad \mathbf{Q}_i = \bar{\mathbf{G}}_{i,:,:}\boldsymbol{W}_Q, \quad \mathbf{K}_i = \bar{\mathbf{G}}_{i,:,:}\boldsymbol{W}_K, \quad \mathbf{V}_i = \bar{\mathbf{G}}_{i,:,:}\boldsymbol{W}_V, \tag{13}$$

where $\boldsymbol{W}_Q, \boldsymbol{W}_K$, and $\boldsymbol{W}_V \in \mathbb{R}^{k \times k}$ are the projection matrices for $\mathbf{Q}, \mathbf{K}$, and $\mathbf{V}$, respectively. $\bar{\mathbf{G}}_{i,:,:}$ is $v_i$'s corresponding 2-D slice after adding the position encoding (Eq. (48)). The READOUT function is a fully connected layer. Ultimately, the trajectory representation-based classification is performed through the classifier formulated in Eq. (9) by replacing $\boldsymbol{h}_0^i$ with $\boldsymbol{h}_i^{tr}$ and the whole model can be flexibly trained via minimizing Eq. (10). For space limitation, we provide details of this readout function and present an even straightforward yet effective MLP-based model in Appendix E.

## 6 OUR APPROACH II - GENERATIVE GRAPH ANOMALY DETECTION

Motivated by the success of generative models in synthesizing high quality data samples, we propose a novel DDPM-based graph diffusion model (namely DIFFAD) to generate auxiliary training samples and complement detectors for more effective anomaly detection.

### 6.1 REVERSE PROCESS FOR DATA DISTRIBUTION MODELING

Let $\{\mathcal{G}_t\}_{t=0}^{T}$ denote a sequence of noised graphs in the forward graph diffusion process, each $\mathcal{G}_t = \{\mathbf{A}, \mathbf{X}_t\}$ and $\mathbf{X}_t \sim q(\mathbf{X}_t|\mathbf{X}, \mathbf{A}) = \mathcal{N}(\mathbf{X}_t; \sqrt{\bar{\alpha}_t}\mathbf{X}, (1 - \bar{\alpha}_t)\mathbf{I})$ (detailed in §5.1). Our goal is to learn the original graph data distribution through the reverse process, which can be practically described as to learn a denoising network in accordance with Eq. (3):

$$\boldsymbol{\theta}^* = \arg\min_{\boldsymbol{\theta}} D_{\text{KL}}[q(\mathbf{X}_{t-1}|\mathbf{X}_t, \mathbf{X}, \mathbf{A}) \parallel p_{\boldsymbol{\theta}}(\mathbf{X}_{t-1}|\mathbf{X}_t, \mathbf{A})]. \tag{14}$$

Given the fact that node attributes of the corrupted graph at each forward step $t$ follow distribution $\mathcal{N}(\mathbf{X}_t; \sqrt{\bar{\alpha}_t}\mathbf{X}, (1 - \bar{\alpha}_t)\mathbf{I})$, learning $\boldsymbol{\theta}^*$ via Eq. (14) is actually estimating the mean value $\sqrt{\bar{\alpha}_{t-1}}\mathbf{X}$ and variance $(1 - \bar{\alpha}_{t-1})\mathbf{I}$ of the prior step $t-1$ using graph $\mathcal{G}_t$ (detailed in Appendix G).

Upon our design Principle II (§4.3), which advises that the denoising network should recover low frequency signals, we opt to use graph convolutional neural network (GCN) (Kipf & Welling, 2017) as the backbone because of its capacity to act as a low-pass filter, attenuating high frequency signals and emphasizing lower frequencies (Nt & Maehara, 2019; Keriven, 2022). From the spatial perspective, GCN inherently explores the local graph semantics, which aligns with our principle I. Our proposed model DIFFAD has two ingredients: a step-dependent GCN (SDN) for learning node representations $\mathbf{Z}_t$ at step $t$ and DEN for estimating the distribution (mean and variance) of $\mathbf{X}_{t-1}$.

#### 6.1.1 STEP-DEPENDENT GCN - SDN

Built on Kipf & Welling (2016), we assume that the latent node representations also conform to a Gaussian distribution which is $p(\mathbf{Z}_t|\mathbf{X}_t, \mathbf{A}, t) \sim \mathcal{N}(\boldsymbol{\mu}_t^{\text{SDN}}, \text{diag}(\boldsymbol{\Sigma}_t^{\text{SDN}})) = \prod_{i=1}^{n} p(\boldsymbol{z}_t^i|\mathbf{X}_t, \mathbf{A}, t)$, with $p(\boldsymbol{z}_t^i|\mathbf{X}_t, \mathbf{A}, t) = \mathcal{N}(\boldsymbol{z}_t^i; \boldsymbol{\mu}_{i,t}, \boldsymbol{\sigma}_{i,t}^2)$. The matrices $\boldsymbol{\mu}_t^{\text{SDN}}$ and $\text{diag}(\boldsymbol{\Sigma}_t^{\text{SDN}})$ summarize the mean and variance vectors $(\boldsymbol{\mu}_{i,t}, \boldsymbol{\sigma}_{i,t}^2)$ of node representations $\boldsymbol{z}_t^i$ at step $t$, which are generated by:

$$\boldsymbol{\mu}_t^{\text{SDN}} = \text{SDN}_{\boldsymbol{\mu}}(\mathbf{X}_t, \mathbf{A}, t) = \underbrace{\tilde{\mathbf{A}}\overbrace{\text{ReLU}[\tilde{\mathbf{A}}(\mathbf{X}_t + \text{TE}(t))\boldsymbol{W}_1^{\text{SDN}}]}^{\text{first GCN layer}}\boldsymbol{W}_2^{\text{SDN}}}_{\text{second GCN layer}}, \tag{15}$$

$$\log[\text{diag}(\boldsymbol{\Sigma}_t^{\text{SDN}})] = \text{SDN}_{\boldsymbol{\sigma}}(\mathbf{X}_t, \mathbf{A}, t) = \tilde{\mathbf{A}}\,\text{ReLU}[\tilde{\mathbf{A}}(\mathbf{X}_t + \text{TE}(t))\boldsymbol{W}_1^{\text{SDN}}]\boldsymbol{W}_3^{\text{SDN}}, \tag{16}$$

where $\tilde{\mathbf{A}} = \mathbf{D}^{-\frac{1}{2}}\mathbf{A}\mathbf{D}^{-\frac{1}{2}}$ is the normalized adjacency matrix. $\boldsymbol{W}_1^{\text{SDN}}, \boldsymbol{W}_2^{\text{SDN}}$ and $\boldsymbol{W}_3^{\text{SDN}}$ are variables in the GCN layers. $\text{SDN}_{\boldsymbol{\mu}}$ and $\text{SDN}_{\boldsymbol{\sigma}}$ share the first layer, parametrized by $\boldsymbol{W}_1^{\text{SDN}}$. We incorporate the diffusion step $t$ in the learning process by encoding it as a matrix given by $\text{TE}(\cdot)$ (see Appendix F).

#### 6.1.2 DISTRIBUTION ESTIMATING GCN - DEN

Then, we propose DEN for predicting the less noisy node attributes $\mathbf{X}_{t-1}$ with $\mathbf{Z}_t$. Empirically, this is to estimate the mean and variance of $\mathbf{X}_{t-1}$ (in Eq. (5)) and we obtain them by

$$p(\mathbf{X}_{t-1}|\mathbf{Z}_t, \bar{\mathbf{Z}}_t, \mathbf{A}) \sim \mathcal{N}(\boldsymbol{\mu}_{t-1}^{\text{DEN}}, \text{diag}(\boldsymbol{\Sigma}_{t-1}^{\text{DEN}})) := \prod_{i=1}^{N} p(\boldsymbol{x}_{t-1}^i|\boldsymbol{z}_t^i, \bar{\boldsymbol{z}}_t^i, \mathbf{A}), \tag{17}$$

with $p(\boldsymbol{x}_{t-1}^i|\boldsymbol{z}_t^i, \bar{\boldsymbol{z}}_t^i, \mathbf{A}) = \mathcal{N}(\boldsymbol{x}_{t-1}^i; \boldsymbol{\mu}_{i,t-1}', \boldsymbol{\sigma}_{i,t-1}'^2)$, where $\bar{\boldsymbol{z}}_t^i \in \bar{\mathbf{Z}}_t$ is the output of the first GCN layer in SDN. We take this residual information from SDN as to prevent oversmoothing and further validate its effectiveness through the ablation study in §J.8. Notably, different from SDN, the matrices $\boldsymbol{\mu}_{t-1}^{\text{DEN}}$ and $\text{diag}(\boldsymbol{\Sigma}_{t-1}^{\text{DEN}})$ summarize the mean vectors $\boldsymbol{\mu}_{i,t-1}'$ and $\boldsymbol{\sigma}_{i,t-1}'^2$ of $\boldsymbol{x}_{t-1}^i$ to describe the distribution of $\mathbf{X}_{t-1}$, and are learned through a two-layered GCN similar to SDN as follows:

$$\boldsymbol{\mu}_{t-1}^{\text{DEN}} = \text{DEN}_{\boldsymbol{\mu}'}(\mathbf{Z}_t, \bar{\mathbf{Z}}_t, \mathbf{A}) = \underbrace{\tilde{\mathbf{A}}\{\overbrace{[\text{ReLU}(\tilde{\mathbf{A}}(\mathbf{Z}_t \oplus \mathbf{Z}_t)\boldsymbol{W}_1^{\text{DEN}})]}^{\text{first GCN layer}} \oplus \bar{\mathbf{Z}}_t\}\boldsymbol{W}_2^{\text{DEN}}}_{\text{second GCN layer}}, \tag{18}$$

$$\log[\text{diag}(\boldsymbol{\Sigma}_{t-1}^{\text{DEN}})] = \text{DEN}_{\boldsymbol{\sigma}'}(\mathbf{Z}_t, \bar{\mathbf{Z}}_t, \mathbf{A}) = \tilde{\mathbf{A}}\{[\text{ReLU}(\tilde{\mathbf{A}}(\mathbf{Z}_t \oplus \mathbf{Z}_t)\boldsymbol{W}_1^{\text{DEN}})] \oplus \bar{\mathbf{Z}}_t\}\boldsymbol{W}_3^{\text{DEN}}, \quad (19)$$

where $\oplus$ is for concatenation, $\boldsymbol{W}_1^{\text{DEN}}, \boldsymbol{W}_2^{\text{DEN}}$ and $\boldsymbol{W}_3^{\text{DEN}}$ parameterize the GCN layers.

### 6.1.3 SIMPLIFIED TRAINING OBJECTIVE OF THE REVERSE PROCESS

Eventually, all parameters $\boldsymbol{\theta} = \{\boldsymbol{W}_i^{\text{SDN}}\}_{i=1}^3 \cup \{\boldsymbol{W}_i^{\text{DEN}}\}_{i=1}^3 \cup \{\boldsymbol{W}_i^{\text{TE}}\}_{i=1}^2$ can be promptly fine-tuned with regard to Eq. (14) at each diffusion step. Likewise for the simplified training objective of DDPM, our training objective can be reformulated as to predict the added noise:

$$\boldsymbol{\theta}^* = \arg\min_{\boldsymbol{\theta}} \mathbb{E}_{\mathbf{X},\boldsymbol{\epsilon}}(||\boldsymbol{\epsilon} - \boldsymbol{\epsilon_\theta}||_2^2), \quad (20)$$

where $\boldsymbol{\epsilon_\theta}$ is the predicted noise. For space limitation, we provide details in Appendix G.

### 6.2 GRAPH GENERATION

Once the whole model is sufficiently trained, we can simply sample $\mathbf{X}_T$ from $\mathcal{N}(\mathbf{0}, \mathbf{I})$ and generate a new graph $\mathcal{G}_a$ by reversing the $T$ step forward diffusion (Eq. (2)) following:

$$\mathbf{X}_{t-1} = \frac{1}{\sqrt{\alpha_t}}\left(\mathbf{X}_t - \frac{1-\alpha_t}{\sqrt{1-\bar{\alpha}_t}}\boldsymbol{\epsilon_\theta}(\mathbf{X}_t, \mathbf{A}, t)\right) + \boldsymbol{\sigma}_t\boldsymbol{\epsilon}^*, \quad (21)$$

where $\boldsymbol{\epsilon}^* \sim \mathcal{N}(\mathbf{0}, \mathbf{I})$, and $\boldsymbol{\sigma}_t^2 = \frac{1-\bar{\alpha}_{t-1}}{1-\bar{\alpha}_t}\beta_t$. The generated sample can be then utilized as auxiliary data to enhance the anomaly detectors. The full algorithms are summarized in Appendix H and I.

### 6.3 GRAPH ANOMALY DETECTION WITH GENERATED SAMPLES

Given a set of generated graphs $\mathbb{G}_a = \{\mathcal{G}_a^1, \ldots, \mathcal{G}_a^{|\mathbb{G}_a|}\}$ and the original graph $\mathcal{G}$, we then train a two-layered GCN classifier (see Appendix F) by reformulating the class-wise objective in Eq. (10) to involve the training signals from the generated graphs, which can be formulated as:

$$\mathcal{L} = -\sum_{y\in\{0,1\}}\sum_{v_i\in\mathbb{V}_y}\frac{1}{|\mathbb{V}_y|}\left[\log\psi(v_i \mid y) + \frac{1}{|\mathbb{G}_a|}\sum_{g=1}^{|\mathbb{G}_a|}\log\psi(v_i^g \mid y)\right], \quad (22)$$

where $\psi(v_i^g \mid y)$ predicts the possibility of node $v_i$ being an anomaly or normal.

## 7 EXPERIMENTS

### 7.1 EXPERIMENTAL SETUP

The nine graph anomaly detection datasets can be categorized into two groups: one with organic anomalies, including YelpChi (Rayana & Akoglu, 2015), Reddit (Wang et al., 2021a), Weibo (Zhao et al., 2020), Tfinance, Tolokers and Questions (Tang et al., 2023); and another with injected anomalies (BlogCatalog (Ding et al., 2019), ACM (Ding et al., 2019), and Cora (Liu et al., 2022b)). Our methods are compared against three GNN detectors built upon GCN (Kipf & Welling, 2017), GAT (Veličković et al., 2018), and GraphSAGE (Hamilton et al., 2017), seven state-of-the-art semi-/supervised anomaly detectors: GeniePath (Liu et al., 2019), FdGars (Wang et al., 2019), BWGNN (Tang et al., 2022), DAGAD (Liu et al., 2022a), GAT-sep (Zhu et al., 2020), AMNet (Chai et al., 2022), GHRN (Gao et al., 2023a), and two contrastive detectors, namely CONAD (Xu et al., 2022b) and CoLA (Liu et al., 2021). We use a training ratio of 20% and report the 5-fold average performance (in percentage) along with the standard deviation using four commonly used metrics: **M**acro-**F1**, **M**acro-**Pre**cision, **M**acro-**Rec**all, and AUC (Ma et al., 2021). More details of the datasets and baselines can be found in Appendix J.

### 7.2 ANOMALY DETECTION PERFORMANCE

From the results in Tables 1 and 4 (Appendix J), we see that DIFFAD achieves the best results on almost all datasets. This confirms the validity of our second recipe that synthesized graphs could

Table 1: Detection results on six datasets (best in **bold**).

| Method | YelpChi | | Reddit | | Weibo | | BlogCatalog | | ACM | | Cora | |
|---|---|---|---|---|---|---|---|---|---|---|---|---|
| | M-F1 | AUC | M-F1 | AUC | M-F1 | AUC | M-F1 | AUC | M-F1 | AUC | M-F1 | AUC |
| GCN | 46.08±0.1 | 57.09±0.1 | 49.15±0.1 | 57.74±0.1 | 82.27±0.1 | 85.64±0.1 | 60.14±0.4 | 68.18±0.7 | 56.77±0.1 | 69.21±0.2 | 53.12±0.3 | 67.75±0.4 |
| GAT | 46.98±0.1 | 58.24±0.1 | 49.24±0.1 | 64.45±0.2 | 85.62±0.1 | 79.86±0.9 | 62.39±0.3 | 71.47±0.3 | 61.58±0.1 | 68.11±0.3 | 63.15±0.3 | 68.90±0.3 |
| GraphSAGE | 60.86±0.2 | 80.36±0.1 | 49.15±0.1 | 51.31±0.2 | 89.20±0.1 | 88.35±0.2 | 63.25±0.2 | 61.31±0.2 | 65.70±0.2 | 64.96±0.2 | 52.24±0.3 | 68.00±0.4 |
| GeniePath | 46.08±0.1 | 48.74±0.1 | 49.15±0.1 | 46.30±0.1 | 55.06±1.7 | 62.94±1.4 | 48.53±0.1 | 52.23±0.1 | 49.08±0.1 | 49.02±0.2 | 48.70±0.1 | 60.27±0.1 |
| FdGars | 49.77±0.1 | 53.01±0.5 | 48.52±0.2 | 63.05±0.1 | 87.65±0.1 | 93.11±0.6 | 42.23±0.3 | 54.59±0.2 | 36.69±0.2 | 65.03±0.1 | 42.10±0.1 | 69.69±0.4 |
| BWGNN | 63.68±0.3 | 80.96±0.1 | 43.29±0.5 | 66.89±0.3 | 89.27±0.3 | 92.29±0.3 | 52.97±0.1 | 51.35±0.3 | 54.93±0.3 | 47.69±0.6 | 52.18±0.2 | 44.93±0.2 |
| DAGAD | 52.08±0.2 | 59.83±0.1 | 49.15±0.1 | 61.49±0.3 | 89.63±0.1 | 91.05±0.1 | 64.49±0.4 | 73.88±0.5 | 72.03±0.2 | 73.74±0.1 | 65.42±0.2 | 68.78±0.3 |
| GAT-sep | 65.93±0.3 | 80.01±0.3 | 49.16±0.1 | 50.22±0.5 | 91.92±0.2 | 95.71±0.1 | 66.82±0.1 | 75.97±0.5 | 71.09±0.4 | 76.43±0.2 | 59.05±0.4 | 66.68±0.8 |
| AMNet | 54.66±0.8 | 64.01±1.2 | 50.39±0.1 | 62.14±0.3 | 91.63±0.1 | 97.11±0.1 | 71.77±0.2 | 72.23±0.3 | 60.11±0.1 | 74.54±0.2 | 53.09±0.1 | 66.09±0.6 |
| GHRN | 65.59±0.1 | 81.92±0.1 | 45.60±0.3 | 66.09±0.4 | 89.26±0.1 | 91.78±0.2 | 56.69±0.2 | 51.62±0.5 | 57.60±0.1 | 36.58±0.2 | 50.80±0.1 | 47.06±0.7 |
| CONAD | 47.42±0.1 | 47.50±0.2 | 46.39±0.1 | 55.78±0.2 | 79.01±0.1 | 90.40±0.1 | 53.87±0.2 | 63.03±0.1 | 53.16±0.1 | 70.86±0.1 | 53.53±0.1 | 70.48±0.7 |
| CoLA | 45.82±0.1 | 61.60±0.1 | 46.09±0.3 | 50.26±0.3 | 49.90±0.2 | 71.59±0.4 | 47.35±0.1 | 58.29±0.2 | 43.77±0.5 | 48.68±0.2 | 48.18±0.5 | 51.86±0.4 |
| FSC | 55.36±0.1 | 75.18±0.1 | 50.88±0.1 | 57.33±0.1 | 90.83±0.1 | 98.11±0.1 | 64.49±0.1 | 66.53±0.3 | 60.97±0.1 | 55.09±0.2 | 65.63±0.1 | 74.08±0.1 |
| TRC-MLP | 55.36±0.1 | 75.64±0.1 | 50.23±0.1 | 58.41±0.1 | 90.75±0.1 | 95.80±0.1 | 53.03±0.1 | 62.36±0.1 | 59.13±0.1 | 72.21±0.1 | 64.68±0.1 | 73.57±0.1 |
| TRC-TRANS | 56.58±0.1 | 72.83±0.2 | 48.21±0.2 | 57.11±0.1 | **92.06±0.1** | **98.17±0.1** | 53.12±0.1 | 54.74±0.1 | 51.64±0.1 | 52.44±0.1 | 65.71±0.1 | **76.32±0.2** |
| DIFFAD | **73.88±0.1** | **87.94±0.1** | **51.85±0.1** | **71.20±0.1** | 90.58±0.1 | 95.46±0.5 | **76.24±0.2** | **77.55±0.5** | **73.91±0.2** | **77.40±0.2** | **69.28±0.2** | 74.05±0.2 |

complement anomaly detectors to better distinguish anomalies, thereby mitigating the shortage of labeled data. While DAGAD also aims to enhance performance through data augmentation, it primarily focuses on combining class-biased features and cannot generate auxiliary training samples. GeniePath, FdGars and BWGNN only investigate anomalies' patterns by proposing new graph signal filtering algorithms or by constructing discriminating features from the raw data. They ignore the challenges imposed by the scarcity of anomalies, thus obtain compromised results. The performance of the three GNN detectors reveals the power of these vanilla GNN backbones but they still suffer from the oversmoothing problem of MP-GNNs (as described in §1). Our non-GNN methods (i.e., FSC, TRC-MLP, and TRC-TRANS), which are built in accordance with the new graph anomaly detection paradigm (first recipe), obtain the top-3 performance on Weibo dataset. We attribute this to the significantly lower feature similarities between anomalies and normal nodes (0.004 vs. 0.993) (Liu et al., 2022b), which makes anomalies' trajectories easier to be distinguished from normal nodes. The competitive results on other datasets demonstrate that this paradigm worth future exploration.

### 7.3 CASE STUDY I - THE EFFICACY OF GENERATED GRAPHS

We further investigate the effectiveness of our generated graphs in enhancing other state-of-the-art detectors by feeding them as additional training samples. For fairness, we generate one graph using DIFFAD and reformulate existing detectors' objectives (similar to Eq. (22)) to enjoy the training signals from the

Table 2: Performance improvement brought by generated graphs.

| Method | YelpChi | | Reddit | |
|---|---|---|---|---|
| | M-F1 | AUC | M-F1 | AUC |
| GAT | 72.68±0.1 (↑ **55%**) | 86.70±0.1 (↑ **49%**) | 51.42±0.1 (↑ **4.4%**) | 66.33±0.2 (↑ **2.9%**) |
| GraphSAGE | 74.21±0.1 (↑ **22%**) | 87.51±0.1 (↑ **8.9%**) | 52.29±0.1 (↑ **6.4%**) | 67.90±0.2 (↑ **32%**) |
| GeniePath | 51.67±0.1 (↑ **12%**) | 59.43±0.1 (↑ **22%**) | 50.15±0.2 (↑ **2.1%**) | 47.05±0.1 (↑ **1.6%**) |
| FdGars | 55.74±0.1 (↑ **15%**) | 68.27±0.1 (↑ **29%**) | 48.55±0.1 (↑ **0.1%**) | 64.03±0.1 (↑ **1.6%**) |
| BWGNN | 64.88±0.2 (↑ **1.9%**) | 81.06±0.1 (↑ **0.1%**) | 47.69±0.1 (↑ **10%**) | 70.31±0.1 (↑ **5.1%**) |
| DAGAD | 53.71±0.3 (↑ **3.1%**) | 60.11±0.1 (↑ **0.5%**) | 52.81±0.3 (↑ **7.4%**) | 69.28±0.1 (↑ **13%**) |
| GAT-sep | 66.91±0.2 (↑ **1.5%**) | 83.79±0.1 (↑ **4.7%**) | 50.15±1.8 (↑ **2.0%**) | 61.55±0.5 (↑ **23%**) |
| AMNet | 61.45±0.3 (↑ **12%**) | 81.69±0.1 (↑ **27%**) | 51.35±0.1 (↑ **1.9%**) | 70.04±0.1 (↑ **12%**) |
| GHRN | 69.36±0.2 (↑ **5.7%**) | 86.32±0.1 (↑ **5.4%**) | 47.53±0.3 (↑ **4.2%**) | 72.59±0.2 (↑ **9.8%**) |
| CONAD | 52.68±0.1 (↑ **11%**) | 50.32±0.1 (↑ **5.9%**) | 49.15±0.1 (↑ **5.9%**) | 57.23±0.1 (↑ **2.6%**) |
| CoLA | 46.14±0.2 (↑ **0.7%**) | 65.42±0.1 (↑ **6.2%**) | 47.04±0.1 (↑ **2.1%**) | 51.53±0.1 (↑ **2.5%**) |

generated graph. We select two real-world datasets, namely YelpChi and Reddit, and report the improved performance and growth rate (in bracket) on M-F1 and AUC in Table 2. As can be seen, the additional graph samples can improve the performance of the existing detectors to different degrees. This empirically proves that the additional samples synthesized by DIFFAD could provide complementary information about anomalies, leading to boosted performance. We also notice that such improvement is dependent on the detectors. We attribute this to the varying capabilities of each method in learning the data distribution and assimilating synthetic information. We report additional experiments on exploring the impact of the generated graphs, key parameters and the skip-step algorithm, as well as an ablation study in Appendix J.

## 8 CONCLUSION

We offer two fresh recipes for graph anomaly detection based on our scrutiny of the forward diffusion process. We discover that anomalies can be distinguished with regard to their distinct dynamics in the diffusion process (first recipe), and the denoising network for generating auxiliary training data (second recipe) needs to be capable of recovering low frequency signals. Upon these findings, we design three non-GNN methods and a generative graph diffusion model to detect anomalies. Our methods deliver record-breaking performance across nine widely-used datasets, with merely 20% of labeled data, and our generated graphs also significantly boost other detectors' performance.

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

# A   RELATED WORK

## A.1   GRAPH ANOMALY DETECTION

Anomalous node detection, particularly contextual anomaly detection, is a key topic in graph anomaly detection (Akoglu et al., 2015; Ma et al., 2021; Gavrilev & Burnaev, 2023). These anomalies are rare compared to the majority, but they are pervasive in real scenarios. Concrete examples include fraudulent/malicious users in online social networks, business frauds in financial systems, and abnormal cortices in brain networks. In semi-/supervised scenarios, graph anomaly detection can be treated as a specific yet significantly different binary node classification task, compared to the generic classification (Tang et al., 2022). The main differences are two folds. Essentially, anomalies are rare and generated by other unknown mechanisms. Directly learning anomalies' distributions usually leads to biased results for the shortage of labeled anomalies and extremely class-imbalanced data (Liu et al., 2022a; Wang et al., 2021b). Moreover, to maximize their influence under a controlled cost, most anomalies tend to form links with normal nodes rather than other anomalies, violating the homophility assumption (Zhao et al., 2021; Dou et al., 2020).

As to explore the graph structure and node attributes to fuse abnormal patterns of anomalies, recent approaches in semi-/supervised graph anomaly detection have shifted to use GNNs (Dou et al., 2020; Tang et al., 2022; Liu et al., 2022b). They attempt to directly assign labels to anomalies by utilizing knowledge from few labeled anomalies in the training data. For instance, BWGNN (Tang et al., 2022) employs the Beta graph wavelet and utilizes few labeled anomalies to learn band-pass filters to capture the signals of anomalies, DAGAD (Liu et al., 2022a) permutes class-biased and unbiased features from few labeled anomalies as to enhance the node representations and performance. For overcoming the scarcity of labeled anomalies, DAGAD augments a set of labeled node representations through random combinations of class-biased and unbiased features. However, these synthesized samples may not follow the original data distribution and alleviate the performance due to randomness.

These existing semi-/supervised approaches achieved considerable results, but they all focus on the vanilla routine of investigating the static patterns of anomalies and distributions (Ma et al., 2021). Besides, they almost rely on GNNs, exhibiting the intrinsic limitations from the message-passing schema, oversmoothing phenomenon, and shadow architectures. Other powerful deep learning frameworks as well as valuable clues including our discoveries in the egonet dissimilarity dynamics have yet to be sufficiently explored in this field. Furthermore, generating additional graph samples that adhere to the original data distribution to tackle the shortage of labeled anomalies is still an unexplored territory.

Notably, there are plenty of unsupervised anomaly detection techniques, such as CONAD (Liu et al., 2021), SL-GAD (Zheng et al., 2021) and Sub-CR (Zhang et al., 2022), that investigate the consistency between anomalies and their neighbors in different contrastive views to measure node irregularity. While these methods investigate the egonet dissimilarity for anomaly detection, they only focus on the original graph, and the dynamics of egonet dissimilarity in the forward diffusion process remains unexplored. Moreover, these unsupervised methods only predict the irregularity (a continuous score) of each node and cannot explicitly classify anomalies. Although a human defined threshold can be applied for labeling anomalies, it is non-trivial to get an effective and practical threshold under the unsupervised setting (Akoglu, 2021; Ma et al., 2021).

## A.2   DDPM

Denoising diffusion probabilistic models (DDPMs) have recently shown their great power in image synthesis (Ho et al., 2020; Nichol & Dhariwal, 2021), time series forecasting (Li et al., 2022), and many other generative tasks (Yang et al., 2022). These remarkable advances show that one can generate high quality samples by capturing the modes of a data distribution through the denoising diffusion process. Practically, DDPM contains two processes, namely the *forward diffusion process* (also called *diffusion process*), which gradually adds scheduled Gaussian noise to the original data $\boldsymbol{x}_0$ through a $T$-step Markov chain such that the eventual distribution at the last step $q(\boldsymbol{x}_T|\boldsymbol{x}_0) \sim \mathcal{N}(\boldsymbol{0}, \mathbf{I})$. And the *reverse process* (or *denoising process*), which strives to recover the data by removing the noise at each time step.

In the forward diffusion process, given the noise variance schedule $\beta_t \in (0, 1)$, the noisy data at step $t$ can be written in a closed form:

$$q(\boldsymbol{x}_t|\boldsymbol{x}_0) = \prod_{s=1}^{t} q(\boldsymbol{x}_s|\boldsymbol{x}_{s-1}) = \mathcal{N}(\boldsymbol{x}_t; \sqrt{\bar{\alpha}_t}\boldsymbol{x}_0, (1 - \bar{\alpha}_t)\mathbf{I}), \tag{23}$$

where $\bar{\alpha}_t = \prod_{s=1}^{t} \alpha_s$ and $\alpha_t = 1 - \beta_t$. While the reverse process attempts to recover the original data from noise following

$$p_{\boldsymbol{\theta}}(\boldsymbol{x}_{0:T}) := p(\boldsymbol{x}_T) \prod_{t=1}^{T} p_{\boldsymbol{\theta}}(\boldsymbol{x}_{t-1}|\boldsymbol{x}_t), \quad p_{\boldsymbol{\theta}}(\boldsymbol{x}_{t-1}|\boldsymbol{x}_t) := \mathcal{N}(\boldsymbol{x}_{t-1}; \boldsymbol{\mu}_{\boldsymbol{\theta}}(\boldsymbol{x}_t, t), \boldsymbol{\Sigma}_{\boldsymbol{\theta}}(\boldsymbol{x}_t, t)), \tag{24}$$

where the mean and variance (i.e., $\boldsymbol{\mu}_{\boldsymbol{\theta}}$ and $\boldsymbol{\Sigma}_{\boldsymbol{\theta}}$) of the data distribution $p_{\boldsymbol{\theta}}(\boldsymbol{x}_{t-1}|\boldsymbol{x}_t)$ are learned using a deep neural network with parameters $\boldsymbol{\theta}$.

According to the results in Nichol & Dhariwal (2021) and Luo (2022), the combination of $p$ and $q$ is typically a VAE (Kingma & Welling, 2014) and its evidence lower bound is given by $L = L_0 + \cdots + L_{t-1} + \cdots + L_T$, in which:

$$L_0 = -\log p_{\boldsymbol{\theta}}(\boldsymbol{x}_0|\boldsymbol{x}_1) \tag{25}$$

$$L_{t-1} = D_{\text{KL}}\big(q(\boldsymbol{x}_{t-1}|\boldsymbol{x}_t, \boldsymbol{x}_0) \parallel p_{\boldsymbol{\theta}}(\boldsymbol{x}_{t-1}|\boldsymbol{x}_t)\big) \tag{26}$$

$$L_T = D_{\text{KL}}\big(q(\boldsymbol{x}_T|\boldsymbol{x}_0) \parallel p(\boldsymbol{x}_T)\big), \tag{27}$$

and since the $L_T$ term is independent to $\boldsymbol{\theta}$, it can be dropped for training purposes while $L_0$ term can be approximated using Monte Carlo (Luo, 2022) or following Ho et al. (2020). Therefore, the rest terms related to the sum of $L_{t-1}$ are expected to be minimized and for brevity, this is formulated as:

$$\arg\min_{\boldsymbol{\theta}} D_{\text{KL}}\big(q(\boldsymbol{x}_{t-1}|\boldsymbol{x}_t, \boldsymbol{x}_0) \parallel p_{\boldsymbol{\theta}}(\boldsymbol{x}_{t-1}|\boldsymbol{x}_t)\big). \tag{28}$$

By minimizing this, the neural network will capture the original data distributions as well as new samples (i.e., $\boldsymbol{x}_0^a$) adhering to the original data distributions can be firmly generated through the reverse process by simply sampling $\boldsymbol{x}_T \sim \mathcal{N}(\mathbf{0}, \mathbf{I})$. Practically, the training objective can be replaced by an equivalent variant (Ho et al., 2020; Luo, 2022; Nichol & Dhariwal, 2021) to minimize the distance between the injected noise $\epsilon$ and prediction at arbitrary steps following:

$$\arg\min_{\boldsymbol{\theta}} \mathbb{E}_{\boldsymbol{x}_0, \epsilon} \left( ||\epsilon - \epsilon_{\boldsymbol{\theta}}(\sqrt{\bar{\alpha}_t}\boldsymbol{x}_0 + \sqrt{1 - \bar{\alpha}_t}\epsilon, t)||_2^2 \right), \tag{29}$$

For space limitation, we recommend referring to relevant literature such as (Ho et al., 2020; Nichol & Dhariwal, 2021; Luo, 2022; Yang et al., 2022) for more details about DDPMs.

To date, although DDPMs have shown superior capability in conventional data generation tasks, the progress on denoising graph diffusion still focuses on weighted graphs and generic graph-level tasks (Jo et al., 2023; Huang et al., 2022; Jing et al., 2022). They all focus on modeling the graph-level data distribution patterns for a set of graphs, while capturing the node attribute and local structure distributions within a single graph, especially for graph anomaly detection has not been explored yet. In this work, we investigate the DDPM framework specifically for graph anomaly detection and following our definition in §2, our forward graph diffusion process is to inject scheduled noise to the node attributes, $\mathbf{X}$, and the graph structure is fixed at all diffusion steps.

# B  PROOF OF PROPOSITION 1

With the definition of *contextual anomalies* in §2, we prove our Proposition 1. Based on Eq. (1) and the reparameterization trick, we can get the corrupted graph $\mathcal{G}_t = \{\mathbf{A}, \mathbf{X}_t\}$ at an arbitrary diffusion

step $t$ by

$$\mathbf{X}_t = \sqrt{\alpha_t}\mathbf{X}_{t-1} + \sqrt{1-\alpha_t}\boldsymbol{\epsilon}^*_{t-1} \tag{30}$$

$$= \sqrt{\alpha_t\alpha_{t-1}}\mathbf{X}_{t-2} + \sqrt{\alpha_t - \alpha_t\alpha_{t-1}}\boldsymbol{\epsilon}^*_{t-2} + \sqrt{1-\alpha_t}\boldsymbol{\epsilon}^*_{t-1} \tag{31}$$

$$= \sqrt{\alpha_t\alpha_{t-1}}\mathbf{X}_{t-2} + \sqrt{\sqrt{\alpha_t - \alpha_t\alpha_{t-1}}^2 + \sqrt{1-\alpha_t}^2}\boldsymbol{\epsilon}_{t-2} \tag{32}$$

$$= \sqrt{\alpha_t\alpha_{t-1}}\mathbf{X}_{t-2} + \sqrt{1-\alpha_t\alpha_{t-1}}\boldsymbol{\epsilon}_{t-2} \tag{33}$$

$$\vdots \tag{34}$$

$$= \sqrt{\prod_{s=1}^{t}\alpha_s}\mathbf{X} + \sqrt{1-\prod_{s=1}^{t}\alpha_s}\boldsymbol{\epsilon} \tag{35}$$

$$= \sqrt{\bar{\alpha}_t}\mathbf{X} + \sqrt{1-\bar{\alpha}_t}\boldsymbol{\epsilon} \tag{36}$$

$$\sim \mathcal{N}\left(\mathbf{X}_t; \sqrt{\bar{\alpha}_t}\mathbf{X}, (1-\bar{\alpha}_t)\mathbf{I}\right), \tag{37}$$

in which $\boldsymbol{\epsilon}^*_{t-1}, \boldsymbol{\epsilon}^*_{t-2}, \boldsymbol{\epsilon} \sim \mathcal{N}(\mathbf{0}, \mathbf{I})$. In Lines 31 and 32, we rewrite $\sqrt{\alpha_t - \alpha_t\alpha_{t-1}}\boldsymbol{\epsilon}^*_{t-2} + \sqrt{1-\alpha_t}\boldsymbol{\epsilon}^*_{t-1}$ as an expression of another Gaussian random variable $\boldsymbol{\epsilon}_{t-2}$, since $\boldsymbol{\epsilon}^*_{t-2}$ and $\boldsymbol{\epsilon}^*_{t-1}$ are two independent Gaussian random variables and their sum is still a Gaussian, whose mean and variance can be obtained by summing the mean and variance of $\boldsymbol{\epsilon}^*_{t-2}$ and $\boldsymbol{\epsilon}^*_{t-1}$, respectively.

Then, for any node $v_i$, its corrupted attribute $\boldsymbol{x}^t_i$ in $\mathbf{X}_t$ can be inherently formulated as:

$$\boldsymbol{x}^t_i = \sqrt{\bar{\alpha}_t}\boldsymbol{x}_i + \sqrt{1-\bar{\alpha}_t}\boldsymbol{\epsilon}, \tag{38}$$

where $\boldsymbol{x}_i$ is the original node attribute vector, and $v_i$'s egonet dissimilarity $\boldsymbol{\omega}_i$ at this arbitrary step $t$ is quantified as:

$$\boldsymbol{\omega}^t_i = \frac{1}{|\operatorname{ego}(v_i)|}\sum_{j\in\operatorname{ego}(v_i)}\left(\boldsymbol{x}^t_i - \boldsymbol{x}^t_j\right) \tag{39}$$

$$= \frac{1}{|\operatorname{ego}(v_i)|}\sum_{j\in\operatorname{ego}(v_i)}\left(\sqrt{\bar{\alpha}_t}\boldsymbol{x}_i + \sqrt{1-\bar{\alpha}_t}\boldsymbol{\epsilon}_i - \sqrt{\bar{\alpha}_t}\boldsymbol{x}_j - \sqrt{1-\bar{\alpha}_t}\boldsymbol{\epsilon}_j\right) \tag{40}$$

$$\approx \frac{\sqrt{\bar{\alpha}_t}}{|\operatorname{ego}(v_i)|}\sum_{j\in\operatorname{ego}(v_i)}\left(\boldsymbol{x}_i - \boldsymbol{x}_j\right) \tag{41}$$

where $\boldsymbol{\epsilon}_i, \boldsymbol{\epsilon}_j$ are the corresponding Gaussian noise added to $v_i$ and $v_j$. $\operatorname{ego}(v_i)$ denotes all $v_i$'s one-hop neighbors, and $|\operatorname{ego}(v_i)|$ is the number of one-hop neighbors.

We can then quantify how fast $\boldsymbol{\omega}^t_i$ changes during the diffusion process as its partial derivative with respect to $t$, which is:

$$\frac{\partial\boldsymbol{\omega}^t_i}{\partial t} \approx \underbrace{\left[\frac{1}{|\operatorname{ego}(v_i)|}\sum_{j\in\operatorname{ego}(v_i)}\left(\boldsymbol{x}_i - \boldsymbol{x}_j\right)\right]}_{\text{①}}\frac{\partial\sqrt{\bar{\alpha}_t}}{\partial t}. \tag{42}$$

Regarding the fact that contextual anomalies' attributes are significantly different from their neighbors, while normal nodes are similar to their egonet neighbors, the absolute values of ① with respect to anomalies will be larger than that of normal nodes. Given $\frac{\partial\sqrt{\bar{\alpha}_t}}{\partial t}$ the same for each node, $\boldsymbol{\omega}^t_i$ of anomalies will change more dramatically than normal nodes.

## C   PROOF OF PROPOSITION 2

We follow the approach of previous works (Grubbs, 1969; Tang et al., 2022) by assuming all node attributes are identically independently drawn from a Gaussian distribution $\mathcal{N}(x; |\mu|, \sigma^2)$, and the coefficient of variance $\sigma/|\mu|$ measures the degree of anomalies in the graph signal $x$. Consequently, the coefficient will be higher if there are more anomalies in the graph. As the forward diffusion

destroys the original graph signal $\boldsymbol{x}$ and $\boldsymbol{x}_t \sim \mathcal{N}(\sqrt{\bar{\alpha}_t}\boldsymbol{x}, 1 - \bar{\alpha}_t)$, we can measure the signal-to-noise ratio at an arbitrary step $t$ by

$$\text{SNR}(t) = \frac{\mu_t^2}{\sigma_t^2} = \frac{\bar{\alpha}_t}{1 - \bar{\alpha}_t}, \tag{43}$$

which denotes the ratio between the original signal and noise being added. Therefore, a higher SNR means more original signal, while a lower SNR indicates more noise.

Recall that the forward diffusion process aims to corrupt the original graph signal by adding noise. As the noise variance scheduler $\beta_t$ increases over the diffusion step, correspondingly, $\bar{\alpha}_t$ and the signal-to-noise ratio SNR decrease progressively. As a result, the measurement of the degree of anomalies, which can be quantified with regard to SNR as:

$$\frac{\sigma}{|\mu|} = \frac{\sqrt{1 - \bar{\alpha}_t}}{\sqrt{\bar{\alpha}_t}} = \sqrt{\frac{1}{\text{SNR}(t)}}, \tag{44}$$

increases as the diffusion proceeds. Then, based on the Proof 1 in Tang et al. (2022), the expectation of low frequency energy ratio $\mathbb{E}(\Gamma_l(\mathbf{X}_t, \mathbf{L}))$ will decrease as $\frac{\sigma}{|\mu|}$ increases during the forward diffusion process.

## D    SKIP-STEP FOR MEMORY OPTIMIZATION AND BATCHING

While we attempt to represent the original graph as $\mathbf{G}$, its memory cost and model fitting time are obviously proportional to the number of diffusion steps. To save the memory and training time costs, we can either reduce the number of scales $T$ in the forward diffusion process or skip potential steps. We refer to the second as skip-step, achieved by a sliding window-based method. Specifically, we generate a downsampled tensor following $\hat{\mathbf{G}}_{:,t,:} = \mathbf{G}_{:,t+s:t+s+w,:}$ for anomaly detection, where $0 < t + s + w \leq T$, $w$ is the windows size and $s$ is the stride size. We further evaluate our models' sensitivity to $w$ and $s$ in §J.6.

$\mathbf{G}$ can also be flexibly divided into smaller batches for training. Unlike graph batching which might cause information loss due to neighborhood sampling, since $\mathbf{G}$ stores a 2-D tensor (matrix) for each node, we can directly apply the existing batching methods in CV and NLP for training our graph anomaly detectors.

## E    IMPLEMENTATION DETAILS OF READOUT AND TRC-MLP

We present details about the READOUT function hereafter. Recall that after the encoding function $g(\cdot; \boldsymbol{\theta}_g)$, we typically obtain a matrix (for brevity, we denote it as $\mathbf{Z}_i \in \mathbb{R}^{T \times d}$) for each $v_i$, in which each row encapsulated the information of a particularly diffusion step. The READOUT function is to generate a vector ($\boldsymbol{h}_i^{tr}$) for each node from this matrix. Similar to the readout function in graph-level tasks (generate a vector representation for the whole graph based on its nodes), potential variants like max-pooling, mean-pooling, or min-pooling can be promptly applied as READOUT. We propose an MLP to achieve this by:

$$\boldsymbol{h}_i^{tr} = \text{SQUEEZE}\left[\text{ReLU}(\mathbf{Z}_i \boldsymbol{W}_1 + \boldsymbol{b}_1)\right] \boldsymbol{W}_2, \tag{45}$$

where $\boldsymbol{W}_1 \in \mathbb{R}^{d \times 1}$ and $\boldsymbol{W}_2 \in \mathbb{R}^{T \times d'}$. The SQUEEZE function transforms the output of the inner layer from size $T \times 1$ to $T$. $d$ and $d'$ are the dimensions of $\mathbf{Z}_i$ and $\boldsymbol{h}_i^{tr}$, respectively. $\boldsymbol{b}_1$ is the bias.

In addition to the Transformer-based model, TRC-TRANS, we propose an even straightforward MLP-based model, which learns $\boldsymbol{h}_i^{tr}$ following:

$$\boldsymbol{h}_i^{tr} = \text{READOUT}[\text{ReLU}(\mathbf{G}_{i,:,:} \boldsymbol{W} + \boldsymbol{b})], \tag{46}$$

while keeping the READOUT function, classifier, and training the same as the Transformer model.

To summarize, from our Observation I, we propose two novel graph anomaly detection methods (i.e., FSC and TRC) by learning anomalies' divergent trajectories in the forward diffusion process. While FSC predicts a hidden state following the trajectory for classifying anomalies, TRC directly learns the representation of each node's evolving trajectory. Both approaches follow our novel paradigm to detect graph anomalies without MP-GNN and achieve even better performance, as presented in §7.

# F    IMPLEMENTATION DETAILS OF DIFFAD AND BASELINES

In this section, we present the diffusion step encoding function $\text{TE}(\cdot)$, non-probabilistic implementations of DIFFAD, followed by implementation details of all baselines.

## F.1    DIFFUSION STEP ENCODING TE

We propose the diffusion step encoding function $\text{TE}(\cdot)$ to involve the step information for training. Practically, we generate a particular embedding matrix for each step $t$, given by:

$$\text{TE}(t) = \text{SeLU}\left[\text{SE}(t)\boldsymbol{W}_1^{\text{TE}}\right]\boldsymbol{W}_2^{\text{TE}}, \tag{47}$$

where $\boldsymbol{W}_1^{\text{TE}}$ and $\boldsymbol{W}_2^{\text{TE}}$ are trainable matrices for projection. $\text{SE}(\cdot)$ is the sinusoidal encoding function and the value of $\text{SE}(t)$'s each dimension is encoded as:

$$\text{SE}(t)_i = \left[\sin\left(t \cdot \exp(\frac{\log 10000}{d/2 - 1} \cdot i)\right), \cos\left(t \cdot \exp(\frac{\log 10000}{d/2 - 1} \cdot i)\right)\right], \tag{48}$$

## F.2    NON-PROBABILISTIC VARIANTS OF DIFFAD

The non-probabilistic implementation of DIFFAD contains non-probabilistic SDN and DEN, which aim to explicitly predict the added noise $\boldsymbol{\epsilon_\theta}$ formulated in Eq. (20), at an arbitrary diffusion step $t$. The non-probabilistic SDN takes $\mathcal{G}_t = \{\mathbf{A}, \mathbf{X}_t\}$ and $t$ as inputs and generates node embeddings by

$$\mathbf{Z}_t = \text{SDN}(\mathbf{A}, \mathbf{X}_t, t) = \text{ReLU}(\tilde{\mathbf{A}}\mathbf{Z}_t'\boldsymbol{W}_2), \quad \mathbf{Z}_t' = \text{ReLU}\left[\tilde{\mathbf{A}}(\mathbf{X}_t + \text{TE}(t))\boldsymbol{W}_1\right] \tag{49}$$

and $\text{TE}(t)$ is the diffusion step encoding function formulated in Eq. (47).

Correspondingly, the DEN takes $\mathbf{A}$, $\mathbf{Z}_t$ and $\mathbf{Z}_t'$ as inputs and predicts the added noise following:

$$\boldsymbol{\epsilon_\theta} = \text{DEN}(\mathbf{Z}_t, \mathbf{Z}_t', \mathbf{A}) = \text{SeLU}\left\{\tilde{\mathbf{A}}\left[\text{ReLU}\left(\tilde{\mathbf{A}}(\mathbf{Z}_t \oplus \mathbf{Z}_t)\boldsymbol{W}_3\right) \oplus \mathbf{Z}_t'\right]\boldsymbol{W}_4\right\}. \tag{50}$$

$\{\boldsymbol{W}_i\}_{i=1}^4$ are trainable variables in DIFFAD. By minimizing $\mathbb{E}_{\mathbf{X},\epsilon}(\|\boldsymbol{\epsilon} - \boldsymbol{\epsilon_\theta}\|_2^2)$, these variables will model the original graph data distribution, empowering DIFFAD to generate auxiliary graph samples that adhere to the distribution of the original graph $\mathcal{G}$.

## F.3    DIFFAD'S GCN DETECTOR

Given the original graph $\mathcal{G}$ and generated graphs $\mathcal{G}_a^i \in \mathbb{G}$, we pass them into a detector, containing two GCN layers and one fully-connected layer, to predict the possibility of each node belonging to anomalies or normal. The two GCN layers can be formulated as:

$$\mathbf{Z} = \text{GCN}(\mathbf{A}, \mathbf{X}) = \text{ReLU}\left(\tilde{\mathbf{A}}\,\text{ReLU}(\tilde{\mathbf{A}}\mathbf{X}\boldsymbol{W}_0)\boldsymbol{W}_1\right), \tag{51}$$

and the fully-connected layer predicts the possibilities following:

$$\psi(v_i \mid y) = \text{MLP}(\boldsymbol{z}_i) = \text{SOFTMAX}\left[\text{ReLU}(z_i\boldsymbol{W}_c + \boldsymbol{b}_c)\right], \tag{52}$$

where denote to the predicted possibility of node $v_i$ being normal ($y = 0$) or anomalous ($y = 1$), respectively. Eventually, we assign a label to each node as $\arg\max_y[\psi(v_i \mid y{=}0), \psi(v_i \mid y{=}1)]$.

## F.4    BASELINE IMPLEMENTATIONS

### F.4.1    CONVENTIONAL GNN-BASED DETECTORS

We have implemented the GCN[2] (Kipf & Welling, 2017), GAT[3] (Veličković et al., 2018), and GraphSAGE[4] (Hamilton et al., 2017) detectors based on their original settings. Specifically, each

---

[2] https://github.com/tkipf/gcn
[3] https://github.com/PetarV-/GAT
[4] https://github.com/williamleif/GraphSAGE

of them stacks a fully connected layer above two GNN layers. The difference is that the GCN detector investigates the graph convolutional neural network layers, the GAT detector adopts the graph attention neural network layers, and the GraphSAGE detector applies the GraphSAGE layers. In our experiment, we follow DIFFAD's configurations and set their GNN layers' dimensions as 256 and 128, respectively. We apply ReLU as the activation function for these three detectors and set the number of training iterations to 200.

### F.4.2 STATE-OF-THE-ART METHODS

We use the published implementations and configurations of GeniePath[5] (Liu et al., 2019), FdGars[5] (Wang et al., 2019), BWGNN[6] (Tang et al., 2022), DAGAD[7] (Liu et al., 2022a), GAT-sep (Zhu et al., 2020), AMNet (Chai et al., 2022), GHRN (Gao et al., 2023a)[8], CONAD[5] (Xu et al., 2022b) and CoLA[5] (Liu et al., 2021) in our experiment. For BWGNN, we report the best results of its two variants (i.e., Homo and Hetero) as published in the original paper (Tang et al., 2022).

## G  RELATION BETWEEN KL DIVERGENCE AND OUR GRAPH DIFFUSION OBJECTIVE

As detailed in §6.1, the learning objective of our proposed graph diffusion process is to minimize the difference between the distributions of the corrupted graph (through the forward graph diffusion) and the recovered graph (through the reverse process). This is formally defined in Eq. (14) as:

$$\arg\min_{\boldsymbol{\theta}} D_{\mathrm{KL}}\big(q(\mathbf{X}_{t-1}|\mathbf{X}_t, \mathbf{X}, \mathbf{A}) \parallel p_{\boldsymbol{\theta}}(\mathbf{X}_{t-1}|\mathbf{X}_t, \mathbf{A})\big). \tag{53}$$

To indicate the relation between this KL divergence and our learning objective formulated in Eq. (20), we first calculate $q(\mathbf{X}_{t-1}|\mathbf{X}_t, \mathbf{X}, \mathbf{A})$ by submitting the Bayes rule:

$$q(\mathbf{X}_{t-1}|\mathbf{X}_t, \mathbf{X}, \mathbf{A}) = \frac{q(\mathbf{X}_t|\mathbf{X}_{t-1}, \mathbf{X}, \mathbf{A})q(\mathbf{X}_{t-1}|\mathbf{X}, \mathbf{A})}{q(\mathbf{X}_t|\mathbf{X}, \mathbf{A})} \tag{54}$$

$$= \frac{\mathcal{N}\big(\mathbf{X}_t; \sqrt{\alpha_t}\mathbf{X}_{t-1}, (1-\alpha_t)\mathbf{I}\big) \cdot \mathcal{N}\big(\mathbf{X}_{t-1}; \sqrt{\bar{\alpha}_{t-1}}\mathbf{X}, (1-\bar{\alpha}_{t-1})\mathbf{I}\big)}{\mathcal{N}\big(\mathbf{X}_t; \sqrt{\bar{\alpha}_t}\mathbf{X}, (1-\bar{\alpha}_t)\mathbf{I}\big)} \tag{55}$$

$$\propto \exp\left\{-\frac{1-\bar{\alpha}_t}{2(1-\alpha_t)(1-\bar{\alpha}_{t-1})}\left[\mathbf{X}_{t-1}^2 - 2\frac{\sqrt{\alpha_t}(1-\bar{\alpha}_{t-1})\mathbf{X}_t + \sqrt{\bar{\alpha}_{t-1}}(1-\alpha_t)\mathbf{X}}{1-\bar{\alpha}_t}\mathbf{X}_{t-1}\right]\right\} \tag{56}$$

$$\propto \mathcal{N}\left(\mathbf{X}_{t-1}; \underbrace{\frac{\sqrt{\alpha_t}(1-\bar{\alpha}_{t-1})\mathbf{X}_t + \sqrt{\bar{\alpha}_{t-1}}(1-\alpha_t)\mathbf{X}}{1-\bar{\alpha}_t}}_{\boldsymbol{\mu}_q(\mathbf{X}_t, \mathbf{X})}, \underbrace{\frac{(1-\alpha_t)(1-\bar{\alpha}_{t-1})}{1-\bar{\alpha}_t}\mathbf{I}}_{\boldsymbol{\sigma}_t^2}\right), \tag{57}$$

We can then rewrite the KL divergence to show that minimizing it is equivalent to: 1) Minimizing the difference between the means of the two distributions; 2) Minimizing the difference between the

---

[5]https://github.com/pygod-team/pygod
[6]https://github.com/squareRoot3/Rethinking-Anomaly-Detection
[7]https://github.com/FanzhenLiu/DAGAD
[8]https://github.com/squareRoot3/GADBench

injected noise and predicted noise. For the first, we have:

$$\arg\min_{\boldsymbol{\theta}} D_{\mathrm{KL}}\big(q(\mathbf{X}_{t-1}|\mathbf{X}_t, \mathbf{X}, \mathbf{A}) \parallel p_{\boldsymbol{\theta}}(\mathbf{X}_{t-1}|\mathbf{X}_t, \mathbf{A})\big) \tag{58}$$

$$= \arg\min_{\boldsymbol{\theta}} D_{\mathrm{KL}}\big(\mathcal{N}(\mathbf{X}_{t-1}; \boldsymbol{\mu}_t, \boldsymbol{\sigma}_t^2), \mathcal{N}(\mathbf{X}_{t-1}; \boldsymbol{\mu}_{\boldsymbol{\theta}}, \boldsymbol{\sigma}_{\boldsymbol{\theta}}^2)\big) \tag{59}$$

$$= \arg\min_{\boldsymbol{\theta}} D_{\mathrm{KL}}\big(\mathcal{N}(\mathbf{X}_{t-1}; \boldsymbol{\mu}_t, \boldsymbol{\sigma}_t^2), \mathcal{N}(\mathbf{X}_{t-1}; \boldsymbol{\mu}_{\boldsymbol{\theta}}, \boldsymbol{\sigma}_t^2)\big) \tag{60}$$

$$= \arg\min_{\boldsymbol{\theta}} \frac{1}{2}\left[\log\frac{\boldsymbol{\sigma}_t^2}{\boldsymbol{\sigma}_t^2} - d + \mathrm{tr}((\boldsymbol{\sigma}_t^2)^{-1}\boldsymbol{\sigma}_t^2) + (\boldsymbol{\mu}_{\boldsymbol{\theta}} - \boldsymbol{\mu}_t)^\top (\boldsymbol{\sigma}_t^2)^{-1}(\boldsymbol{\mu}_{\boldsymbol{\theta}} - \boldsymbol{\mu}_t)\right] \tag{61}$$

$$= \arg\min_{\boldsymbol{\theta}} \frac{1}{2}\left[(\boldsymbol{\mu}_{\boldsymbol{\theta}} - \boldsymbol{\mu}_t)^\top (\boldsymbol{\sigma}_t^2)^{-1}(\boldsymbol{\mu}_{\boldsymbol{\theta}} - \boldsymbol{\mu}_t)\right] \tag{62}$$

$$= \arg\min_{\boldsymbol{\theta}} \frac{1}{2\sigma_t^2}\left[||\boldsymbol{\mu}_{\boldsymbol{\theta}} - \boldsymbol{\mu}_t||_2^2\right], \tag{63}$$

where in Line 59, we can directly calculate $\boldsymbol{\sigma}_{\boldsymbol{\theta}}^2$ at a specific time step from the noise variance scheduler $\beta_t$ as in Line 57. Therefore, it is replaced by $\boldsymbol{\sigma}_t^2$ in Line 60. Line 61 is based on the definition of KL divergence between two Gaussian distributions. As such, we show that learning to minimize the KL divergence is practically equivalent to minimizing the difference between the means of the two distributions. In our implementation, $\boldsymbol{\mu}_{\boldsymbol{\theta}}$ is the output of DEN, $\boldsymbol{\mu}^{\mathrm{DEN}}$ (given by Eq. (18)). Similarly, for the second, we have:

$$\arg\min_{\boldsymbol{\theta}} D_{\mathrm{KL}}\big(q(\mathbf{X}_{t-1}|\mathbf{X}_t, \mathbf{X}, \mathbf{A}) \parallel p_{\boldsymbol{\theta}}(\mathbf{X}_{t-1}|\mathbf{X}_t, \mathbf{A})\big) \tag{64}$$

$$= \arg\min_{\boldsymbol{\theta}} D_{\mathrm{KL}}\big(\mathcal{N}(\mathbf{X}_{t-1}; \boldsymbol{\mu}_t, \boldsymbol{\sigma}_t^2), \mathcal{N}(\mathbf{X}_{t-1}; \boldsymbol{\mu}_{\boldsymbol{\theta}}, \boldsymbol{\sigma}_{\boldsymbol{\theta}}^2)\big) \tag{65}$$

$$= \arg\min_{\boldsymbol{\theta}} D_{\mathrm{KL}}\big(\mathcal{N}(\mathbf{X}_{t-1}; \boldsymbol{\mu}_t, \boldsymbol{\sigma}_t^2), \mathcal{N}(\mathbf{X}_{t-1}; \boldsymbol{\mu}_{\boldsymbol{\theta}}, \boldsymbol{\sigma}_t^2)\big) \tag{66}$$

$$= \arg\min_{\boldsymbol{\theta}} \frac{1}{2\boldsymbol{\sigma}_t^2}\left[||\frac{1}{\sqrt{\alpha_t}}\mathbf{X}_t - \frac{1-\alpha_t}{\sqrt{1-\bar{\alpha}_t}\sqrt{\alpha_t}}\boldsymbol{\epsilon}_{\boldsymbol{\theta}}(\mathbf{X}_t, \mathbf{A}, t) - \frac{1}{\sqrt{\alpha_t}}\mathbf{X}_t + \frac{1-\alpha_t}{\sqrt{1-\bar{\alpha}_t}\sqrt{\alpha_t}}\boldsymbol{\epsilon}||_2^2\right] \tag{67}$$

$$= \arg\min_{\boldsymbol{\theta}} \frac{1}{2\boldsymbol{\sigma}_t^2}\left[||\frac{1-\alpha_t}{\sqrt{1-\bar{\alpha}_t}\sqrt{\alpha_t}}\big(\boldsymbol{\epsilon} - \boldsymbol{\epsilon}_{\boldsymbol{\theta}}(\mathbf{X}_t, \mathbf{A}, t)\big)||_2^2\right] \tag{68}$$

$$= \arg\min_{\boldsymbol{\theta}} \frac{1}{2\boldsymbol{\sigma}_t^2}\frac{(1-\alpha_t)^2}{(1-\bar{\alpha}_t)(\alpha_t)}\left[||\boldsymbol{\epsilon} - \boldsymbol{\epsilon}_{\boldsymbol{\theta}}(\mathbf{X}_t, \mathbf{A}, t)||_2^2\right] \tag{69}$$

$$\propto \arg\min_{\boldsymbol{\theta}}\left[||\boldsymbol{\epsilon} - \boldsymbol{\epsilon}_{\boldsymbol{\theta}}(\mathbf{X}_t, \mathbf{A}, t)||_2^2\right], \tag{70}$$

Hence, we can see that learning to minimize the KL divergence is also equivalent to minimizing the error between the added noise $\boldsymbol{\epsilon}$ and the predicted noise $\boldsymbol{\epsilon}_{\boldsymbol{\theta}}(\mathbf{X}_t, \mathbf{A}, t)$. For interested readers, we refer to two comprehensive reviews Luo (2022) and Yang et al. (2022).

## H    GENERATIVE ALGORITHM

After training SDN and DEN sufficiently, we can then randomly sample $\mathbf{X}_t \sim \mathcal{N}(\mathbf{0}, \mathbf{I})$ and reverse the process (as summarized in Algorithm 1) progressively to obtain a new graph $\mathcal{G}_a$ following Eq. (21).

As given in Eq. (57), we can get $\mathbf{X}_{t-1}$ from $\mathbf{X}_t$ and $\mathbf{X}$ following:

$$\mathbf{X}_{t-1} = \frac{\sqrt{\alpha_t}(1 - \bar{\alpha}_{t-1})\mathbf{X}_t + \sqrt{\bar{\alpha}_{t-1}}(1 - \alpha_t)\mathbf{X}}{1 - \bar{\alpha}_t} + \sqrt{\frac{(1 - \alpha_t)(1 - \bar{\alpha}_{t-1})}{1 - \bar{\alpha}_t}}\boldsymbol{\epsilon}^* \tag{71}$$

$$= \frac{\sqrt{\alpha_t}(1 - \bar{\alpha}_{t-1})\mathbf{X}_t + \sqrt{\bar{\alpha}_{t-1}}(1 - \alpha_t)\mathbf{X}}{1 - \bar{\alpha}_t} + \sqrt{\frac{(1 - \bar{\alpha}_{t-1})\beta_t}{1 - \bar{\alpha}_t}}\boldsymbol{\epsilon}^* \tag{72}$$

$$= \frac{\sqrt{\alpha_t}(1 - \bar{\alpha}_{t-1})}{1 - \bar{\alpha}_t}\mathbf{X}_t + \frac{1 - \alpha_t}{1 - \bar{\alpha}_t}\frac{\mathbf{X}_t - \sqrt{1 - \bar{\alpha}_t}\boldsymbol{\epsilon}}{\sqrt{\alpha_t}} + \boldsymbol{\sigma}_t\boldsymbol{\epsilon}^* \tag{73}$$

$$= \frac{1}{\sqrt{\alpha_t}}\mathbf{X}_t - \frac{1 - \alpha_t}{\sqrt{1 - \bar{\alpha}_t}\sqrt{\alpha_t}}\boldsymbol{\epsilon} + \boldsymbol{\sigma}_t\boldsymbol{\epsilon}^* \tag{74}$$

$$= \frac{1}{\sqrt{\alpha_t}}\left(\mathbf{X}_t - \frac{1 - \alpha_t}{\sqrt{1 - \bar{\alpha}_t}}\boldsymbol{\epsilon}_{\boldsymbol{\theta}}(\mathbf{X}_t, \mathbf{A}, t)\right) + \boldsymbol{\sigma}_t\boldsymbol{\epsilon}^*, \quad \text{with} \quad \boldsymbol{\epsilon}^* \sim \mathcal{N}(\mathbf{0}, \mathbf{I}), \tag{75}$$

by applying the reparameterization trick and estimating the added nopise $\boldsymbol{\epsilon}$ using SDN and DEN, given by $\boldsymbol{\epsilon}_{\boldsymbol{\theta}}(\mathbf{X}_t, \mathbf{A}, t)$. $\boldsymbol{\epsilon}^* \sim \mathcal{N}(\mathbf{0}, \mathbf{I})$. In our implementation, we directly assign $\sqrt{\frac{(1-\alpha_t)(1-\bar{\alpha}_{t-1})}{1-\bar{\alpha}_t}}$ to $\boldsymbol{\sigma}_t$ for calculation.

---

**Algorithm 1:** DIFFAD

---

**Input:** Graph $\mathcal{G} = \{\mathbf{A}, \mathbf{X}\}$, $\boldsymbol{y}_{\text{train}}$, diffusion parameters $\beta$, $\alpha$, $\bar{\alpha}$.
**Output:** Generated graphs $\mathcal{G}_a$ and predicted label $y'$ for each node

1 **while** *Not Converged* **do**
2      //*Train the generative model*
3      $t \sim \text{Uniform}(1, \cdots, T)$
4      $\boldsymbol{\epsilon}_t \sim \mathcal{N}(\mathbf{0}, \mathbf{I})$
5      $\mathbf{Z}_t, \mathbf{Z}'_t \leftarrow \text{SDN}(\mathbf{X}_t, \mathbf{A}, t)$
6      $\boldsymbol{\epsilon}'_t \leftarrow \text{DEN}(\mathbf{Z}_t, \mathbf{Z}'_t, \mathbf{A})$
7      Update parameters in SDN and DEN with Gradients
8          $\nabla_{\boldsymbol{W}} \mathbb{E}_{\mathbf{X}, \epsilon}(||\boldsymbol{\epsilon}_t - \boldsymbol{\epsilon}'_t||_2^2)$
9 **end**
10 $\mathbf{X}_{a,T} \sim \mathcal{N}(\mathbf{0}, \mathbf{I})$
11 **for** $t \leftarrow T$ **to** 1 **do**
12      //*Generate graphs*
13      $\boldsymbol{\epsilon} \sim \mathcal{N}(\mathbf{0}, \mathbf{I})$
14      $\mathbf{Z}_t, \mathbf{Z}'_t \leftarrow \text{SDN}(\mathbf{X}_{a,t}, \mathbf{A}, t)$
15      $\boldsymbol{\epsilon}' \leftarrow \text{DEN}(\mathbf{Z}_t, \mathbf{Z}'_t, \mathbf{A})$
16      $\mathbf{X}_{a,t-1} \leftarrow \frac{1}{\sqrt{\alpha_t}}\left(\mathbf{X}_{a,t} - \frac{1-\alpha_t}{\sqrt{1-\bar{\alpha}_t}}\boldsymbol{\epsilon}'\right) + \sqrt{\frac{(1-\alpha_t)(1-\bar{\alpha}_{t-1})}{1-\bar{\alpha}_t}}\boldsymbol{\epsilon}$
17 **end**
18 $\mathbf{X}_a \leftarrow \mathbf{X}_{a,0}$
19 $\mathcal{G}_a \leftarrow \{\mathbf{A}, \mathbf{X}_a\}$
20 **while** *Not Converged* **do**
21      //*Train the classifier*
22      $\psi(v_i \mid y) \leftarrow \text{GCN}(\mathbf{A}, \mathbf{X}, \boldsymbol{y}_{\text{train}})$
23      $\psi(v_i^a \mid y) \leftarrow \text{GCN}(\mathbf{A}, \mathbf{X}_a, \boldsymbol{y}_{\text{train}})$
24      Update parameters in GCN with Gradients measured by Eq. (22)
25 **end**
26 $y'_i \leftarrow \arg\max_y[\psi(v_i \mid y{=}0), \psi(v_i \mid y{=}1)]$      //*prediction*
27 **return** $\mathcal{G}_a, y'_i$

---

## I   SUMMARY OF ALL ALGORITHMS AND COMPLEXITY ANALYSIS

The algorithms for our proposed methods, namely DIFFAD, FSC, TRC-MLP, TRC-TRANS are summarized in Algorithm 1 and Algorithm 2, respectively. As can be seen, DIFFAD encompasses three stages, i.e., training the generative model, graph generation and training the classifier. For training the generative model, we take an arbitrary time step $t$ from $\{0, \ldots, T\}$ and train the SDN and

DEN to predict the added noise $\epsilon_t$ with regard to $\mathbf{X}_t$, $\mathbf{A}$ and $t$. All the parameters in the generative model are fine-tuned until the training objective, as formulated in Eq. (20), converges. We then generate an auxiliary graph $\mathcal{G}_a$ by reversing the diffusion process and this generated graph is utilized to train the classifier together with the original graph $\mathcal{G}$ (Lines 18-25). Different from DIFFAD, FSC, TRC-MLP, TRC-TRANS involve no reverse diffusion process. All these three methods take the tensor $\mathbf{G}$ as input, which can be obtained through the forward diffusion process, and employ different neural networks, i.e., MLP, LSTM and Transformer, to classify nodes as anomalous or normal with regard to the evolving patterns.

## I.1 COMPLEXITY ANALYSIS

*The complexity of* DIFFAD. The primary computational cost of DIFFAD stems from the training of the generative model, graph generation, and classifier training. In the generative model training, as both SDN and DEN adopt GCN as the backbone, the cost of each training iteration is approximately $\mathcal{O}(4nkd)$, where $n$ is the number of nodes in $\mathcal{G}$, $k$ is the dimensionality of node attributes, and $d$ is the dimension of the GCN layers. The graph generation stage involves SDN and DEN for inference, and the total cost of generating one graph is about $\mathcal{O}(4nkdT)$, with $T$ diffusion steps. Lastly, the cost of each iteration for training the GCN classifier approximates $\mathcal{O}(2nkd)$. In practice, the diffusion step is commonly set to 1000, while the training iterations of SDN, DEN, and the classifier are less than 200. Hence, the overall cost of the three stages in DIFFAD is approximately $\mathcal{O}(4.5nkdT) \sim \mathcal{O}(nkdT)$.

*The complexity of* FSC. As summarized in Algorithm 2, the cost of each training iteration arises from the computation of the states, i.e., $\boldsymbol{h}_t^i$, $\boldsymbol{c}_t^i$, $\boldsymbol{\tau}_t^i$, $\boldsymbol{f}_t^i$, $\boldsymbol{g}_t^i$ and $\boldsymbol{o}_t^i$. Assuming the dimension of the neural networks layers in FSC is $d$, this cost is approximately $\mathcal{O}(8nkdT + 2nd) \sim \mathcal{O}(nkdT)$, where $\mathcal{O}(2nd)$ is the cost for training the classifier in Line 19.

*The complexity of* TRC-MLP. TRC-MLP only involves multiple layers of fully connected neural network. From Lines 26-33 in Algorithm 2, its cost in each training iteration is approximately $\mathcal{O}(nkdT)$.

*The complexity of* TRC-TRANS. TRC-TRANS applies the $\mathbf{Q}$, $\mathbf{K}$, and $\mathbf{V}$ attention blocks for learning the representation of each node's evolving trajectory. In each training iteration, the computational cost of Lines 38-40 is about $\mathcal{O}(3nkdT)$, where $W_Q, W_K, W_V \in \mathbf{R}^{k \times d}$. The cost for computing the self-attention in Line 41 is $\mathcal{O}(2ndT^2)$ for the whole graph. For reading out the trajectory representation in Line 42, its cost is approximately $\mathcal{O}(ndT)$, and the classifier training cost in Line 42 is $\mathcal{O}(2nT)$. Therefore, the overall cost of TRC-TRANS is similar to the conventional Transformers, which is quadratic to the diffusion steps $T$.

## J EXPERIMENT DETAILS AND ADDITIONAL RESULTS

### J.1 DATASETS

The nine graph anomaly detection datasets can be categorized into two groups: one with organic anomalies, including YelpChi (Rayana & Akoglu, 2015), Reddit (Wang et al., 2021a), Weibo (Zhao et al., 2020), Tfinance (Tang et al., 2022), Tolokers (Tang et al., 2023) and Questions (Tang et al., 2023); and another with injected anomalies, encompassing BlogCatalog (Ding et al., 2019), ACM (Ding et al., 2019), and Cora (Liu et al., 2022b). The statistics of these datasets are summarized in Table 3 and they are available in our GitHub repository[9].

**Organic datasets.**

*YelpChi* (Rayana & Akoglu, 2015) is an online review dataset with spam and legitimate reviews of hotels and restaurants. All reviews are labeled by Yelp. There are three types of relations in this dataset and for a fair comparison, we test each baseline on all these relations and report the best performance. Table 3 reports the total number of edges.

*Reddit* (Wang et al., 2021a) compromises collected posts from subreddit. The ground-truth anomalies are banned users from Reddit.

---

[9] https://github.com/DiffAD/DiffAD

**Algorithm 2:** FSC, TRC-MLP, TRC-TRANS

**Input:** Graph $\mathcal{G} = \{\mathbf{A}, \mathbf{X}\}$, $\boldsymbol{y}_{\text{train}}$, diffusion parameters $\beta, \alpha, \bar{\alpha}$.
**Output:** Predicted label $y'_i$ for each node

1  Calculate graph Laplacian $\mathbf{L}$
2  **for** $t \leftarrow 1$ **to** $T$ **do**
3      *//Store $\mathcal{G}$ as a tensor* $\mathbf{G}$
4      $\epsilon \sim \mathcal{N}(\mathbf{0}, \mathbf{I})$
5      $\mathbf{X}_t \leftarrow \sqrt{\bar{\alpha}_t}\mathbf{X} + (1 - \bar{\alpha}_t)\epsilon$
6      $\mathbf{G}_{:,t,:} \leftarrow \mathbf{L}\mathbf{X}_t$
7  **end**
8  // FSC
9  **if** *method is* FSC **then**
10     **while** *Not Converged* **do**
11         **for** *each node $v_i$, $t \leftarrow T$* **to** *0* **do**
12             $\boldsymbol{\tau}_t^i \leftarrow tanh(\boldsymbol{W}_\tau \mathbf{G}_{i,t,:} + \mathbf{U}_\tau \boldsymbol{h}_{t+1}^i + \boldsymbol{b}_\tau)$
13             $\boldsymbol{f}_t^i \leftarrow tanh(\boldsymbol{W}_f \mathbf{G}_{i,t,:} + \mathbf{U}_f \boldsymbol{h}_{t+1}^i + \boldsymbol{b}_f)$
14             $\boldsymbol{g}_t^i \leftarrow tanh(\boldsymbol{W}_g \mathbf{G}_{i,t,:} + \mathbf{U}_g \boldsymbol{h}_{t+1}^i + \boldsymbol{b}_g)$
15             $\boldsymbol{o}_t^i \leftarrow tanh(\boldsymbol{W}_o \mathbf{G}_{i,t,:} + \mathbf{U}_o \boldsymbol{h}_{t+1}^i + \boldsymbol{b}_o)$
16             $\boldsymbol{c}_t^i \leftarrow \boldsymbol{f}_t^i \odot \boldsymbol{c}_{t+1}^i + \boldsymbol{\tau}_t^i \odot \boldsymbol{g}_t^j$
17             $\boldsymbol{h}_t^i \leftarrow \boldsymbol{o}_t^i \odot \sigma(\boldsymbol{f}_t^i \odot \boldsymbol{c}_{t+1}^i + \boldsymbol{\tau}_t^i \odot \boldsymbol{g}_t^i)$
18         **end**
19         $\psi(v_i \mid y) \leftarrow \text{SOFTMAX}(h_0^i \boldsymbol{W}_c + \boldsymbol{b}_c)$
20         Take gradient step based on Eq. (10)
21     **end**
22     $y'_i \leftarrow \arg\max_y[\psi(v_i \mid y{=}0), \psi(v_i \mid y{=}1)]$     *//prediction*
23     **return** $y'_i$
24 **end**
25 // TRC-MLP
26 **if** *method is* TRC-MLP **then**
27     **while** *Not Converged* **do**
28         $h_i^{tr} \leftarrow \text{READOUT}[\text{ReLU}(\mathbf{G}_{i,:,:}\boldsymbol{W} + \boldsymbol{b})]$
29         $\psi(v_i \mid y) \leftarrow \text{SOFTMAX}(h_i^{tr}\boldsymbol{W}_c + \boldsymbol{b}_c)$
30         Take gradient step based on Eq. (10)
31     **end**
32     $y'_j \leftarrow \arg\max_y[\psi(v_j \mid y{=}0), \psi(v_j \mid y{=}1)]$     *//prediction*
33     **return** $y'_i$
34 **end**
35 // TRC-TRANS
36 **if** *method is* TRC-TRANS **then**
37     **while** *Not Converged* **do**
38         $\mathbf{Q}_i \leftarrow \mathbf{G}'_{i,:,:}\boldsymbol{W}_Q$
39         $\mathbf{K}_i \leftarrow \mathbf{G}'_{i,:,:}\boldsymbol{W}_K$
40         $\mathbf{V}_i \leftarrow \mathbf{G}'_{i,:,:}\boldsymbol{W}_V$
41         $\text{ATTN}(\mathbf{G}_{i,:,:}) \leftarrow \text{SOFTMAX}(\frac{\mathbf{Q}_i \mathbf{K}_i^T}{\sqrt{k}})\mathbf{V}_i$
42         $\boldsymbol{h}_i^{tr} \leftarrow \text{READOUT}\{\text{FFN}[\text{ATTN}(\mathbf{G}_{i,:,:})]\}$
43         $\psi(v_i \mid y) \leftarrow \text{SOFTMAX}(\boldsymbol{h}_i^{tr}\boldsymbol{W}_c + \boldsymbol{b}_c)$
44         Take gradient step based on Eq. (10)
45     **end**
46     $y'_j \leftarrow \arg\max_y[\psi(v_j \mid y{=}0), \psi(v_j \mid y{=}1)]$     *//prediction*
47     **return** $y'_i$
48 **end**

Table 3: Dataset statistics

| Dataset | #Nodes | #Edges | #Features | #Degree | #Anomalies | Anomaly Ratio |
|---------|--------|--------|-----------|---------|------------|---------------|
| YelpChi | 45,954 | 3,846,979 | 32 | 83.7 | 6677 | 16.9% |
| Reddit | 10,984 | 168,016 | 64 | 15.3 | 366 | 3.3% |
| Weibo | 8,405 | 407,963 | 400 | 48.5 | 868 | 10.3% |
| Tfinance | 39,357 | 21,222,543 | 10 | 539.2 | 1810 | 4.6% |
| Tolokers | 11,758 | 519,000 | 10 | 44.1 | 2563 | 21.8% |
| Questions | 48,921 | 153,540 | 301 | 3.1 | 1467 | 3% |
| BlogCatalog | 5,196 | 172,759 | 8,189 | 33.2 | 298 | 5.7% |
| ACM | 16,484 | 74,073 | 8,337 | 4.5 | 597 | 3.6% |
| Cora | 2,708 | 11,060 | 1,433 | 4.1 | 138 | 5.1% |

*Weibo* (Zhao et al., 2020) is collected from the Tencent-Weibo platform. The ground-truth anomalies are users who have engaged in more than five suspicious events.

*Tfinance* (Tang et al., 2022) is an online review dataset with spam and legitimate reviews of hotels and restaurants. All reviews are labeled by Yelp. There are three types of relations in this dataset and for a fair comparison, we test each baseline on all these relations and report the best performance. Table 3 reports the total number of edges.

*Tolokers* (Tang et al., 2023) is collected from the Toloka crowdsourcing platform (Platonov et al., 2023). In this dataset, anomalies denote workers who have been banned in at least one project.

*Questions* (Tang et al., 2023) is from the Yandex Q website. Users of this question-answering website are denoted as nodes and an edge is built between two users if one user's question has been answered by another (Platonov et al., 2023). Anomalies are users that are no longer active after August 2022.

**Synthetic datasets.** These three synthetic datasets are downloaded from site.[10]

*BlogCatalog* (Tang & Liu, 2009) is collected from an online blog sharing network. The edges denote the follower-followee relations among users and tags associated with users are node attributes.

*ACM* (Tang et al., 2008) is built based on the ACM scientific citation network, in which publications are represented as nodes and their citation relations are represented as edges.

*Cora* (Liu et al., 2022b) is another scientific citation network containing publications and their citation links.

### J.2 EXPERIMENTAL SETTING AND IMPLEMENTATION DETAILS

We measure the performance of all methods with regard to four commonly used anomaly detection metrics, namely **M**acro-**F1**, **M**acro-**Pre**cision, **M**acro-**Rec**all, and AUC (Ma et al., 2021). The three macro-level metrics provide a balanced evaluation of the detection performance on both anomalies and normal nodes, while AUC measures the area under the ROC curve.

We compare our methods, FSC, TRC-MLP, TRC-TRANS, and DIFFAD, with three GNN detectors and seven state-of-the-art semi-/supervised anomaly detectors. The three GNN detectors are built upon GCN (Kipf & Welling, 2017), GAT (Veličković et al., 2018), and GraphSAGE (Hamilton et al., 2017), respectively. Seven state-of-the-art detectors include GeniePath (Liu et al., 2019), FdGars (Wang et al., 2019), BWGNN (Tang et al., 2022), DAGAD (Liu et al., 2022a) GAT-sep (Zhu et al., 2020), AMNet (Chai et al., 2022), GHRN (Gao et al., 2023a) and two contrastive detectors, namely CONAD (Xu et al., 2022b) and CoLA (Liu et al., 2021). There are other baselines proposed before BWGNN and DAGAD, such as CARE-GNN (Dou et al., 2020) and GraphConsis (Liu et al., 2020), but we do not include them since BWGNN and DAGAD have demonstrated superiority over them.

In our experiment, we follow the published configurations of all methods. Implementation details of baselines and our method are provided in the prior section in Appendix F. We used a training ratio of

---

[10]https://github.com/pygod-team/data

Table 4: Detection results on nine datasets (best in **bold**).

| Method | YelpChi | | | | Reddit | | | | Weibo | | | |
|---|---|---|---|---|---|---|---|---|---|---|---|---|
| | M-F1 | AUC | M-Pre | M-Rec | M-F1 | AUC | M-Pre | M-Rec | M-F1 | AUC | M-Pre | M-Rec |
| GCN | 46.08±0.1 | 57.09±0.1 | 42.73±0.1 | 50.00±0.1 | 49.15±0.1 | 57.74±0.1 | 48.33±0.1 | 50.00±0.1 | 82.27±0.1 | 85.64±0.1 | 82.38±0.1 | 82.18±0.1 |
| GAT | 46.98±0.1 | 58.24±0.1 | 70.97±1.6 | 50.24±0.1 | 49.24±0.1 | 64.45±0.2 | 48.34±0.1 | 50.00±0.1 | 85.62±0.1 | 79.86±0.9 | 88.57±0.2 | 83.22±0.1 |
| GraphSAGE | 60.86±0.2 | 80.36±0.1 | 75.11±0.2 | 58.62±0.2 | 49.15±0.1 | 51.31±0.2 | 48.33±0.1 | 50.00±0.1 | 89.20±0.1 | 88.35±0.2 | 90.16±0.1 | 88.43±0.2 |
| GeniePath | 46.08±0.1 | 48.74±0.1 | 42.73±0.1 | 49.97±0.1 | 49.15±0.1 | 46.30±0.1 | 48.80±0.1 | 50.30±0.1 | 87.65±0.1 | 93.11±0.6 | 88.68±0.1 | 86.90±0.3 |
| FdGars | 49.77±0.1 | 53.01±0.5 | 51.65±0.1 | 51.55±0.2 | 48.52±0.2 | 63.05±0.1 | 50.80±0.1 | 53.57±0.3 | 89.27±0.3 | 92.29±0.3 | 88.82±0.1 | 89.77±0.1 |
| BWGNN | 63.68±0.3 | 80.96±0.1 | 63.30±0.1 | 72.84±0.1 | 43.29±0.5 | 66.89±0.3 | 51.55±0.1 | 60.47±0.3 | 89.63±0.1 | 91.05±0.1 | 88.74±0.1 | 87.54±0.1 |
| DAGAD | 52.08±0.2 | 59.83±0.1 | 60.81±0.1 | 52.74±0.1 | 49.15±0.1 | 61.49±0.3 | 48.71±0.1 | 50.22±0.1 | 91.92±0.2 | 95.71±0.1 | 85.95±0.2 | 93.98±0.1 |
| GAT-sep | 65.93±0.3 | 80.01±0.3 | 72.10±0.1 | 63.74±0.4 | 49.16±0.1 | 50.22±0.5 | 48.33±0.1 | 50.01±0.1 | 91.63±0.1 | 97.11±0.1 | 86.86±0.7 | 89.56±0.2 |
| AMNet | 54.66±0.8 | 64.01±1.2 | 55.68±1.3 | 55.51±0.6 | 50.39±0.1 | 62.14±0.3 | 50.94±0.2 | 50.82±0.1 | 89.26±0.1 | 91.78±0.2 | 89.11±0.1 | 89.43±0.1 |
| GHRN | 65.59±0.1 | 81.92±0.1 | 64.28±0.1 | 74.14±0.3 | 45.60±0.3 | 66.09±0.4 | 51.57±0.1 | 59.79±0.3 | 79.01±0.1 | 90.40±0.1 | 74.91±0.1 | 88.25±0.1 |
| CONAD | 47.42±0.1 | 47.50±0.2 | 47.35±0.1 | 45.51±0.1 | 46.39±0.1 | 55.78±0.2 | 49.58±0.1 | 48.02±0.1 | 49.90±0.2 | 71.59±0.4 | 50.53±0.1 | 50.80±0.2 |
| CoLA | 45.82±0.1 | 61.60±0.1 | 46.26±0.1 | 45.54±0.1 | 46.09±0.3 | 50.26±0.3 | 49.48±0.1 | 50.43±0.2 | | | | |
| FSC | 55.36±0.1 | 75.18±0.1 | 61.34±0.1 | 72.77±0.1 | 50.88±0.1 | 57.33±0.1 | 54.36±0.5 | 55.97±0.1 | 90.83±0.1 | 98.11±0.1 | 91.49±0.1 | 92.83±0.1 |
| FSC(linear) | 54.78±0.1 | 69.98±0.1 | 59.94±0.1 | 69.96±0.1 | 51.19±0.1 | 56.54±0.1 | 53.58±0.3 | 55.45±0.1 | 90.35±0.1 | 97.80±0.1 | 90.56±0.1 | 93.32±0.1 |
| TRC-MLP | 55.36±0.1 | 75.64±0.1 | 63.13±0.2 | 73.12±0.1 | 50.23±0.1 | 58.41±0.1 | 53.88±0.4 | 56.60±0.1 | 90.75±0.1 | 95.80±0.1 | 90.80±0.1 | 92.90±0.1 |
| TRC-MLP(linear) | 54.77±0.1 | 72.13±0.1 | 59.67±0.1 | 69.62±0.1 | 49.63±0.1 | 55.76±0.1 | 51.07±0.1 | 54.93±0.1 | 86.04±0.2 | 91.79±0.2 | 88.69±0.2 | 91.55±0.1 |
| TRC-TRANS | 56.58±0.1 | 72.83±0.2 | 61.01±0.1 | 72.10±0.1 | 48.21±0.2 | 57.11±0.1 | 50.81±0.1 | 56.19±0.1 | **92.06±0.1** | **98.17±0.1** | **92.55±0.1** | 94.66±0.1 |
| TRC-TRANS(linear) | 55.88±0.1 | 72.56±0.2 | 60.93±0.1 | 71.90±0.1 | 44.85±0.4 | 54.94±0.1 | 50.67±0.1 | 54.62±0.1 | 92.05±0.1 | 95.05±0.1 | 92.64±0.1 | **95.05±0.1** |
| DIFFAD | **73.88±0.1** | **87.94±0.1** | **77.81±0.8** | **80.98±0.1** | **51.85±0.1** | **71.20±0.1** | **54.36±0.3** | **66.35±0.1** | 90.58±0.1 | 95.46±0.5 | 92.14±0.9 | 91.55±0.9 |

| Method | Tfinance | | | | Tolokers | | | | Questions | | | |
|---|---|---|---|---|---|---|---|---|---|---|---|---|
| | M-F1 | AUC | M-Pre | M-Rec | M-F1 | AUC | M-Pre | M-Rec | M-F1 | AUC | M-Pre | M-Rec |
| GCN | 72.68±0.1 | 81.10±0.4 | 84.94±0.8 | 71.49±0.1 | 55.80±0.2 | 71.69±0.1 | 60.86±0.1 | 55.87±0.1 | 53.00±0.1 | 52.76±0.1 | 53.67±0.5 | 52.04±0.1 |
| GAT | 48.52±0.2 | 54.60±0.5 | 49.73±0.2 | 52.17±0.3 | 55.43±0.2 | 71.97±0.1 | 60.07±0.1 | 55.69±0.2 | 50.42±0.2 | 54.70±0.1 | 52.92±0.1 | 50.64±0.1 |
| GraphSAGE | 70.97±0.1 | 43.51±0.4 | 85.63±0.1 | 65.99±0.1 | 55.66±0.2 | 72.31±0.1 | 60.43±0.1 | 55.73±0.1 | 56.37±0.1 | 51.42±0.1 | 54.44±0.1 | 54.38±0.1 |
| GeniePath | 48.84±0.1 | 53.65±0.1 | 47.71±0.1 | 51.02±0.2 | 43.88±0.1 | 49.22±0.1 | 39.09±0.2 | 50.13±0.1 | 49.24±0.1 | 49.71±0.1 | 48.51±0.1 | 50.11±0.1 |
| FdGars | 81.86±0.1 | 78.29±0.1 | 54.55±0.1 | 70.76±0.1 | 54.05±0.1 | 52.76±0.2 | 54.19±0.1 | 53.96±0.1 | 48.93±0.1 | 54.18±0.1 | 50.54±0.1 | 54.39±0.1 |
| BWGNN | 85.28±0.7 | 91.91±0.3 | 91.54±0.5 | 78.17±0.8 | 57.35±0.9 | 64.14±1.6 | 57.29±1.1 | 58.06±0.6 | 56.45±0.1 | 56.15±0.1 | 56.17±0.1 | 56.82±0.1 |
| DAGAD | 70.89±0.6 | 88.05±0.2 | 72.18±1.1 | 78.64±0.6 | 63.10±0.1 | 72.67±0.1 | 61.07±0.1 | 64.09±0.2 | 55.39±0.1 | 60.57±0.1 | 54.61±0.1 | 56.99±0.1 |
| GAT-sep | 83.20±0.1 | 91.56±0.1 | 85.28±0.1 | 79.42±0.1 | 63.12±0.7 | 72.06±0.4 | 60.56±0.4 | 63.14±0.7 | 56.05±0.1 | 70.18±0.1 | 55.19±0.3 | 57.37±0.1 |
| AMnet | 82.98±0.1 | 91.17±0.4 | 86.65±0.2 | 76.88±0.2 | 56.74±0.5 | 65.33±0.1 | 57.03±0.1 | 57.89±0.2 | 55.63±0.1 | 61.86±0.1 | 55.67±0.1 | 56.86±0.1 |
| GHRN | 81.64±1.1 | 91.89±0.4 | 86.79±1.2 | 78.66±0.8 | 60.56±0.5 | 71.27±1.2 | 61.78±0.1 | 59.86±0.5 | 56.81±0.1 | 56.14±0.1 | 56.44±0.1 | 57.32±0.1 |
| CONAD | 43.84±0.1 | 82.24±0.1 | 47.38±0.1 | 41.09±0.1 | 46.54±0.1 | 61.24±0.1 | 46.37±0.1 | 47.13±0.5 | 48.64±0.1 | 50.36±0.1 | 50.54±0.1 | 52.37±0.1 |
| CoLA | 43.77±0.1 | 57.68±0.2 | 47.59±0.1 | 41.44±0.2 | 50.78±0.1 | 55.45±0.1 | 51.10±0.1 | 50.91±0.1 | 47.24±0.1 | 56.97±0.1 | 50.42±0.1 | 52.28±0.1 |
| FSC | 82.70±0.2 | 89.04±0.1 | 86.63±0.4 | 78.73±0.2 | 61.90±0.1 | 70.01±0.1 | 59.26±0.1 | 63.23±0.1 | 56.34±0.1 | 67.80±0.1 | 54.82±0.1 | 63.95±0.1 |
| FSC(linear) | 81.67±0.1 | 86.32±0.1 | 79.57±0.9 | 76.71±0.1 | 58.90±0.1 | 68.43±0.1 | 59.32±0.1 | 63.56±0.1 | 55.79±0.1 | 67.94±0.1 | 54.59±0.1 | 63.20±0.1 |
| TRC-MLP | 54.41±0.2 | 64.28±0.3 | 71.03±0.2 | 62.94±0.3 | 54.81±0.1 | 57.51±0.1 | 57.60±0.3 | 56.32±0.1 | 55.34±0.1 | 63.46±0.1 | 55.94±0.1 | 59.13±0.2 |
| TRC-MLP(linear) | 55.25±0.3 | 62.05±0.4 | 80.88±1.6 | 59.94±0.4 | 54.77±0.1 | 59.28±0.2 | 55.09±0.1 | 56.72±0.2 | 52.96±0.1 | 62.16±0.1 | 52.67±0.1 | 58.55±0.3 |
| TRC-TRANS | 80.43±0.1 | 89.28±0.1 | 87.72±0.1 | 78.75±0.1 | 59.68±0.1 | 70.07±0.1 | 58.91±0.1 | 62.98±0.1 | 56.32±0.1 | 68.22±0.1 | 55.48±0.1 | 56.54±0.1 |
| TRC-TRANS(linear) | 78.53±0.1 | 85.65±0.1 | 82.72±0.1 | 77.55±0.1 | 60.98±0.1 | 70.37±0.1 | 58.94±0.1 | 63.03±0.1 | 50.17±0.1 | 57.31±0.2 | 54.51±1.6 | 50.52±0.1 |
| DIFFAD | **86.16±0.1** | **92.14±0.1** | **94.72±0.1** | **80.98±0.1** | **63.35±0.1** | **74.33±0.1** | **63.05±0.1** | **68.64±0.1** | **57.28±0.1** | **70.27±0.1** | **56.81±0.1** | **65.97±0.1** |

| Method | BlogCatalog | | | | ACM | | | | Cora | | | |
|---|---|---|---|---|---|---|---|---|---|---|---|---|
| | M-F1 | AUC | M-Pre | M-Rec | M-F1 | AUC | M-Pre | M-Rec | M-F1 | AUC | M-Pre | M-Rec |
| GCN | 60.14±0.4 | 68.18±0.7 | 69.36±0.8 | 57.60±0.8 | 56.77±0.1 | 69.21±0.2 | 58.19±0.1 | 55.91±0.1 | 53.12±0.3 | 67.75±0.4 | 65.25±0.5 | 52.71±0.2 |
| GAT | 62.39±0.3 | 71.47±0.3 | 69.51±0.1 | 64.96±0.4 | 61.58±0.1 | 68.11±0.2 | 66.59±0.2 | 59.10±0.1 | 63.15±0.3 | 68.90±0.3 | 65.31±0.3 | 67.43±0.2 |
| GraphSAGE | 63.25±0.2 | 61.31±0.2 | 73.95±0.4 | 59.68±0.1 | 65.70±0.2 | 64.96±0.2 | 67.88±0.2 | 61.26±0.1 | 52.24±0.3 | 68.00±0.4 | 57.01±0.5 | 51.89±0.2 |
| GeniePath | 48.53±0.1 | 52.23±0.1 | 47.14±0.1 | 50.00±0.1 | 49.08±0.1 | 49.02±0.2 | 48.19±0.1 | 50.11±0.1 | 48.70±0.1 | 60.27±0.1 | 47.46±0.1 | 50.04±0.1 |
| FdGars | 42.23±0.3 | 54.59±0.2 | 50.62±0.1 | 52.61±0.2 | 36.69±0.2 | 65.03±0.1 | 51.06±0.1 | 57.56±0.2 | 42.10±0.1 | 69.69±0.4 | 52.44±0.1 | 62.57±0.2 |
| BWGNN | 52.97±0.1 | 51.35±0.3 | 53.44±0.1 | 59.24±0.1 | 54.93±0.3 | 47.69±0.6 | 55.27±0.4 | 56.70±0.2 | 52.18±0.2 | 44.93±0.2 | 53.99±0.2 | 51.85±0.1 |
| DAGAD | 64.49±0.4 | 73.88±0.5 | 73.69±0.8 | 63.26±0.6 | 72.03±0.2 | 73.74±0.1 | 72.57±0.2 | 71.61±0.2 | 65.42±0.2 | 68.78±0.3 | 66.35±0.4 | 64.49±0.2 |
| GAT-sep | 66.82±0.1 | 75.97±0.5 | 67.44±0.1 | 66.40±0.2 | 71.09±0.4 | 76.43±0.2 | 72.03±0.4 | 70.41±0.4 | 59.05±0.4 | 66.68±0.8 | 60.06±0.4 | 58.40±0.4 |
| AMNet | 71.77±0.2 | 72.23±0.3 | 70.29±0.2 | 73.76±0.2 | 60.11±0.1 | 74.54±0.2 | 58.93±0.2 | 62.41±0.1 | 53.09±0.1 | 66.09±0.6 | 53.18±0.1 | 53.13±0.1 |
| GHRN | 56.69±0.2 | 51.62±0.5 | 56.13±0.2 | 60.55±0.2 | 57.60±0.1 | 36.58±0.2 | 72.53±0.3 | 55.09±0.1 | 50.80±0.1 | 47.06±0.7 | 53.73±0.2 | 50.92±0.1 |
| CONAD | 53.87±0.2 | 63.03±0.1 | 58.97±0.1 | 52.90±0.1 | 53.16±0.1 | 70.86±0.1 | 54.28±0.1 | 60.12±0.1 | 53.53±0.1 | 70.48±0.7 | 54.53±0.2 | 66.17±0.1 |
| CoLA | 47.35±0.1 | 58.29±0.2 | 52.14±0.1 | 58.63±0.1 | 43.77±0.5 | 48.68±0.2 | 51.78±0.5 | 50.03±0.2 | 48.18±0.5 | 51.86±0.4 | 51.65±0.2 | 52.48±0.3 |
| FSC | 64.49±0.1 | 66.53±0.3 | 83.38±0.3 | 62.02±0.1 | 60.97±0.1 | 55.09±0.2 | 55.60±0.1 | 60.97±0.2 | 65.63±0.1 | 74.08±0.1 | 64.30±0.2 | 71.13±0.1 |
| FSC(linear) | 52.32±0.1 | 56.79±0.1 | 74.64±0.2 | 54.92±0.1 | 50.67±0.1 | 54.58±0.1 | 54.40±0.2 | 53.72±0.1 | 64.89±0.1 | 74.11±0.1 | 64.21±0.1 | 71.32±0.1 |
| TRC-MLP | 53.03±0.1 | 62.36±0.1 | 87.16±0.1 | 59.32±0.1 | 59.13±0.1 | 72.21±0.1 | 59.68±0.3 | 71.36±0.1 | 63.57±0.1 | 73.57±0.1 | 64.29±0.2 | 70.96±0.1 |
| TRC-MLP(linear) | 48.73±0.1 | 51.15±0.1 | 50.43±0.1 | 50.20±0.1 | 50.01±0.4 | 56.76±0.7 | 52.53±0.2 | 55.52±0.2 | 63.64±0.3 | 73.54±0.1 | 63.56±0.1 | 70.26±0.1 |
| TRC-TRANS | 53.12±0.1 | 54.74±0.1 | 60.51±0.3 | 53.69±0.1 | 51.64±0.1 | 52.44±0.1 | 52.42±0.1 | 52.28±0.1 | 65.71±0.1 | **76.32±0.2** | 66.35±0.3 | **72.91±0.1** |
| TRC-TRANS(linear) | 52.24±0.1 | 53.42±0.1 | 67.96±0.3 | 53.42±0.1 | 51.32±0.1 | 51.26±0.1 | 50.42±0.1 | 51.73±0.1 | 48.59±0.2 | 58.42±0.1 | 52.17±0.1 | 56.28±0.1 |
| DIFFAD | **76.24±0.2** | **77.55±0.5** | **87.88±0.6** | **74.95±0.3** | **73.91±0.2** | **77.40±0.2** | **76.21±0.1** | **74.41±0.2** | **69.28±0.2** | 74.05±0.2 | **68.66±0.2** | 71.08±0.2 |

20% and performed 5-fold test to ensure the robustness and reliability of the results. Unless specially defined in our provided configuration files (under 'configs' directory in our GitHub repository), we set the number of diffusion steps to 1000 and use the cosine scheduler in all our methods. The two GCN layers' dimensions in SDN, DEN and the anomaly detector are 128 for the first layers and 64 for the second layers. For the two constrastive detectors, we classify the top $k$ nodes as anomalies with regard to the anomaly scores, where $k$ is the number of ground-truth anomalies in the test set. Details of our data, splits, and experimental setup are also available in our GitHub repository. We conducted all the experiments on a Rocky Linux 8.6 (Green Obsidian) server equipped with a 12-core CPU, 1 Nvidia V100 GPU (with 30GB RAM), and 100GB RAM.

## J.3 CASE STUDY II - PERFORMANCE V.S. NUMBER OF GENERATED GRAPHS

As DIFFAD is capable of generating an auxiliary graph by simply sampling Gaussian noise and reversing the diffusion process. In this study, we explore whether the performance will improve as more generated graphs are involved for training. The results on four datasets (Fig. 4 and Tables 1, 4) illustrate that augmenting the quantity of generated graphs does not yield a significant performance boost. We attribute this to the limited diversity among the generated graphs and generating diversified samples will be a pivotal focus of our future research.

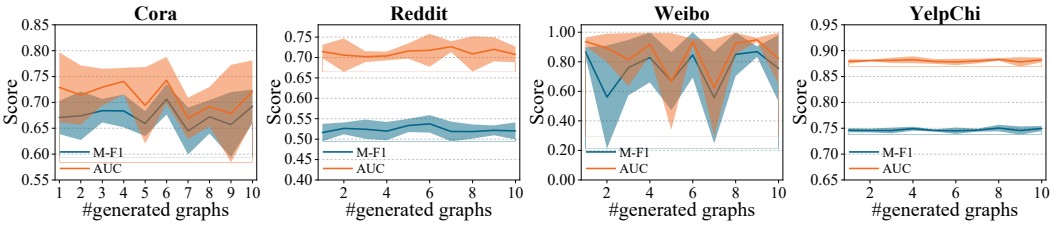

Figure 4: DIFFAD's performance across the different numbers of generated graphs.

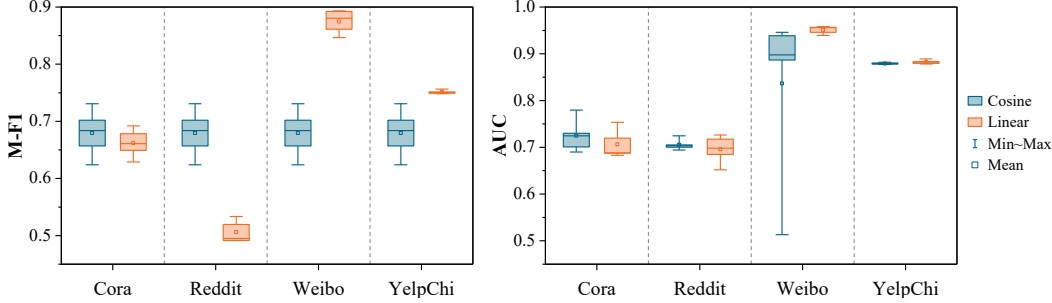

Figure 5: DIFFAD's performance using schedulers (linear vs. cosine) regarding M-F1 and AUC.

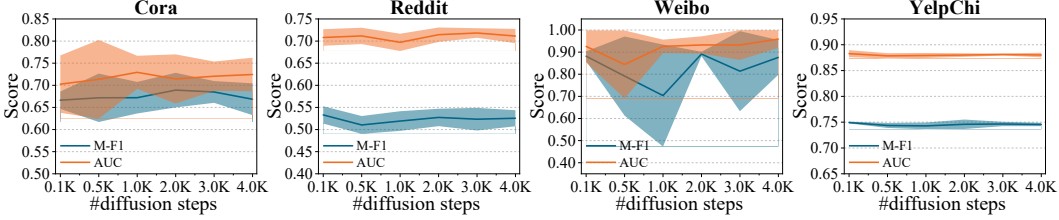

Figure 6: DIFFAD's performance across diffusion steps regarding M-F1 and AUC.

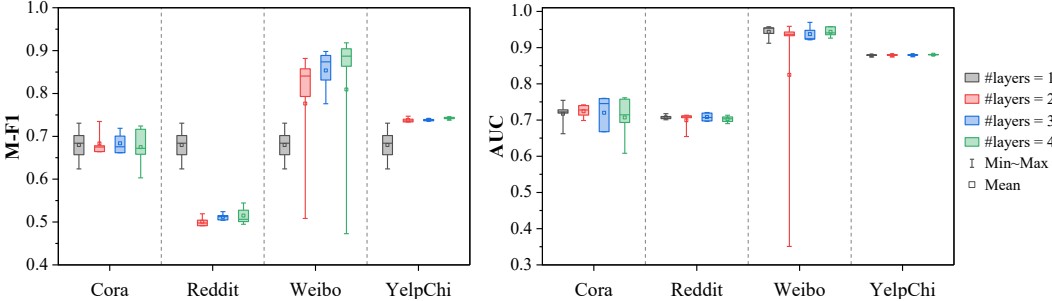

Figure 7: DIFFAD's performance on the different numbers of GNN layers in SDN&DEN regarding M-F1 and AUC.

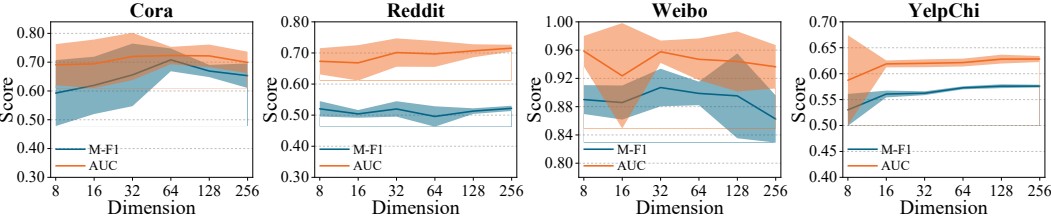

Figure 8: DIFFAD's performance on different GNN dimensions regarding M-F1 and AUC.

### J.4 CASE STUDY III - SENSITIVITY TO DIFFERENT DIFFUSION SETTINGS

**Cosine scheduler v.s. linear scheduler.** The noise variance scheduler plays a key role in controlling the amount of noise added to the original data at each diffusion step and manipulating the diffusion process (Nichol & Dhariwal, 2021). From Fig. 5, we can see that for DIFFAD, the AUC is not much affected by the choice of the scheduler, but the better F1 score on Reddit is achieved using the cosine scheduler while the linear scheduler is better on Weibo. Meanwhile, for FSC, TRC-MLP, and TRC-TRANS, we found that the cosine scheduler is the slightly better linear scheduler.

**The impact of diffusion steps $T$.** The number of diffusion steps decides how fast the original data will be corrupted to standard Gaussian noise. In order to validate whether more diffusion steps will lead to better performance, we set the diffusion steps as $\{100, 500, 1000, 2000, 4000\}$ respectively and report the results on four datasets in Fig. 6. While prolonging the diffusion step corrupts the data more smoothly, its impact on the anomaly detection performance is not significant.

### J.5 CASE STUDY IV - DIFFAD'S SENSITIVITY TO GNN DIMENSIONS AND LAYERS

**The impact of stacking GNN layers.** We conduct tests on the number of GNN layers (set as $\{1, 2, 3, 4\}$) in both SDN and DEN and depict the results in Fig. 7, the AUC score is slightly affected by the number of GNN layers, but the F1 score on Weibo is significantly improved when stacking more GNN layers.

**The impact of GCN detector's dimension.** We further validate how DIFFAD's GCN detector is sensitive to the GNN dimensionality. We fix the number of layers to 2, and illustrate the results in Fig. 8. As can be seen, the detector achieves the best performance when the two GNN layers' dimensions are 128 and 64, respectively.

### J.6 CASE STUDY V - PERFORMANCE WITH SKIP-STEP

Recall that in §5.1 and Appendix D, we have presented a skip-step method to reduce the store cost of **G** and the computational cost of FSC, TRC-MLP and TRC-TRANS. In our implementation, we set the window size $w$ equal to the stride size $s$ in skip-step and calculate the ratio of remaining steps (called keep ratio), as $\frac{w+s}{T}$. We set $T$ as 1000 and vary this ratio among $\{0.01, 0.02, 0.03, 0.04, 0.05, 0.1, 0.2, 0.3, 0.4, 0.5\}$.

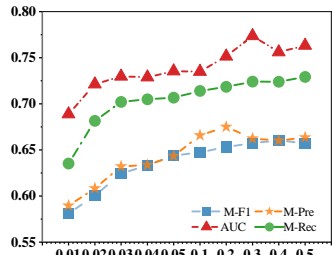

| Variants | M-F1 | AUC | M-Pre | M-Rec |
|---|---|---|---|---|
| **FSC** | **65.63±0.1** | **74.08±0.1** | **64.30±0.2** | **71.13±0.1** |
| ⊘Class-wise loss | 48.67±0.1 | 65.32±0.7 | 48.48±0.2 | 50.00±0.1 |
| **TRC-MLP** | **64.68±0.1** | **73.57±0.1** | **64.29±0.2** | **70.96±0.1** |
| ⊘Class-wise loss | 52.23±0.6 | 61.37±0.8 | 56.05±0.5 | 55.46±0.7 |
| **TRC-TRANS** | **65.71±0.1** | **76.32±0.2** | **66.35±0.3** | **72.91±0.1** |
| ⊘Class-wise loss | 46.76±0.3 | 57.08±0.1 | 51.50±0.1 | 57.08±0.1 |
| **DIFFAD** | **69.28±0.2** | **74.05±0.2** | **68.66±0.2** | **71.08±0.2** |
| ⊘Class-wise loss | 48.70±0.1 | 51.43±0.7 | 47.46±0.1 | 50.00±0.1 |
| ⊘Generation | 66.35±0.3 | 68.95±0.4 | 66.67±0.5 | 67.70±0.4 |
| ⊘Generation & ⊘Class-wise loss | 53.12±0.3 | 67.75±0.4 | 56.21±0.5 | 52.71±0.2 |
| ⊘Residual links | 67.64±0.1 | 70.97±0.2 | 64.15±0.3 | 69.16±0.2 |

Figure 9: Skip-step v.s. performance on Cora.

Table 5: Ablation test on Cora (best in **bold**). '⊘' means the corresponding functional module is detached

As depicted in Fig. 9, downsampling the input tensor has a considerable impact on the detection performance but regarding the fact that the input tensor size increases proportionally to the keep ratio, a trade-off between the performance and cost should be taken in practice.

### J.7 DISCUSSION ON DETECTION PERFORMANCE AGAINST INITIAL EGONET DISSIMILARITY

Since our Observation 1 (presented in Section 4.2) demonstrates the potential of exploring egonet dissimilarity for graph anomaly detection, we are also interested in studying the impact of the initial egonet dissimilarity on the detection performance. Intuitively, lower egonet dissimilarity denotes that anomalies and normal nodes' attributes are similar to each other, making anomalies even harder to be discrimanted from normal nodes. We empirically clarify this impact by comparing the discrepancy in detection performance on datasets that exhibit different initial egonet dissimilarities.

From the results summarized in Tables 1 and 4, we observe that all methods perform significantly better on the Weibo dataset than on Reddit. This is mainly because anomalies have significantly lower feature similarity with normal nodes on the Weibo dataset, while the Reddit dataset exhibits similar average neighbor feature similarities in both classes, according to the statistics reported in (Liu et al., 2022b). This performance drop experienced by all methods highlights the need to address this significant challenge posed by low initial egonet dissimilarity, and we denote this as an important direction for future efforts.

## J.8 ABLATION STUDY

In our ablation study, we demonstrate the efficacy of the key ingredients in our proposed methods: The class-wise loss function formulated in Eq. (10) (for FSC, TRC-MLP and TRC-TRANS) and Eq. (22) (for DIFFAD), generated graph samples, and the effectiveness of the residual links between SDN and DEN. For validating Eq. (10), we replace the class-wise loss in FSC, TRC-MLP and TRC-TRANS as the conventional binary cross-entropy loss. For the rest study with DIFFAD, we implemented four variants of DIFFAD by detaching particular function modules. As reported in Table 5, the row corresponding to '⊘Class-wise loss' represents the variant that replaces our proposed class-wise loss function by the conventional cross-entropy loss to train the anomaly detector. The '⊘Generation' row denotes the results achieved by DIFFAD without using any generated graphs, and '⊘Generation & ⊘Class-wise loss' is the GCN anomaly detector, which uses cross-entropy as the loss function and ignores generated graphs. We also evaluate how the residual links impact the detection performance by comparing DIFFAD and the variant '⊘Residual links', which ignores the residual links between SDN and DEN.

From the results, it is evident that the class-wise loss enables the anomaly detector to focus on both anomalies and regular nodes, leading to better detection performance. DIFFAD's better performance over '⊘Generation' indicates that incorporating generated graphs is an effective strategy for boosting the detection performance. Moreover, the compromised performance of '⊘Residual links' validates that the residual links benefit SDN and DEN to generate better graph samples, which further enhance the detection performance.

