# OpenReview forum: "New recipes for graph anomaly detection: Forward diffusion dynamics and graph generation"
_ICLR.cc/2024/Conference — Submitted to ICLR 2024_

### Official Review · Reviewer_1PRg · 2023-10-28

**Soundness:** 2 fair
**Presentation:** 3 good
**Contribution:** 2 fair
**Rating:** 5
**Confidence:** 3

**Summary:**

The paper investigates the problem of graph anomaly detection in a semi-supervised setting where anomalous nodes tend to differ from their neighbors in terms of the distribution of node features (not structural differences). Under the assumption that the mean attribute values are the same across anomalous and non-anomalous nodes, the authors prove that in a step-wise diffusion process that adds incremental noise to the attribute matrix X, the egonet dissimilarity $\Omega = LX$ for nodes that have higher variance attributes will change more drastically than for those which have lower variance. Based on this observation, the authors propose (non-GNN) neural models that learn from the sequence of egonet dissimilarities to detect anomalies. Moreover, they prove that the relative accumulated energy present in low frequency signals decreases as X gets increasingly corrupted. In light of this observation, they propose a denoising network that utilizes one GCN (low-pass filter) per step. This allows them to generate additional attribute vectors for each node based on their fully-corrupted counterparts. The proposed model, DiffAD, is evaluated on six datasets w.r.t. Macro-F1 and AUC and compared with four graph anomaly detection baselines. In another set of experiments, they show that the generated attribute matrices X bring the performance of GraphSAGE very close to that of DiffAD.

**Strengths:**

S1. Proposes an alternative to GNN-based solutions for detecting anomalous nodes in a graph (which has applications related to spam and fraud detection).

S2. Proposed method, DiffAD, outperforms recent baselines on six datasets.

S3. Proposed graph generation technique leads to substantial performance improvements when used for training classic GNN methods.

S4. The proposed approaches are designed based on the theoretical principles proven in the paper.

S5. The overall writing quality is great, the text is easy to follow and the appendices are used in a thoughtful way.

S6. Overall reproducibility is high; code is also provided (in an anonymous github repo).

**Weaknesses:**

W1. In the preliminary study setup, the attribute distribution chosen for anomalous nodes is substantially different from that of the normal nodes. When noise is incrementally added through diffusion steps, it is not surprising that the egonet dissimilarity drops faster for anomalous nodes.

W2. The authors show that this rapid egonet dissimilarity decrease holds for CORA dataset, but the anomalies in the data were synthetically injected by Liu et al., 2022. For organic anomalies, this assumption might also be true on average (Liu et al., 2022), but the fact that DiffAD  does not account for structural anomalies will likely prevent it from detecting anomalies if their initial egonet dissimilarity is low.

W3. It is not clear how to extend these principles to consider for structural anomalies. The paper dismisses the need to consider structural anomalies that "form more densely [connected] links with other nodes", but there could be anomalies that also keep the density somewhat unchanged.

W4. Some potential typos; some word choices can be improved.

**Questions:**

Q1. Is Observation I an expected result? If so, why?

Q2. A fundamental question not addressed by the paper is: what is the anomaly detection performance on anomalous nodes whose initial ego dissimilarity is low?

Q3. Is it possible to extend the diffusion model to a case where A is changed over time?

Q4. Please clarify:
- In Eq. (18), should one of the $Z_t$ in the concatenation of the first GCN layer be $\bar Z_t$?
- In Eq. (4), specify that this is the accumulated energy ration at rank $l$
- p.2: Confronting -> Adhering to the principles
- p.5: Eventual -> Eventually

---

> ### Author Response · Authors · 2023-11-16
> **Response to Reviewer 1PRg - Part 1/2.**
>
> Dear Reviewer 1PRg,
>
> We greatly appreciate your valuable time and constructive comments. We are pleased that you found our paper to be clear and of good quality. We hope our answers can fully address your concerns.
>
> **W1 and Q1**: In the preliminary study setup, the attribute distribution chosen for anomalous nodes is substantially different from that of the normal nodes. When noise is incrementally added through diffusion steps, it is not surprising that the egonet dissimilarity drops faster for anomalous nodes. Is Observation I an expected result? If so, why?
>
> **R1**: Yes, Observation I aligns with our expectations. In our preliminary study, we set the attribute distribution of contextual anomalies to be substantially different from normal nodes, following the definition in [1], [2], [3]. According to the definition, contextual anomalies are generated by different mechanisms, and their attributes significantly differ from those of their neighbors. Our preliminary study setup adheres to the implementation in [4]. We provided the rigorous analysis of this phenomenon through the proof of Proposition 1 in Appendix B.
>
> **W2**: The authors show that this rapid egonet dissimilarity decrease holds for CORA dataset, but the anomalies in the data were synthetically injected by Liu et al., 2022. For organic anomalies, this assumption might also be true on average (Liu et al., 2022), but the fact that DiffAD does not account for structural anomalies will likely prevent it from detecting anomalies if their initial egonet dissimilarity is low. A fundamental question not addressed by the paper is: what is the anomaly detection performance on anomalous nodes whose initial ego dissimilarity is low?
>
> **R2**: In this paper, we assessed DiffAD's performance on three real datasets, namely, YelpChi, Reddit, and Weibo, apart from the synthesized datasets published in [1], i.e., CORA, ACM, BlogCatalog. Furthermore, we have conducted additional experiments on another three real-world datasets: Tfinance, Tolokers, and Questions, as reported in our **General Response to All Reviewers - Additional Experiments**. Overall, we have validate DiffAD's efficacy on **six real datasets and three synthetic datasets**. From the experimental results, even though these datasets encompass different types of anomalies, including contextual and structural anomalies, DiffAD consistently achieves superior performance compared to the baselines (three new methods have been added).
>
> We also want to emphasize that in real-scenarios, some anomalies can be categorized as both contextual anomalies and structural anomalies (such as the anomalies in Weibo dataset [3]). In addition, to detect contextual anomalies, we have explored both the graph structure and attributes. Although our proposed methods do not explicitly measure the abnormal structural patterns, some structural anomalies can also be detected due to their irregular attributes. From the results on Reddit, in which `the outliers in the Reddit have similar average neighbor feature similarities for outliers and inliers` [3] and the initial egonet dissimilarity is low, we can see that DiffAD achieves the best results with regard to all metrics.
>
> **W3**: It is not clear how to extend these principles to consider for structural anomalies. The paper dismisses the need to consider structural anomalies that "form more densely [connected] links with other nodes", but there could be anomalies that also keep the density somewhat unchanged.
>
> **R3**: Structural anomaly detection is a significant are. In fact, the GCN classifer employed in DiffAD is capable of detecting structural anomalies - just as the conventional GCN-based anomaly detectors employed in many existing methods like [5] [6]. For anomalies that exhibit similar density, we can explore how their attributes are different from their neighbors for detecting anomalies, which is the focus of this work.
>
>
> **Q2**: A fundamental question not addressed by the paper is: what is the anomaly detection performance on anomalous nodes whose initial ego dissimilarity is low?
>
> **R4**: Good question. In fact, we have also studied this in our experiment. We specifically include Reddit, where `the outliers in the Reddit have similar average neighbor feature similarities for outliers and inliers` [3] and the initial egonet dissimilarity is low, to validate these questioned scenarios. From the results reported in Tables 1 and 4, we can see that DiffAD achieves the best results with regard to all metrics.
>
> [1] Graph based anomaly detection and description: A survey (DMKD 2015).
>
> [2] A comprehensive survey on graph anomaly detection with deep learning (TKDE 2021).
>
> [3] BOND: Benchmarking unsupervised outlier node detection on static attributed graphs (NeurIPS 2022).
>
> [4] Rethinking graph neural networks for anomaly detection (ICML 2022).
>
> [5] Dagad: Data augmentation for graph anomaly detection (ICDM 2022).
>
> [6] Deep anomaly detection on attributed networks (SIAM 2019).

---

> > ### Comment · Reviewer_1PRg · 2023-11-16
> >
> > I appreciate the detailed response provided by the authors.
> >
> > R.R.1: Yes, I agree that this is an expected result. Yet, I can certainly see the value of formally proving the proposition.
> >
> > R.R.2: While the proposed method outperforms all the baselines on Reddit, my understanding is that the anomaly detection performance will be worse for nodes that have an initially low egonet dissimilarity. Again, no results were shown for this specific subset of nodes.
> >
> > R.R.4: It is still not clear why $Z_t$ needs to be "duplicated" (via concatenation) in Eqs. (18) and (19). Given the first layer is shared by the mean and the log-variance, it seems that you want to compute both outputs at once, but in this case the operation can be expressed as a single equation where $W_2^\mathrm{DEN}$ and $W_3^\mathrm{DEN}$ are concatenated. Please feel free to correct me if this is wrong.

---

> > > ### Author Response · Authors · 2023-11-17
> > > **Second round response to Reviewer 1PRg.**
> > >
> > > Thanks very much for your prompt response and the acknowledgment of our Observation I in **R.R.1**. For clarity, we refer to our response as **R.RR2** to your comments in **R.R.2**. This convention also applies to **R.R.4**.
> > >
> > > **R.R.2**: While the proposed method outperforms all the baselines on Reddit, my understanding is that the anomaly detection performance will be worse for nodes that have an initially low egonet dissimilarity. Again, no results were shown for this specific subset of nodes.
> > >
> > > **R.RR2**: Your understanding is correct. The detection performance is indeed compromised by anomalies with low egonet dissimilarity, which is a well-acknowledged challenge in graph anomaly detection [2]. In fact, your observation is reflected in our experiments by comparing the results of all methods on Reddit and Weibo. As you can see in Tables 1 and 4, the detection performance of all methods on Weibo is significantly better than that on Reddit. This is mainly because nodes in the Weibo dataset exhibit `significantly lower feature similarities between anomalies and normal nodes (0.004 vs. 0.993)`, while the Reddit dataset has similar average neighbor feature similarities for outliers and inliers [3]. We really appreciate your insightful comment, and we add this observation as a `Discussion on detection performance against initial egonet dissimilarity` in Appendix J.7 on Page 26.
> > >
> > > **R.R.4**: It is still not clear why $\mathbf{Z_t}$ needs to be "duplicated" (via concatenation) in Eqs. (18) and (19). Given the first layer is shared by the mean and the log-variance, it seems that you want to compute both outputs at once, but in this case the operation can be expressed as a single equation where $W^{DEN}_2$ and $W^{DEN}_3$ are concatenated.
> > >
> > > **R.RR4**: You are right. We duplicate $\mathbf{Z_t}$ to compute the mean and log-variance simultaneously, and Eqs. (18) and (19) can indeed be reformulated into a single equation for this purpose. However, for clarity and consistency with Eqs. (15) and (16), we choose to represent them separately to denote the different learning processes of the mean and log-variance.
> > >
> > > [2] A comprehensive survey on graph anomaly detection with deep learning (TKDE 2021).
> > >
> > > [3] BOND: Benchmarking unsupervised outlier node detection on static attributed graphs (NeurIPS 2022).

---

> ### Author Response · Authors · 2023-11-16
> **Response to Reviewer 1PRg - Part 2/2.**
>
> **Q3**: Is it possible to extend the diffusion model to a case where A is changed over time?
>
> **R5**: We appreciate the reviewer's interests in extending diffusion models to gradually corrupt the graph structure. It is worth noting that in the context of anomaly detection on static attributed graphs, the adjacency matrix contains binary/discrete values 1 and 0 to indicate the presence of edges. Different from the continuous node features, simply adding Gaussian noise to the discrete graph structure cannot meet the demand for gradually corrupting the graph structure, and from the equivalent score-based diffusion perspective [1] [2], calculating the derivative of a discrete-valued variable in the diffusion process is non-trivial. In fact, applying denoising diffusion models for discrete data remains an open problem and we recognize this as a potential research direction to detect anomalies.
>
> **Q4**. Please clarify:
>
> - In Eq. (18), should one of the $\mathbf{Z}_t$ in the concatenation of the first GCN layer be $\mathbf{\bar{Z}}_t$?
>   **R6.1**: No, $\mathbf{\bar{Z}}_t$ is concatenated only in the last layer.
> - In Eq. (4), specify that this is the accumulated energy ration at rank $l$
>   **R6.2**: Thanks to the constructive comment. Eq. (4) is to the measure the accumulated energy ratio at rank $l$. We have revised the description above Eq. (4) as:
>
>   - `We further delve into the overall ratios of low and high frequency signals with regard to different thresholds at each step t by measuring the accumulated energy ratio at rank l following Tang et al. (2022)`.
>
> - p.2: Confronting -> Adhering to the principles, p.5: Eventual -> Eventually
> - **R6.3**: Point taken. We have corrected both typos in the revision.

---

> ### Author Response · Authors · 2023-11-23
> **Revised response to structural anomaly detection.**
>
> Dear Reviewer 1PRg,
>
> Regarding your concern on structural anomaly detection, we would like to revise our prior response to your **R3** that the GCN classifer employed in DiffAD is capable of detecting structural anomalies following the message-passing scheme - just as the conventional GCN-based anomaly detectors employed in many existing methods, such as DAGAD [1] and other unsupervised detectors DOMINANT [2]. The experimental results on nine datasets validate the superiority of our method over the baselines. We will revise the manuscript with these details.
>
> [1] Dagad: Data augmentation for graph anomaly detection (ICDM 2022).
>
> [2] Deep anomaly detection on attributed networks (SIAM 2019).
>
> We sincerely hope that our response can address all your concerns.

---

### Official Review · Reviewer_GBF8 · 2023-10-29

**Soundness:** 3 good
**Presentation:** 3 good
**Contribution:** 3 good
**Rating:** 5
**Confidence:** 4

**Summary:**

This paper proposes denoising diffusion and graph generative models for detecting graph anomalies. The denoising diffusion model learns to reconstruct the original graph from noisy samples, while the graph generative model learns to generate new graphs that align with the original graph semantics. The authors evaluate the proposed methods on six datasets and compare them with several state-of-the-art methods, showing promising results in terms of detection accuracy, precision, recall, and AUC. Overall, this paper presents valuable contributions to the field of graph anomaly detection, with potential applications in various domains.

**Strengths:**

1. the authors of this paper present their proposed methods and results in a clear and organized manner, and they provide a thorough discussion of related work and future research directions.

2. Authors integrate graph diffusion into the GAD task to address the challenges of limited prior information about anomalous nodes by generating more training samples.

**Weaknesses:**

1. Graph anomaly detection often includes two types: attribute anomalies and structural anomalies. The author disrupts the original distribution by continuously injecting noise, which may be useful for detecting attribute anomalies. However, is this also effective for detecting structural anomalies?

2.  when injecting scheduled noise to node attributes as the forward diffusion process, anomalies’ egonet dissimilarities change more dramatically than normal nodes, Please provide a clear explanation for this phenomenon.   To calculate egonet dissimilarities, you utilize the formula: LX, which is a graph convolution operation without learnable parameters, why LX can be egonet dissimilarities?

**Questions:**

1. Why are the results of the baseline methods you listed far lower than those reported in their paper？ For example, GCN\ GAT \GraphSage on Cora\ACM datasets

2. baselines are too few to demonstrate the superiority of your method,  you should choose some famous GAD methods, such as COLA,  SL-GAD rather than GCN GAT, etc

---

> ### Author Response · Authors · 2023-11-23
> **Updated Response to Reviewer GBF8.**
>
> Dear Reviewer GBF8,
>
> Thanks for your valuable review and the recognition of integrating graph generation for anomaly detection. We provide detailed answers to your questions as follows, and we hope that our response can address your concerns.
>
> **W1**: Graph anomaly detection often includes two types: attribute anomalies and structural anomalies. The author disrupts the original distribution by continuously injecting noise, which may be useful for detecting attribute anomalies. However, is this also effective for detecting structural anomalies?
>
> **R1**: Our GCN classifier employed in DiffAD is capable of detecting structural anomalies following the message-passing scheme - just as the conventional GCN-based anomaly detectors employed in many existing methods, such as DAGAD [4] and other unsupervised detectors DOMINANT [5]. The experimental results and cases studies in Section 7 demonstrate the efficacy of our proposed method.
>
> **W2**: When injecting scheduled noise to node attributes as the forward diffusion process, anomalies’ egonet dissimilarities change more dramatically than normal nodes, Please provide a clear explanation for this phenomenon. To calculate egonet dissimilarities, you utilize the formula: LX, which is a graph convolution operation without learnable parameters, why LX can be egonet dissimilarities?
>
> **R2**: We explained this Observation I in Section 4.2 with a rigorous proof of Proposition 1 in Appendix B. Regarding your question on the egonet dissimilarity, since $\mathbf{L}\mathbf{X} = (\mathbf{I} - \mathbf{D}^{\frac{1}{2}}\mathbf{A}\mathbf{D}^{\frac{1}{2}})\mathbf{X}$, for each row $i$ in $\mathbf{L}\mathbf{X}$, the value of the $j$-th column denotes how node $v_i$'s attribute is different from its egonet neighbors (1-hop neighbors) regarding the $j$-th node feature. Therefore, row $i$ in $\mathbf{L}\mathbf{X}$ denotes how node $v_i$'s attributes are different from its egonet neighbors, which is the egonet dissimilarity defined in Section 2. Notably, $\mathbf{A}$ does not contain self-links.
>
> **Q1**: Why are the results of the baseline methods you listed far lower than those reported in their paper? For example, GCN/GAT/GraphSage on Cora/ACM datasets.
>
> **R3**: The original GCN, GAT and GraphSAGE are for node classification on pure Cora/ACM datasets. However, in graph anomaly detection, these datasets have been pre-processed by injecting anomalies [3]. We use the synthesized Cora/ACM datasets published in previous graph anomaly detection papers and the links for these datasets are also provided in our anonymous github.
>
> **Q2**: Baselines are too few to demonstrate the superiority of your method, you should choose some famous GAD methods, such as COLA, SL-GAD rather than GCN GAT, etc.
>
> **R4**: Thanks for the constructive comment. We have conducted additional experiments with three most recent semi-/supervised baselines and two contrastive detectors, i.e., CoLA and CONAD, and the results are reported in the **General Response to All Reviewers - Additional Experiments**. From the results, we can see that our method outperforms all baselines.
>
> [1] Score-Based Generative Modeling through Stochastic Differential Equations (ICLR 2020).
>
> [2] Understanding Diffusion Models: A Unified Perspective (arxiv 2022).
>
> [3] BOND: Benchmarking unsupervised outlier node detection on static attributed graphs (NeurIPS 2022).
>
> [4] Dagad: Data augmentation for graph anomaly detection (ICDM 2022).
>
> [5] Deep anomaly detection on attributed networks (SIAM 2019).

---

### Official Review · Reviewer_Ak28 · 2023-10-30

**Soundness:** 3 good
**Presentation:** 3 good
**Contribution:** 3 good
**Rating:** 6
**Confidence:** 3

**Summary:**

This paper proposes a DiffAD, which is a DDPM-based anomaly detector to detect anomalous nodes in a graph that does not need to learn the data distribution explicitly to detect anomalies. The proposed model first learns the diffusion dynamics of ego-net dissimilarities of diffusion steps and then generates auxiliary training samples to enhance a detector's performance. The empirical results on six real-world datasets illustrate the effectiveness of the proposed model.

**Strengths:**

1. The paper provides evidence on the reason why the forward and denoised processes are designed.
2. The empirical results are very promising, DiffAD outperforms SOTA  by a large margin.
3. The loss function can resolve the class imbalance in anomaly datasets.

**Weaknesses:**

1. The definition of 'ego-net dissimilarity` is actually a GNN embedding in literature, so 'dissimilarity` is an inaccurate word.
2. The time complexity should be very high due to the nature of the transformer and DDPM.
3. The motivation is to learn to classify anomalies and normal nodes by not explicitly learning the data distribution,  therefore you propose to learn from the trajectories produced by the forward process. I don't see a strong motivation for data generation for detectors.
4. The training process for the whole is not clear.
5.  DiffAD still needs sufficient data in order to train the model while anomalous nodes are usually very few.

**Questions:**

1. How do you train the overall DDPM model?
2. Why does the forward process need a separate loss function that utilizes the label information?

---

> ### Author Response · Authors · 2023-11-16
> **Response to Reviewer Ak28.**
>
> Dear Reviewer Ak28,
>
> Thank you for your insightful review. We provide detailed answers to your questions as follows, and we hope that our response can address your concerns.
>
> **W1**: The definition of 'ego-net dissimilarity' is actually a GNN embedding in literature, so 'dissimilarity' is an inaccurate word.
>
> **R1**: Thanks for the reminder. In this paper, we employ 'egonet dissimilarity' to specifically denote how each node's attributes are different from its egonet neighbors. We have revised the footnote on Page 3 to clarify this and to remove confusion, we can change it to 'egonet proximity' for the same purpose in the final version.
>
> - `To eliminate confusion, we use 'egonet dissimilarity' to specifically denote how each node's attributes differ from its egonet neighbors, distinct from the embedding method used in previous works.`
>
> **W2**: The time complexity should be very high due to the nature of the transformer and DDPM.
>
> **R2**: For the transformer model (TRC-Trans), its complexity is the same as other graph transformer models. In the meanwhile, we present a skip-step algorithm in Section 5.1 to reduce the computational cost and the impact of this skip-step algorithm is also validated in Appendix J.6. For the generative method (DiffAD), it is true that our sampling process also needs to reverse the diffusion step by step, the same as the DDPM models.
>
> In fact, reducing the computational cost of the Transformer model and accelerating the DDPM sampling process are currently hot research areas. These topics remain under investigation, and since this work primarily focuses on anomaly detection, we defer further enhancements to these methods to our future research efforts. In addition, with regard to your concern on the complexity, we have presented the complexity analysis in Appendix I.
>
> **W3**: The motivation is to learn to classify anomalies and normal nodes by not explicitly learning the data distribution, therefore you propose to learn from the trajectories produced by the forward process. I don't see a strong motivation for data generation for detectors.
>
> **R3**: Thanks for your recognition of our Approach I, which focuses on learning the diffusion dynamics for anomaly detection. In our generative Approach II, we address the challenge of limited knowledge and labeled anomalies in real scenarios by augmenting the number of labeled training data. This augmentation enables the classifier to learn a more effective decision boundary. The experimental results in Table 2 (Section 7.3) demonstrate that the generated graphs significantly enhance the detection performance of all existing anomaly detectors.
>
> **W4 and Q1**: The training process for the whole is not clear. How do you train the overall DDPM model?
>
> **R4**: As summarized in Lines 1-9 in Algorithm 1 in Appendix I, the DDPM model is trained following the objective formulated in Eqs. (20) and (70). Specifically, we take an arbitrary time step $t$ and train the SDN and DEN to predict the added noise $\boldsymbol{\epsilon}_t$ with regard to $\mathbf{X}_t, \mathbf{A}$ and $t$ until the model converges. For clarity, we add a paragraph in Appendix I to summarized the whole training process.
>
> - `For training the generative model, we take an arbitrary time step` $t$ `from` $\{0,\dots, T\}$ `and train the SDN and DEN to predict the added noise` $\boldsymbol{\epsilon}_t$ `with regard to` $\mathbf{X}_t, \mathbf{A}$ `and` $t$`. All the parameters in the generative model are fine-tuned until the training objective, as formulated in Eq. (20), converges...`
>
> **W5**: DiffAD still needs sufficient data in order to train the model while anomalous nodes are usually very few.
>
> **R5**: It is true that (semi-)supervised anomaly detectors need sufficient data to learn a robust decision boundary. In this work, we mitigate the shortage of labeled anomalies by designing DiffAD to generate additional samples that align with the original graph data distribution without using labeled data. We then take advantage of the few labeled anomalies and the generated auxiliary node features to boost the detection performance. In our experiment, we simulate the real-scenario by setting the training ratio to 20%. For more extreme cases, where only one or two nodes are labeled, these remain open problems that we will devote more effort to in the future.
>
> **Q2**: Why does the forward process need a separate loss function that utilizes the label information?
>
> **R6**: We propose three methods, namely FSC, TRC and DIFFAD, for graph anomaly detection. In FSC and TRC, we train classifiers to identify anomalies with regard to the dynamics of the egonet dissimilarity. Therefore, label information is utilized. However, in DiffAD, the forward and reverse process involves no label information for graph generation, as formulated in Eqs. (20) and (70).

---

### Official Review · Reviewer_JRWq · 2023-10-30

**Soundness:** 3 good
**Presentation:** 3 good
**Contribution:** 3 good
**Rating:** 5
**Confidence:** 4

**Summary:**

This paper studies the application of the forward (add noise) and backward (denoise) processes of the diffusion model to graph anomaly detection, respectively. During forward diffusion, the authors exploited the more significant egonet dissimilarity change of the anomalies and proposed the use of LSTM/Transformer/MLP to capture this change information and perform anomaly detection. During backward diffusion, the authors proposed a graph generation method that recovers low-frequency energy as a data augmentation. Theoretical analyses and comprehensive experiments are performed.

**Strengths:**

1. The paper is well written and the charts are professional.

2. The exploration experiment is very detailed and the two propositions are well illustrated.

3. Substantial theoretical analyses and experimental results are provided.

**Weaknesses:**

1.  The paper's focus on a limited type of anomaly (contextual anomaly).

2.  While the analysis of the forward diffusion process is well-done, the proposed method (Approach I) lacks sufficient evidence to demonstrate its superiority.

3.  The experiments presented in the paper have some shortcomings that should be addressed.

**Questions:**

1.  The primary focus of this paper is on contextual anomalies that exhibit marked differences from their neighbors. However, it raises concerns about the representativeness of such anomalies in real-world diverse datasets. This could limit the applicability of the proposed approach. To address this, expanding the experimental evaluation to real datasets[1] would enhance the paper's contributions.

2.  While it is intuitive that adding noise would lead to a rapid drop in egonet dissimilarity for anomalies, it raises questions about whether complex methods in Approach I are necessary for anomaly detection. At the initial stages, the egonet dissimilarity of anomalies significantly outweighs that of normal instances as illustrated in Fig.2. Therefore, it's worth considering if exploiting this difference directly could lead to more efficient detection.

3.  The experimental results in Table 1 highlight that Propositions 1-based methods (FSC, TRC) often fail to outperform baseline approaches on multiple datasets. This further emphasizes the need to scrutinize the necessity and validity of Approach I.

4.  Concerning loss function (Eq. 22), it's vital to ensure that the nodes in the graph $G_a$ generated using denoising techniques align with the class of the original graph $G$. Any inconsistencies could lead to conflicting optimization. Specifically, for an anomaly $v_i$ in graph $G$, how to ensure the generated corresponding node $v_i$ in $G_a$ is also an anomaly?

5.  Although unsupervised methods cannot make use of label information, many of them [2][3] also leverage local inconsistencies for anomaly detection. Providing a comparative analysis against these existing methods would help establish DIFFAD's superiority effectively.

6. The proposed method looks a bit complicated and it would be nice to have comparisons in terms of complexity and computational efficiency.


[1] GADBench: Revisiting and Benchmarking Supervised Graph Anomaly Detection(arxiv, 2023)

[2] Anomaly detection on attributed networks via contrastive self-supervised learning(, 2021)

[3] Reconstruction Enhanced Multi-View Contrastive Learning for Anomaly Detection on Attributed Networks (IJCAI 2022)

---

> ### Author Response · Authors · 2023-11-16
> **Response to Reviewer JRWq - Part 1/2.**
>
> Dear Reviewer JRWq,
>
> We greatly appreciate your valuable time and constructive comments. We hope our answers can fully address your concerns.
>
> **Q1**: The primary focus of this paper is on contextual anomalies that exhibit marked differences from their neighbors. However, it raises concerns about the representativeness of such anomalies in real-world diverse datasets. This could limit the applicability of the proposed approach. To address this, expanding the experimental evaluation to real datasets[1] would enhance the paper's contributions.
>
> **R1**: We have conducted additional experiments on three more real-world datasets in [1], and the results are reported in the **General Response to All Reviewers - Additional Experiments**. It is worth mentioning that we already reported the results on the YelpChi, Reddit and Weibo datasets (in Tables 1 and 4), which are all collected from real-world websites.
>
> **Q2**: While it is intuitive that adding noise would lead to a rapid drop in egonet dissimilarity for anomalies, it raises questions about whether complex methods in Approach I are necessary for anomaly detection. At the initial stages, the egonet dissimilarity of anomalies significantly outweighs that of normal instances as illustrated in Fig.2. Therefore, it's worth considering if exploiting this difference directly could lead to more efficient detection.
>
> **R2**: Thank you for the insightful comment. In fact, the example in Fig. 2 is to illustrate our motivation of Approach I. The node feature in this preliminary study is 1-dimensional, as claimed in Section 4.1 - `The node attributes are randomly drawn from two Gaussian distributions, N(1, 1) for the normal class and N(1, 5) for anomalies, following Tang et al. (2022)`.
>
> Given the high dimensionality of data in real-scenarios, the egonet dissimilarity will be divergent in each dimension. Such multi-dimensional data and pattern pose significant challenges to detect anomalies based on egonet dissimilarity at the initial stage, and this is one of the pivotal reasons for employing deep learning techniques for high-dimensional data analysis and graph anomaly detection. To this end, we propose two methods in Approach I to explore such patterns. Investigating other simple, effective methods based on egonet dissimilarity for high-dimensional graph data learning is a promising direction worth exploration.
>
> **Q3**: The experimental results in Table 1 highlight that Propositions 1-based methods (FSC, TRC) often fail to outperform baseline approaches on multiple datasets. This further emphasizes the need to scrutinize the necessity and validity of Approach I.
>
> **R3**: From the results on Weibo dataset, we can see that FSC and TRC methods achieve better results than others and `we attribute this to the significantly lower feature similarities between anomalies and normal nodes (0.004 vs. 0.993), which makes anomalies’ trajectories easier to be distinguished from normal nodes` (in Section 7.2). Although their performance are not the best on other datasets, we propose FSC and TRC to explore novel clues for graph anomaly detection and our experiment demonstrate that when the egonet dissimilarity is significant, FSC and TRC are even more effective than the baselines.

---

> ### Author Response · Authors · 2023-11-16
> **Response to Reviewer JRWq - Part 2/2.**
>
> **Q4**: Concerning loss function (Eq. 22), it's vital to ensure that the nodes in the graph $\mathcal{G}_a$ generated using denoising techniques align with the class of the original graph $\mathcal{G}$. Any inconsistencies could lead to conflicting optimization. Specifically, for an anomaly $v_i$ in graph $\mathcal{G}$, how to ensure the generated corresponding node $v_i$ in $\mathcal{G}_a$ is also an anomaly?
>
> **R4**: We appreciate the reviewer for raising an insightful question regarding the consistency between the generated graph $\mathcal{G}_a$ and $\mathcal{G}$. Indeed, this aspect is pivotal for generating additional samples to enhance detection performance. Within the DDPM framework, this consistency subjects to the denoising neural network's ability to learn the original data distribution.
>
> In our work, we aim to ensure this consistency by adhering to the two principles detailed on Page 5, as elaborated through our preliminary studies in Section 4. Specifically, by Principle I, we aim to ensure `the prior distribution` $p(\mathbf{X}_a|\mathbf{A})$ `of the generated graph aligns with the original distribution` $p(\mathbf{X}|\mathbf{A})$, `enabling the classifier to learn a more effective decision boundary to distinguish anomalies` and by Principle II, we expect to `recover the low frequency energy` to discriminate nodes in different classes, i.e., anomalous and normal.
>
> Our proposed DiffAD model achieves both by employing GCN to capture the original graph $\mathcal{G}$'s local semantics and recover low frequency signals. We empirically validate the effectiveness of the generated graphs through extensive experiments in Section 7.2. The experimental results not only demonstrate the performance of DiffAD but also the efficacy of the generated graphs on boosting existing detectors, detailed in Section 7.3.
>
> **Q5**: Although unsupervised methods cannot make use of label information, many of them [2]  [3] also leverage local inconsistencies for anomaly detection. Providing a comparative analysis against these existing methods would help establish DIFFAD's superiority effectively.
>
> **R5**: Thanks for the constructive comment. For the limited space, we have added a discussion on these methods exploring local inconsistencies for anomaly detection in the related works in Appendix A.1.
>
> -`Notably, there are plenty of unsupervised anomaly detection techniques, such as CONAD [2], SL-GAD [3] and Sub-CR [4], that investigate the consistency between anomalies and their neighbors in different contrastive views to measure node irregularity. While these methods investigate the egonet dissimilarity for anomaly detection, they only focus on the original graph, and the dynamics of egonet dissimilarity in the forward diffusion process remains unexplored.`
>
> **Q6**: The proposed method looks a bit complicated and it would be nice to have comparisons in terms of complexity and computational efficiency.
>
> **R6**: Per your suggestion, we have added the complexity analysis in Appendix I.
>
> [1] GADBench: Revisiting and Benchmarking Supervised Graph Anomaly Detection (NeurIPS, 2023).
>
> [2] Anomaly detection on attributed networks via contrastive self-supervised learning (TNNLS 2021).
>
> [3] Generative and contrastive self-supervised learning for graph anomaly detection (TKDE 2021).
>
> [4] Reconstruction Enhanced Multi-View Contrastive Learning for Anomaly Detection on Attributed Networks (IJCAI 2022).

---

> ### Comment · Reviewer_JRWq · 2023-11-23
> **Response to Authors**
>
> Dear Authors,
>
> Thanks for your reply. After carefully reading all comments and responses, I keep my rating unchanged. The proposed idea is interesting, but the method is a bit complicated and the performance improvment is limited. Moreover, the consideration about  structural anomalies is also missing, limiting the applicability of the proposed approach.
>
> Best Regards

---

> ### Author Response · Authors · 2023-11-23
> **Second round response to Reviewer JRWq.**
>
> Dear Reviewer JRWq,
>
> Thanks very much for your response. We are delighted that you recognize our proposed idea interesting. To address your concerns, we make the following clarification.
>
> **C1**. The method is a bit complicated.
>
> **R1-C1**. Graph anomaly detection is naturally a complicated problem, given the high-dimensional graph attributes and complex structure. In this paper, we have proposed two novel approaches to `identify novel clues for graph anomaly detection` (Approach I, the implementations include FSC, TRC-MLP and TRC-TRANS), and `address the shortage of labeled training data` (Approach II, DiffAD). All our methods are not more complicated than the corresponding and fundamental LSTM, MLP, Transformer and GCN models.
>
> From the results, we can see that our best method outperforms the baselines on nine widely-used datasets. We'd also like to highlight that our Approach I provides a novel view on how anomalies are different from normal nodes - anomalies' evolving patterns are different from normal nodes in the diffusion process, and our Approach II is feasible of tackling the shortage of labeled anomalies, which is a primary and significant challenge in anomaly detection, by synthesizing high-quality and auxiliary samples.
>
> **C2**. The performance improvement is limited.
>
> **R2-C2**. From the results in `General Response to All Reviewers` and Table 4 in the manuscript, we can see that our methods achieve significantly better performance, as also recognized by Reviewer `Ak28`. For your reference, we report the performance gap between the baselines and our method ($\text{our methods' best result} - \text{baseline's result}$) in `General Response to All Reviewers - Performance gap`. We also conduct pairwise t-test (with a 95% level of confidence) to demonstrate the significance of improvement and the results are provided in our `General Response to All Reviewers - Statistical significance of performance improvement`.
>
> Moreover, as explained in **R1-C1**, our generated graphs can also significantly improve the performance of the baselines.
>
> **C3**. The consideration about structural anomalies is also missing, limiting the applicability of the proposed approach.
>
> **R3-C3**. Thanks for the constructive comment. It is true this our work primarily focuses on anomalies that exhibit deviating attributes compared to their egonet neighbors.
>
> However, in addition to our **R2** to Reviewer `1PRg`, we want to emphasize that categorizing graph anomalies as structural or contextual is to describe how anomalies distinct to normal nodes with respect to the graph structure and attributes. An anomalous node can belong to both categories and some structural anomalies can also be detected due to their irregular attributes. Therefore, our methods are capable of detecting structural anomalies that exhibit these characters, such as those in the widely-used Weibo dataset [1]. Furthermore, the competitive results of BWGNN, which is designed only for contextual anomaly detection, align with our argument, while our best-performing method surpasses the overall performance of the baselines.
>
> Lastly, we state that we target contextual anomaly detection to concisely formulate the problem - `we explore both the graph structure and attribute to identify anomalies that exhibit deviating attributes to their neighbors` (defined in Section 2).
>
> We sincerely hope that our response can address your concerns, and we will gladly answer any additional questions you may have.
>
> [1] BOND: Benchmarking unsupervised outlier node detection on static attributed graphs (NeurIPS 2022).
>
> [2] Rethinking graph neural networks for anomaly detection (ICML 2022).

---

> ### Author Response · Authors · 2023-11-23
> **Additional response to structural anomaly detection.**
>
> Regarding your concern on structural anomaly detection, in addition to our prior response **R3-C3**, we would like to highlight that the GCN classier employed in DiffAD is capable of detecting structural anomalies following the message-passing scheme - just as the conventional GCN-based anomaly detectors employed in many existing methods, such as DAGAD [1] and other unsupervised detectors DOMINANT [2]. We will revise the manuscript to include these details.
>
> [1] Dagad: Data augmentation for graph anomaly detection (ICDM 2022).
>
> [2] Deep anomaly detection on attributed networks (SIAM 2019).
>
> We sincerely hope that our response can address all your concerns, and we will gladly answer any additional questions you may have.

---

### Official Review · Reviewer_o8nC · 2023-11-01

**Soundness:** 3 good
**Presentation:** 2 fair
**Contribution:** 3 good
**Rating:** 5
**Confidence:** 3

**Summary:**

The authors propose a model for anomaly detection in graphs, particularly "contextual" anomalies, which are nodes whose attributes are significantly different from neighboring nodes. The main proposed idea is to use a denoising diffusion probabilistic model (DDPM) to learn a distribution of the graph that can be used to add/remove noise and generate synthetic graphs. The authors then use the DDPM in 2 different ways. First, they combine it with an LSTM to learn to classify nodes in a sequence of increasingly noise graphs as anomalous or normal. Second, they use the DDPM to generate synthetic graphs that can be used to improve predictive performance of other previously proposed models for anomaly detection on graphs.

**Strengths:**

- Using a DDPM to model noise in a graph is interesting and avoids issues with over-smoothing when using graph neural networks for anomaly detection
- Preliminary study is insightful and effective at motivating the proposed methods

**Weaknesses:**

- Limited experimental evaluation. No discussion of ablation or hyperparameter turning.
- No comparison to recently-proposed methods for graph anomaly detection, such as CoLA or CONAD. The justification is that such methods are not classifiers and the just assign an "anomaly score" to nodes. However, one could try a simple baseline of thresholding this anomaly score.
- No discussion of running time
- No discussion of inference for the DDPM

**Questions:**

1. Would it be possible to compare the proposed method to contrastive detectors by applying a threshold?
2. How is the inference for the DDPM performed?
3. What is the computational complexity of the proposed method as a whole?

---

> ### Author Response · Authors · 2023-11-16
> **Response to Reviewer o8nC**
>
> Dear Reviewer o8nC,
>
> Thanks for your valuable time and review. We have conducted additional experiments and analyzed the algorithm complexity in the revision. Below, we provide detailed answers to your questions, and we hope that our response can address your concerns.
>
> **W1**: Limited experimental evaluation. No discussion of ablation or hyperparameter turning.
>
> **R1**: For space limitation, we reported the additional ablation study and hyperparameter sensitivity analysis in Appendix J. These case studies are also referred to in Section 7:
>
> - `We report additional experiments on exploring the impact of the generated graphs, key parameters and the skip-step algorithm, as well as an ablation study in Appendix J.`
>
> **W2 and Q1**: No comparison to recently-proposed methods for graph anomaly detection, such as CoLA [1] or CONAD [2]. The justification is that such methods are not classifiers and the just assign an "anomaly score" to nodes. However, one could try a simple baseline of thresholding this anomaly score. Would it be possible to compare the proposed method to contrastive detectors by applying a threshold?
>
> **R2**: To address your concern about baselines, we have compared DiffAD with three up-to-date semi-/supervised methods and two contrastive detectors, i.e., CoLA and CONAD, on three additional datasets, as reported in the **General Response to All Reviewers - Additional Experiments**. From the results, we can see that our method outperforms all baselines on all datasets.
>
> **W3 and Q3**: No discussion of running time. What is the computational complexity of the proposed method as a whole?
>
> **R3**: Thanks for the constructive comment. We have added the complexity analysis in Appendix I.
>
> **W4 and Q2**: No discussion of inference for the DDPM. How is the inference for the DDPM performed?
>
> **R4**: We provided the discussion on the reverse process (inference) in Section 3.2 and Appendix A.2. Our inference algorithm (formulated in Eq. (21)) was explained in Section 6.2 and Appendix H.
>
> [1] Anomaly detection on attributed networks via contrastive self-supervised learning (TNNLS 2021).
>
> [2] Contrastive attributed network anomaly detection with data augmentation (PAKDD 2022).

---

### Official Review · Reviewer_nduN · 2023-11-04

**Soundness:** 3 good
**Presentation:** 1 poor
**Contribution:** 3 good
**Rating:** 5
**Confidence:** 4

**Summary:**

The authors investigate the behaviors of normal and anomalous nodes in a diffusion process. The authors made two observations that may help design new algorithms for supervised/semi-supervised anomaly detection. The first observation leads to algorithms that are capable of capturing local information in egonet. The second observation leads to the generation of additional graphs based on the diffusion process that can be utilized as auxiliary data to enhance the anomaly detectors. The authors then demonstrate the benefit of the models compared with baselines on several public datasets.

**Strengths:**

Strengths:
- The authors present an interesting analysis of the diffusion process on a graph.
- The proposed approach based on the two observations is also interesting.
- The first observation leads to three non-GNN methods for anomaly detection in graphs.
- The second observation leads to a graph generation technique that can help improve the performance of anomaly detectors.
- The authors demonstrate the benefit of the models against several GNN baselines.
- The graph generation procedure can be used to enhance many other GNN models with a slight modification of the training objective.

**Weaknesses:**

I have a few concerns and questions regarding the paper:
1) One of my biggest concerns about this paper is its clarity of presentation. The paper is hard to understand. One factor is that the authors did not describe enough about how the diffusion process works on graphs. Section 3.2 only explains the generic diffusion process on regular data (x). How it applies in the context of graphs, which contain structure and attributes, is not clearly explained. After explaining Section 3.2, the authors directly jump to the graph observations without explaining how the diffusion process on graphs works.
2) Since the authors did not clearly explain the diffusion process on graphs, many questions may arise about this, such as:
    - How does the structure of the graph (A) contribute to the next node feature representation (X_{t+1})?
    - Any effect of the neighboring node on a particular node representation in the next step?
    - The diffusion process itself tries to model the data distribution, i.e., for graph cases, it models the data distribution of G={A, X}. As the first approach utilizes the diffusion information, why do the authors claim that the approach does not need to learn P(A, X)?
    - And more questions that depend on the answer to the questions above.
3) The principle derived from the first observation is not surprising, i.e., "The denoising neural network should be capable of capturing the local information in egonets". In fact, I would argue that this principle is the bedrock of nearly all graph anomaly detection models.
4) The baselines used in the experiments are relatively inadequate, particularly in the second group. I would suggest the authors to add more baselines in the comparisons, such as:
    - GHRN [1]
    - GDN [2]
    - H2-FDetector [3]
    - GAGA [4]
    - etc.


[1] Yuan Gao, Xiang Wang, Xiangnan He, Zhenguang Liu, Huamin Feng, and Yongdong Zhang. Addressing heterophily in graph anomaly detection: A perspective of graph spectrum. In Proceedings of the ACM Web Conference 2023, pp. 1528–1538, 2023a.

[2] Yuan Gao, Xiang Wang, Xiangnan He, Zhenguang Liu, Huamin Feng, and Yongdong Zhang. Alleviating structural distribution shift in graph anomaly detection. In Proceedings of the Sixteenth ACM International Conference on Web Search and Data Mining, pp. 357–365, 2023b.

[3] Fengzhao Shi, Yanan Cao, Yanmin Shang, Yuchen Zhou, Chuan Zhou, and Jia Wu. H2-fdetector: A gnn-based fraud detector with homophilic and heterophilic connections. In Proceedings of the ACM Web Conference 2022, pp. 1486–1494, 2022.

[4] Yuchen Wang, Jinghui Zhang, Zhengjie Huang, Weibin Li, Shikun Feng, Ziheng Ma, Yu Sun, Dianhai Yu, Fang Dong, Jiahui Jin, et al. Label information enhanced fraud detection against low homophily in graphs. In Proceedings of the ACM Web Conference 2023, pp. 406–416, 2023.

**Questions:**

Please answer my questions in the previous section.

---

> ### Author Response · Authors · 2023-11-16
> **Response to Reviewer nduN - Part 1/2.**
>
> Dear Reviewer nduN,
>
> Thanks very much for your valuable comments. Regarding your concerns on the presentation, diffusion process and baselines, we have carefully revised the manuscript and conducted additional experiments. We hope our answers can fully address your concerns.
>
> **W1**: One of my biggest concerns about this paper is its clarity of presentation. The paper is hard to understand. One factor is that the authors did not describe enough about how the diffusion process works on graphs. Section 3.2 only explains the generic diffusion process on regular data (x). How it applies in the context of graphs, which contain structure and attributes, is not clearly explained. After explaining Section 3.2, the authors directly jump to the graph observations without explaining how the diffusion process on graphs works.
>
> **R1**: We provided preliminaries on the forward graph diffusion process in Section 2 - `The forward graph diffusion is referred to as injecting` $T$ `scales of scheduled noise to node attributes with the fixed graph structure`, which is to `gradually add scheduled Gaussian noise to the original data` $\mathbf{x}_0$ `through a `$T$`-step Markov chain such that the eventual distribution at the last step` q($\mathbf{x}_T$ |$\mathbf{x}_0$) ∼ $\mathcal{N}(0, \mathbf{I})$, formulated in Eq. (1) in Section 3.2. Per your comment and for space limitation, we have added another paragraph at the end of A.2 in Appendix A to better explain the forward graph diffusion process.
>
> - `In this work, we investigate the DDPM framework specifically for graph anomaly detection and following our definition in Section 2, our forward graph diffusion process is to inject scheduled noise to the node attributes,` $\mathbf{X}$,` and the graph structure is fixed at all diffusion steps.`
>
> **W2.1**: How does the structure of the graph (A) contribute to the next node feature representation $\mathbf{X}_{t+1}$?
>
> **R2.1**: In the forward diffusion process (as explained in Section 2), we fix the graph structure $\mathbf{A}$ and it does not affect the corrupted node features $\mathbf{X}_{t+1}$ at the next diffusion step.
>
> **W2.2**: Any effect of the neighboring node on a particular node representation in the next step?
>
> **R2.2**: In addition to **R2.1** and following the DDPM framework, for a specific node $v_i$, its neighbors do not affect the corrupted node features $\mathbf{x}_{t+1}$ at $t+1$ in the `forward diffusion process`. However, it is worth noting that when measuring the egonet dissimilarity and graph energy distribution in the forward process, we can observe divergent evolving patterns between anomalies and normal nodes (our observation I), as well as the graph energy shifts (observation II).
>
> In the `reverse process`, to obtain $\mathbf{X_t}$ from $\mathbf{X_{t+1}}$ and ensure that the generated graph adheres to the original graph semantics, we investigate GCN as a low-pass filter to recover low-frequency signals, following our devised design principles I and II in Section 4. From the spatial view, GCN is aggregating neighboring nodes' information to the target node; thus, neighbors will affect the target node in the reverse process. This can be considered as to generate node attributes conditioned on their neighbors and graph structure, which meets our expectation on synthesizing graphs that adhere to the original graph semantics.
>
> **W2.3**: The diffusion process itself tries to model the data distribution, i.e., for graph cases, it models the data distribution of G={A, X}. As the first approach utilizes the diffusion information, why do the authors claim that the approach does not need to learn P(A, X)?
>
> **R2.3**: We appreciate the reviewer's insightful comment. By claiming `we first explore a novel paradigm to detect anomalies on a static graph without the need to explicitly learn its data distribution`, we want to state that in distinction to the conventional GNN methods, which employ neural networks to explicitly model the data distribution $p(\mathbf{A}, \mathbf{X})$ for downstream tasks, our Approach I alternatively involves neural networks to learn `the egonet dissimilarity changes in the forward diffusion process`. Instead of classifying nodes by learning the distribution $p(\mathbf{A}, \mathbf{X})$, our Approach I is proposed to learn the evolving patterns from $\lbrace \boldsymbol{\Omega_t} \rbrace_{t=0}^T$. In other words, our proposed methods, namely forward sequence classification (Section 5.2) and trajectory representation learning (Section 5.3), learn the evolving patterns rather than $p(\mathbf{A}, \mathbf{X})$ to detect anomalies.

---

> ### Author Response · Authors · 2023-11-16
> **Response to Reviewer nduN - Part 2/2.**
>
> **W3**: The principle derived from the first observation is not surprising, i.e., "The denoising neural network should be capable of capturing the local information in egonets". In fact, I would argue that this principle is the bedrock of nearly all graph anomaly detection models.
>
> **R3**: The denoising neural network is not for distinguishing anomalies; rather, it is proposed to reverse the diffusion process for graph generation, as explained in Section 3.2 and Section 6.1 - `Our goal is to learn the original graph data distribution through the reverse process, which can be practically described as to learn a denoising network` and `we train a two-layered GCN classifier` with the generated graphs (detailed in Section 6.3) for detecting anomalies.
>
> **W4**: The baselines used in the experiments are relatively inadequate, particularly in the second group. I would suggest the authors to add more baselines in the comparisons, such as: GHRN [1], GDN [2], H2-FDetector [3], GAGA [4] etc.
>
> **R4**: Thanks for the constructive comment. We have conducted additional experiments with three recent baselines and the results are reported in the **General Response to All Reviewers - Additional Experiments**. Notably, we exclude GDN [2], H2-FDetector [3] and GAGA [4] because these methods are designed for fraud detection in heterogeneous graphs with more than two types of edges, which is beyond the scope of this paper. Meanwhile, we have cited them in the revision to acknowledge their significance.
>
> [1] Yuan Gao, Xiang Wang, Xiangnan He, Zhenguang Liu, Huamin Feng, and Yongdong Zhang. Addressing heterophily in graph anomaly detection: A perspective of graph spectrum. In Proceedings of the ACM Web Conference 2023, pp. 1528–1538, 2023a.
>
> [2] Yuan Gao, Xiang Wang, Xiangnan He, Zhenguang Liu, Huamin Feng, and Yongdong Zhang. Alleviating structural distribution shift in graph anomaly detection. In Proceedings of the Sixteenth ACM International Conference on Web Search and Data Mining, pp. 357–365, 2023b.
>
> [3] Fengzhao Shi, Yanan Cao, Yanmin Shang, Yuchen Zhou, Chuan Zhou, and Jia Wu. H2-fdetector: A gnn-based fraud detector with homophilic and heterophilic connections. In Proceedings of the ACM Web Conference 2022, pp. 1486–1494, 2022.
>
> [4] Yuchen Wang, Jinghui Zhang, Zhengjie Huang, Weibin Li, Shikun Feng, Ziheng Ma, Yu Sun, Dianhai Yu, Fang Dong, Jiahui Jin, et al. Label information enhanced fraud detection against low homophily in graphs. In Proceedings of the ACM Web Conference 2023, pp. 406–416, 2023.

---

### Author Response · Authors · 2023-11-16
**General Response to All Reviewers - Additional Experiments on three real-world datasets and three state-of-the-art baselines - Part 1/3**

We sincerely appreciate the reviewers for dedicating their valuable time to the review. In response to the comments on real-world datasets and baselines, we have conducted additional experiments on three real-world datasets — Tfinance, Tolokers, and Questions. We have also included three additional semi-/supervised graph anomaly detection baselines: GHRN [1], Amnet [2], and GAT-sep [3]. Our experimental setup strictly follows the published configurations of these baselines, and the three real-world datasets are from the most recent paper on semi-/supervised graph anomaly detection, GADBench [4]. The results demonstrate that our methods consistently outperform the baselines on all datasets, and our generated graphs can also boost the performance the three news detectors on YelpChi and Reddit. All the results ( Table 4 in Appendix J.2, Page 24) and details about the datasets have been updated in the revision (marked in blue).

|`M-F1`|Tfinance|Tolokers|Questions|
|:---------:|----------|---------|---------|
|GCN|72.68+-0.1|55.80+-0.2|53.00+-0.1|
|GAT|48.52+-0.2|55.43+-0.2|50.42+-0.2|
|GraphSAGE|70.97+-0.1|55.66+-0.2|56.37+-0.1|
|GeniePath|48.84+-0.1|43.88+-0.1|49.24+-0.1|
|FdGars|81.86+-0.1|54.05+-0.1|48.93+-0.1|
|BWGNN|85.28+-0.7|57.35+-0.9|56.45+-0.1|
|DAGAD|70.89+-0.6|63.10+-0.1|55.39+-0.1|
|`GAT-sep`|83.20+-0.1|63.12+-0.7|56.05+-0.1|
|`AMnet`|82.98+-0.1|56.74+-0.5|55.63+-0.1|
|`GHRN`|81.64+-1.1|60.56+-0.5|56.81+-0.1|
|**Ours (Best)**|**86.16+-0.1**|**63.35+-0.1**|**57.28+-0.1**|

|`AUC`|Tfinance|Tolokers|Questions|
|:---------:|---------|---------|---------|
|GCN|81.10+-0.4|71.69+-0.1|52.76+-0.1|
|GAT|54.60+-0.5|71.97+-0.1|54.70+-0.1|
|GraphSAGE|43.51+-0.4|72.31+-0.1|51.42+-0.1|
|GeniePath|53.65+-0.1|49.22+-0.1|49.71+-0.1|
|FdGars|78.29+-0.1|52.76+-0.2|54.18+-0.1|
|BWGNN|91.91+-0.3|64.14+-1.6|56.15+-0.1|
|DAGAD|88.05+-0.2|72.67+-0.1|60.57+-0.1|
|`GAT-sep`|91.56+-0.1|72.06+-0.4|70.18+-0.1|
|`AMnet`|91.17+-0.4|65.33+-0.1|61.86+-0.1|
|`GHRN`|91.89+-0.4|71.27+-1.2|56.14+-0.1|
|**Ours (Best)**|**92.14+-0.1**|**74.33+-0.1**|**70.27+-0.1**|

|`M-Pre`|Tfinance|Tolokers|Questions|
|:------:|-------|-------|-------|
|GCN|84.94+-0.8|60.86+-0.1|53.67+-0.5|
|GAT|49.73+-0.2|60.07+-0.1|52.92+-0.1|
|GraphSAGE|85.63+-0.1|60.43+-0.1|54.44+-0.1|
|GeniePath|47.71+-0.1|39.09+-0.2|48.51+-0.1|
|FdGars|54.55+-0.1|54.19+-0.1|50.54+-0.1|
|BWGNN|91.54+-0.5|57.29+-1.1|56.17+-0.1|
|DAGAD|72.18+-1.1|61.07+-0.1|54.61+-0.1|
|`GAT-sep`|85.28+-0.1|60.56+-0.4|55.19+-0.3|
|`AMnet`|86.65+-0.2|57.03+-0.1|55.67+-0.1|
|`GHRN`|86.79+-1.2|61.78+-0.1|56.44+-0.1|
|**Ours (Best)**|**94.72+-0.1**|**63.05+-0.1**|**56.81+-0.1**|

|`M-Rec`|Tfinance|Tolokers|Questions|
|:-------:|-------|-------|-------|
|GCN|71.49+-0.1|55.87+-0.1|52.04+-0.1|
|GAT|52.17+-0.3|55.69+-0.2|50.64+-0.1|
|GraphSAGE|65.99+-0.1|55.73+-0.1|54.38+-0.1|
|GeniePath|51.02+-0.2|50.13+-0.1|50.11+-0.1|
|FdGars|70.76+-0.1|53.96+-0.1|54.39+-0.1|
|BWGNN|78.17+-0.8|58.06+-0.6|56.82+-0.1|
|DAGAD|78.64+-0.6|64.09+-0.2|56.99+-0.1|
|`GAT-sep`|79.42+-0.1|63.14+-0.7|57.37+-0.1|
|`AMnet`|76.88+-0.2|57.89+-0.2|56.86+-0.1|
|`GHRN`|78.66+-0.8|59.86+-0.5|57.32+-0.1|
|**Ours (Best)**|**80.98+-0.1**|**68.64+-0.1**|**65.97+-0.1**|

[1] Addressing heterophily in graph anomaly detection: A perspective of graph spectrum (WWW 2023).

[2] Can Abnormality be Detected by Graph Neural Networks? (IJCAI 2022).

[3] Beyond homophily in graph neural networks: Current limitations and effective designs (NeurIPS 2020).

[4] GADBench: Revisiting and Benchmarking Supervised Graph Anomaly Detection (arxiv, 2023).

---

### Author Response · Authors · 2023-11-16
**General Response to All Reviewers - Three new baselines' results on six datasets and performance improvement brought by generated graphs. - Part 2/3**

**Three new baselines' results on six datasets: YelpChi, Reddit, Weibo, BlogCatalog, ACM, and Cora.**

| `M-F1` |YelpChi|Reddit|Weibo|BlogCatalog|ACM|Cora|
|:---------:|-----------------------|-----------------------|-----------------------|-----------------------|-----------------------|-----------------------|
|`GAT-sep`|65.93+-0.3|49.16+-0.1|91.92+-0.2|66.82+-0.1|71.09+-0.4|59.05+-0.4|
|`AMnet`|54.66+-0.8|50.39+-0.1|91.63+-0.1|71.77+-0.2|60.11+-0.1|53.09+-0.1|
|`GHRN`|65.59+-0.1|45.60+-0.3|89.26+-0.1|56.69+-0.2|57.60+-0.1|50.80+-0.1|
|**Ours (Best)**|**73.88+-0.1**|**51.85+-0.1**|**92.06+-0.1**|**76.24+-0.2**|**73.91+-0.2**|**69.28+-0.2**|

| `AUC` |YelpChi|Reddit|Weibo|BlogCatalog|ACM|Cora|
|:---------:|-----------------------|-----------------------|-----------------------|-----------------------|-----------------------|-----------------------|
|`GAT-sep`|80.01+-0.3|50.22+-0.5|95.71+-0.1|75.97+-0.5|76.43+-0.2|66.68+-0.8|
|`AMnet`|64.01+-1.2|62.14+-0.3|97.11+-0.1|72.23+-0.3|74.54+-0.2|66.09+-0.6|
|`GHRN`|81.92+-0.1|66.09+-0.4|91.78+-0.2|51.62+-0.5|36.58+-0.2|47.06+-0.7|
|**Ours (Best)**|**87.94+-0.1**|**71.20+-0.1**|**98.17+-0.1**|**77.55+-0.5**|**77.40+-0.2**|**76.32+-0.2**|

| `M-Pre` |YelpChi|Reddit|Weibo|BlogCatalog|ACM|Cora|
|:---------:|-----------------------|-----------------------|-----------------------|-----------------------|-----------------------|-----------------------|
|`GAT-sep`|72.10+-0.1|48.33+-0.1|85.95+-0.2|67.44+-0.1|72.03+-0.4|60.06+-0.4|
|`AMnet`|55.68+-1.3|50.94+-0.2|86.86+-0.7|70.29+-0.2|58.93+-0.2|53.18+-0.1|
|`GHRN`|64.28+-0.1|51.57+-0.1|89.11+-0.1|56.13+-0.2|72.53+-0.3|53.73+-0.2|
|**Ours (Best)**|**77.81+-0.8**|**54.36+-0.3**|**92.55+-0.1**|**87.88+-0.6**|**76.21+-0.1**|**68.66+-0.2**|

| `M-Rec` |YelpChi|Reddit|Weibo|BlogCatalog|ACM|Cora|
|:---------:|-----------------------|-----------------------|-----------------------|-----------------------|-----------------------|-----------------------|
|`GAT-sep`|63.74+-0.4|50.01+-0.1|93.98+-0.1|66.40+-0.2|70.41+-0.4|58.40+-0.4|
|`AMnet`|55.51+-0.6|50.82+-0.1|89.56+-0.2|73.76+-0.2|62.41+-0.1|53.13+-0.1|
|`GHRN`|74.14+-0.3|59.79+-0.3|89.43+-0.1|60.55+-0.2|55.09+-0.1|50.92+-0.1|
|**Ours (Best)**|**80.98+-0.1**|**66.35+-0.1**|**95.05+-0.1**|**74.95+-0.3**|**74.41+-0.2**|**72.91+-0.1**|

**Performance improvement brought by generated graphs with regard to `M-F1`**

|`Method`|YelpChi|Reddit|
|:---------:|-----------------------|-----------------------|
|`GAT-sep`|66.91+-0.2 (↑1.5%)| 50.15+-1.8 (↑2.0%) |
|`AMnet`|61.45+-0.3 (↑12%)|51.35+-0.1 (↑1.9%)|
|`GHRN`|69.36+-0.2 (↑5.7%)|47.53+-0.3 (↑4.2%)|
|`CONAD`|52.68+-0.1 (↑11%)|49.15+-0.1 (↑5.9%)|
|`CoLA`|46.14+-0.2 (↑0.7%)|47.04 +-0.1(↑2.1%)|

**Performance improvement brought by generated graphs with regard to `AUC`**

|`Method`|YelpChi|Reddit|
|:---------:|-----------------------|-----------------------|
|`GAT-sep`|83.79+-0.1 (↑4.7%) | 61.55+-0.5 (↑23%)  |
|`AMnet`|81.69+-0.1 (↑27%) | 70.04+-0.1 (↑12%) |
|`GHRN`|86.32+-0.1 (↑5.4%)| 72.59+-0.2 (↑9.8%)|
|`CONAD`|50.32+-0.1 (↑5.9%)|57.23+-0.1 (↑2.6%)|
|`CoLA`| 65.42+-0.1 (↑6.2%)|51.53+-0.1 (↑2.5%)|

---

### Author Response · Authors · 2023-11-21
**General Response to All Reviewers - Comparison with two contrastive learning (unsupervised) methods - Part 3/3**

General Response to All Reviewers - Comparison with two contrastive learning (unsupervised) methods - Part 3/3

According to Reviewers o8nC and GBF8's comments on comparing our method with unsupervised methods, such as CoLA [1] and CONAD [2], we have conducted additional experiments by setting a threshold for classification. We strictly follow the implementations in the original papers and set the threshold according to the anomaly ratio of each dataset. Specifically, we classify the top $k$ nodes as anomalies with regard to the anomaly scores, where $k$ is the number of ground-truth anomalies in the test set. From the results, we can see that our method outperforms these contrastive learning (unsupervised) methods.

We have also revised our responses to Reviewers o8nC and GBF8 in accordance with this experiment. All the results and descriptions are updated in the revised manuscript in Table 4, Page 24.

|`M-F1`|YelpChi|Reddit|Weibo|Tfinance|Tolokers|Questions|BlogCatalog|ACM|Cora|
|:---------:|-----------------------|-----------------------|-----------------------|-----------------------|-----------------------|-----------------------|-----------------------|-----------------------|-----------------------|
|`CONAD`| 47.42+-0.1 | 46.39+-0.1 | 79.01+-0.1 | 43.84+-0.1 |46.54+-0.1|48.64+-0.1| 53.87+-0.2 | 53.16+-0.1 | 53.53+-0.1 |
|`CoLA`| 45.82+-0.1 | 46.09+-0.3 | 49.90+-0.2 | 43.77+-0.1 | 50.78+-0.1 | 47.24+-0.1| 47.35+-0.1 | 43.77+-0.5 | 48.18+-0.5 |
|**Ours (Best)**|**73.88+-0.1**|**51.85+-0.1**|**92.06+-0.1**|**86.16+-0.1**|**63.35+-0.2**|**57.28+-0.1**|**76.24+-0.2**|**73.91+-0.2**|**69.28+-0.2**|

|`AUC`|YelpChi|Reddit|Weibo|Tfinance|Tolokers|Questions|BlogCatalog|ACM|Cora|
|:---------:|-----------------------|-----------------------|-----------------------|-----------------------|-----------------------|-----------------------|-----------------------|-----------------------|-----------------------|
|`CONAD`| 47.50+-0.2 | 55.78+-0.2 | 90.40+-0.1 | 82.24+-0.1 | 61.24+-0.1 | 50.36+-0.1 | 63.03+-0.1 | 70.86+-0.1 | 70.48+-0.7 |
|`CoLA`| 61.60+-0.1 | 50.26+-0.3 | 71.59+-0.4 | 57.68+-0.2 | 55.45+-0.1 | 56.97+-0.1 | 58.29+-0.2 | 48.68+-0.2 | 51.86+-0.4 |
|**Ours (Best)**|**87.94+-0.1**|**71.20+-0.1**|**98.17+-0.1**|**92.14+-0.1**|**74.33+-0.1**|**70.27+-0.1**|**77.40+-0.2**|**73.91+-0.2**|**76.32+-0.2**|

|`M-Pre`|YelpChi|Reddit|Weibo|Tfinance|Tolokers|Questions|BlogCatalog|ACM|Cora|
|:---------:|-----------------------|-----------------------|-----------------------|-----------------------|-----------------------|-----------------------|-----------------------|-----------------------|-----------------------|
|`CONAD`| 47.35+-0.1 | 49.58+-0.1 | 74.91+-0.1 | 47.38+-0.1 | 46.37+-0.1 | 50.54+-0.1 | 58.97+-0.1 | 54.28+-0.1 | 54.53+-0.2 |
|`CoLA`| 46.26+-0.1 | 49.48+-0.1 | 50.53+-0.1 | 47.59+-0.1 | 51.10+-0.1 | 50.42+-0.1 | 52.14+-0.1 |  51.78+-0.5 | 51.65+-0.2 |
|**Ours (Best)**|**77.81+-0.8**|**54.36+-0.3**|**92.55+-0.1**|**94.72+-0.1**|**63.05+-0.1**|**56.81+-0.1**|**87.88+-0.6**|**76.21+-0.1**|**68.66+-0.2**|

|`M-Rec`|YelpChi|Reddit|Weibo|Tfinance|Tolokers|Questions|BlogCatalog|ACM|Cora|
|:---------:|-----------------------|-----------------------|-----------------------|-----------------------|-----------------------|-----------------------|-----------------------|-----------------------|-----------------------|
|`CONAD`| 45.51+-0.1 | 48.02+-0.1 | 88.25+-0.1 | 41.09+-0.1 | 47.13+-0.5 | 52.37+-0.1 | 52.90+-0.1 | 60.12+-0.1 | 66.17+-0.1 |
|`CoLA`| 45.54+-0.1 | 50.43+-0.2 | 50.80+-0.2 | 41.44+-0.2 | 50.91+-0.1 | 52.28+-0.1 | 58.63+-0.1 | 50.03+-0.2 | 52.48+-0.3 |
|**Ours (Best)**|**80.98+-0.1**|**66.35+-0.1**|**95.05+-0.1**|**80.98+-0.1**|**68.64+-0.1**|**65.97+-0.1**|**74.95+-0.3**|**74.41+-0.2**|**72.91+-0.1**|

[1] Anomaly detection on attributed networks via contrastive self-supervised learning (TNNLS 2021).

[2] Contrastive attributed network anomaly detection with data augmentation (PAKDD 2022).

---

### Author Response · Authors · 2023-11-21
**Gentle Reminder.**

Dear Reviewers,

We sincerely appreciate your valuable review. In response to your constructive comments, we have:

**(1)** conducted additional experiments (including five new baselines and evaluations on three recently published real-world datasets);

**(2)** provided the complexity analysis of the algorithms;

and **(3)** carefully prepared our response to your questions.

We hope that the additional results could address your concerns about the performance on detecting real-world anomalies. We also hope that the further complexity analysis and revision could enhance the clarity of our problem settings, observations, motivations, and methods. As the author response period is coming to an end, we would appreciate it if you could consider our response and we are more than willing to address any further comments or questions.

Once again, we thank the reviewers for the valuable feedback, which has undoubtedly contributed to improving the quality of our work.

---

### Author Response · Authors · 2023-11-23
**General Response to All Reviewers - Performance gap**

Per Reviewer JRWq's comment on performance improvement, we clarify the performance gap in this general response. The performance gap is measured as $\text{our methods' best result} - \text{baseline's result}$.

|`M-F1`|YelpChi|Reddit|Weibo|Tfinance|Tolokers|Questions|BlogCatalog|ACM|Cora|
|:---------:|-----------------------|-----------------------|-----------------------|-----------------------|-----------------------|-----------------------|-----------------------|-----------------------|-----------------------|
GCN|27.80|2.70|9.79|13.48|7.55|4.28|16.10|17.14|16.16|
GAT|26.90|2.61|6.44|37.64|7.92|6.86|13.85|12.33|6.13|
GraphSAGE|13.02|2.70|2.86|15.19|7.69|0.91|12.99|8.21|17.04|
GeniePath|27.80|2.70|37.00|37.32|19.47|8.04|27.71|24.83|20.58|
FdGars|24.11|3.33|4.41|4.30|9.30|8.35|34.01|37.22|27.18|
BWGNN|10.20|8.56|2.79|0.88|6.00|0.83|23.27|18.98|17.10|
DAGAD|21.80|2.70|2.43|15.27|0.25|1.89|11.75|1.88|3.86|
GAT|7.95|2.69|0.14|2.96|0.23|1.23|9.42|2.82|10.23|
AMNet|19.22|1.46|0.43|3.18|6.61|1.65|4.47|13.80|16.19|
GHRN|8.29|6.25|2.80|4.52|2.79|0.47|19.55|16.31|18.48|
CONAD|26.46|5.46|13.05|42.32|16.81|8.64|22.37|20.75|15.75|
CoLA|28.06|5.76|42.16|42.39|12.57|10.04|28.89|30.14|21.10|

|`AUC`|YelpChi|Reddit|Weibo|Tfinance|Tolokers|Questions|BlogCatalog|ACM|Cora|
|:---------:|-----------------------|-----------------------|-----------------------|-----------------------|-----------------------|-----------------------|-----------------------|-----------------------|-----------------------|
GCN|30.85|13.46|12.53|11.04|2.64|17.51|9.37|8.19|8.57|
GAT|29.70|6.75|18.31|37.54|2.36|15.57|6.08|9.29|7.42|
GraphSAGE|7.58|19.89|9.82|48.63|2.02|18.85|16.24|12.44|8.32|
GeniePath|39.20|24.90|35.23|38.49|25.11|20.56|25.32|28.38|16.05|
FdGars|34.93|8.15|5.06|13.85|21.57|16.09|22.96|12.37|6.63|
BWGNN|6.98|4.31|5.88|0.23|10.19|14.12|26.20|29.71|31.39|
DAGAD|28.11|9.71|7.12|4.09|1.66|9.70|3.67|3.66|7.54|
GAT|7.93|20.98|2.46|0.58|2.27|0.09|1.58|0.97|9.64|
AMNet|23.93|9.06|1.06|0.97|9.00|8.41|5.32|2.86|10.23|
GHRN|6.02|5.11|6.39|0.25|3.06|14.13|25.93|40.82|29.26|
CONAD|40.44|15.42|7.77|9.90|13.09|19.91|14.52|6.54|5.84|
CoLA|26.34|20.94|26.58|34.46|18.88|13.30|19.26|28.72|24.46|

|`M-Pre`|YelpChi|Reddit|Weibo|Tfinance|Tolokers|Questions|BlogCatalog|ACM|Cora|
|:---------:|-----------------------|-----------------------|-----------------------|-----------------------|-----------------------|-----------------------|-----------------------|-----------------------|-----------------------|
GCN|35.08|6.03|10.17|9.78|2.19|3.14|18.52|18.02|12.45|
GAT|6.84|6.02|3.98|44.99|2.98|3.89|18.37|9.62|3.35|
GraphSAGE|2.70|6.03|2.39|9.09|2.62|2.37|13.93|8.33|11.65|
GeniePath|35.08|5.56|39.03|47.01|23.96|8.30|40.74|28.02|21.20|
FdGars|26.16|3.56|3.87|40.17|8.86|6.27|37.26|25.15|16.22|
BWGNN|14.51|2.81|3.73|3.18|5.76|0.64|34.44|20.94|14.67|
DAGAD|17.00|5.65|3.81|22.54|1.98|2.20|14.19|3.64|2.31|
GAT|5.71|6.03|6.60|9.44|2.49|1.62|20.44|4.18|8.60|
AMNet|22.13|3.42|5.69|8.07|6.02|1.14|17.59|17.28|15.48|
GHRN|13.53|2.79|3.44|7.93|1.27|0.37|31.75|3.68|14.93|
CONAD|30.46|4.78|17.64|47.34|16.68|6.27|28.91|21.93|14.13|
CoLA|31.55|4.88|42.02|47.13|11.95|6.39|35.74|24.43|17.01|

|`M-Rec`|YelpChi|Reddit|Weibo|Tfinance|Tolokers|Questions|BlogCatalog|ACM|Cora|
|:---------:|-----------------------|-----------------------|-----------------------|-----------------------|-----------------------|-----------------------|-----------------------|-----------------------|-----------------------|
GCN|30.98|16.35|12.87|9.49|12.77|13.93|17.35|18.50|20.20|
GAT|30.74|16.35|11.83|28.81|12.95|15.33|9.99|15.31|5.48|
GraphSAGE|22.36|16.35|6.62|14.99|12.91|11.59|15.27|13.15|21.02|
GeniePath|31.01|16.05|38.17|29.96|18.51|15.86|24.95|24.30|22.87|
FdGars|29.43|12.78|8.15|10.22|14.68|11.58|22.34|16.85|10.34|
BWGNN|8.14|5.88|5.28|2.81|10.58|9.15|15.71|17.71|21.06|
DAGAD|28.24|16.13|7.51|2.34|4.55|8.98|11.69|2.80|8.42|
GAT|17.24|16.34|1.07|1.56|5.50|8.60|8.55|4.00|14.51|
AMNet|25.47|15.53|5.49|4.10|10.75|9.11|1.19|12.00|19.78|
GHRN|6.84|6.56|5.62|2.32|8.78|8.65|14.40|19.32|21.99|
CONAD|35.47|18.33|6.80|39.89|21.51|13.60|22.05|14.29|6.74|
CoLA|35.44|15.92|44.25|39.54|17.73|13.69|16.32|24.38|20.43|

---

### Author Response · Authors · 2023-11-23
**General Response to All Reviewers - Statistical Significance of performance improvement.**

We conduct pairwise 𝑡-tests (with a 95% level of confidence) between our method and the baselines to demonstrate the significance of improvement. Each value in the tables is the 𝑝-value of the pairwise 𝑡-test between our method and the baselines. Taken GCN as an example, 'GCN-D' denotes the pairwise t-test between GCN and our method.

According to statistical theory, since all the 𝑝-values are less than 0.05 in the tables, we can safely draw the conclusion that our method has achieved a statistically significant improvement compared to baselines.

|`M-F1`|GCN-D|GAT-D|GeniePath-D|FdGars-D|BWGNN-D|DAGAD-D|GAT-sep-D|AMNet-D|GHRN-D|CONAD-D|CoLA-D|
|:------:|:------:|:------:|:------:|:------:|:------:|:------:|:------:|:------:|:------:|:------:|:------:|
|YelpChi|6.8e-08|7.3e-12|5.4e-13|2.0e-04|8.0e-12|3.5e-08|2.9e-10|3.2e-09|4.5e-05|9.6e-04|5.8e-13|
|Reddit|7.2e-07|6.8e-05|1.4e-09|1.1e-11|9.6e-11|1.1e-11|8.0e-03|3.8e-06|8.2e-12|3.9e-05|6.9e-06|
|Weibo|6.3e-04|1.9e-03|9.5e-06|6.1e-10|6.5e-12|7.4e-07|6.3e-05|2.6e-04|1.1e-10|3.2e-07|2.1e-03|
|Tfinance|1.6e-10|4.5e-03|7.7e-08|9.3e-09|6.9e-07|5.3e-06|9.4e-07|7.9e-07|1.3e-12|4.6e-12|6.8e-05|
|Tolokers|5.3e-08|5.0e-10|6.0e-09|6.0e-05|8.4e-08|7.6e-08|6.3e-11|3.8e-06|9.7e-03|4.0e-06|3.0e-05|
|Questions|1.3e-12|6.4e-06|4.6e-12|7.6e-07|8.6e-13|1.8e-12|9.7e-11|3.6e-10|5.8e-05|1.1e-11|7.3e-06|
|BlogCatalog|3.5e-07|3.5e-03|3.5e-13|8.0e-08|7.2e-05|5.3e-13|7.3e-12|9.2e-14|7.4e-09|5.1e-10|3.1e-06|
|ACM|4.9e-07|5.5e-08|8.8e-10|5.5e-08|6.5e-03|4.9e-10|4.8e-09|8.7e-06|2.6e-12|6.2e-07|7.0e-04|
|Cora|8.7e-11|1.4e-05|8.9e-05|6.0e-03|8.5e-13|3.5e-07|8.7e-03|7.4e-09|2.8e-13|2.2e-13|9.5e-04|


|`AUC`|GCN-D|GAT-D|GeniePath-D|FdGars-D|BWGNN-D|DAGAD-D|GAT-sep-D|AMNet-D|GHRN-D|CONAD-D|CoLA-D|
|:------:|:------:|:------:|:------:|:------:|:------:|:------:|:------:|:------:|:------:|:------:|:------:|
|YelpChi|2.3e-12|8.3e-05|6.7e-07|6.1e-09|2.6e-13|1.4e-07|6.6e-13|2.5e-12|7.5e-09|7.5e-12|1.1e-15|
|Reddit|4.5e-13|3.3e-11|9.5e-10|7.5e-03|3.8e-11|9.1e-04|5.9e-12|5.7e-06|6.8e-11|3.8e-03|7.4e-06|
|Weibo|5.7e-07|4.7e-12|3.1e-10|7.6e-11|9.4e-08|3.4e-03|5.1e-07|7.6e-08|4.1e-08|6.4e-10|4.3e-07|
|Tfinance|3.6e-03|7.0e-11|9.9e-09|6.8e-04|6.4e-13|7.8e-10|9.9e-05|9.8e-07|9.5e-08|5.2e-10|6.6e-03|
|Tolokers|4.4e-10|5.0e-06|9.8e-08|9.3e-11|4.9e-10|6.4e-04|6.3e-05|6.7e-04|9.6e-05|7.4e-12|1.0e-11|
|Questions|1.7e-13|1.2e-04|9.0e-12|6.8e-07|3.9e-06|3.2e-04|9.4e-08|6.9e-10|4.1e-10|1.9e-06|6.1e-03|
|BlogCatalog|1.8e-10|1.3e-12|7.3e-11|3.7e-03|1.4e-05|7.2e-12|2.1e-03|8.2e-05|2.5e-09|1.0e-09|9.0e-03|
|ACM|1.0e-11|7.4e-07|9.6e-11|8.2e-10|9.2e-08|3.2e-07|1.7e-07|1.9e-11|9.6e-12|7.0e-04|1.5e-04|
|Cora|7.1e-11|5.4e-07|4.9e-08|8.6e-05|5.0e-04|2.9e-07|2.6e-05|6.0e-03|3.2e-13|5.9e-13|2.6e-08|


|`M-Pre`|GCN-D|GAT-D|GeniePath-D|FdGars-D|BWGNN-D|DAGAD-D|GAT-sep-D|AMNet-D|GHRN-D|CONAD-D|CoLA-D|
|:------:|:------:|:------:|:------:|:------:|:------:|:------:|:------:|:------:|:------:|:------:|:------:|
|YelpChi|9.2e-06|8.3e-08|8.7e-11|3.3e-09|7.4e-12|8.3e-09|5.8e-09|5.3e-11|1.3e-13|5.1e-09|1.1e-05|
|Reddit|2.6e-06|5.8e-10|8.3e-10|9.8e-09|8.2e-09|2.3e-07|3.9e-13|3.4e-07|8.3e-13|3.0e-03|2.6e-09|
|Weibo|4.9e-08|4.6e-07|8.9e-12|1.2e-12|9.2e-08|2.1e-10|7.7e-05|7.9e-05|4.2e-03|6.7e-06|3.8e-04|
|Tfinance|5.4e-11|1.4e-07|6.1e-08|8.0e-05|1.9e-07|5.0e-03|9.2e-04|6.9e-06|6.5e-12|8.9e-13|5.0e-13|
|Tolokers|5.2e-10|5.5e-06|6.5e-10|3.6e-07|4.9e-08|6.9e-08|6.0e-04|7.5e-12|8.6e-07|6.5e-13|9.6e-10|
|Questions|2.7e-13|9.7e-10|9.7e-07|8.3e-05|4.2e-12|9.3e-05|1.3e-03|5.0e-05|8.0e-12|3.0e-09|5.3e-10|
|BlogCatalog|9.3e-03|3.6e-09|3.9e-09|2.0e-03|8.6e-07|9.3e-05|6.1e-12|6.2e-08|5.2e-04|1.6e-10|7.7e-09|
|ACM|7.6e-06|9.7e-07|5.5e-13|8.9e-10|5.0e-12|2.2e-11|2.0e-10|7.2e-11|5.2e-13|4.3e-05|9.0e-05|
|Cora|1.7e-06|6.1e-04|6.3e-11|2.2e-12|8.4e-05|7.1e-05|9.3e-03|5.2e-04|7.6e-08|3.6e-09|2.6e-07|


|`M-Rec`|GCN-D|GAT-D|GeniePath-D|FdGars-D|BWGNN-D|DAGAD-D|GAT-sep-D|AMNet-D|GHRN-D|CONAD-D|CoLA-D|
|:------:|:------:|:------:|:------:|:------:|:------:|:------:|:------:|:------:|:------:|:------:|:------:|
|YelpChi|5.6e-09|7.5e-08|6.4e-06|3.3e-12|8.9e-04|8.6e-08|1.1e-06|3.8e-11|5.4e-11|8.4e-11|2.8e-03|
|Reddit|5.7e-11|9.0e-08|7.1e-10|2.0e-10|7.9e-07|3.0e-11|8.2e-08|3.9e-12|5.0e-06|4.1e-12|6.0e-06|
|Weibo|7.9e-10|3.3e-04|2.8e-13|2.2e-03|3.2e-09|3.7e-04|4.6e-11|2.2e-08|6.0e-03|8.3e-12|3.9e-05|
|Tfinance|6.6e-09|2.6e-11|8.4e-04|8.3e-10|6.5e-05|3.6e-05|5.9e-07|3.9e-08|9.0e-08|5.3e-12|6.2e-13|
|Tolokers|5.7e-03|7.5e-06|2.6e-03|5.4e-09|8.1e-06|5.3e-12|1.1e-13|4.0e-05|5.4e-11|7.6e-11|8.2e-11|
|Questions|1.4e-12|1.6e-04|6.0e-04|9.6e-12|2.0e-06|8.1e-05|8.5e-12|5.8e-05|6.9e-10|9.6e-06|4.9e-08|
|BlogCatalog|3.1e-13|9.9e-10|3.6e-11|5.0e-10|6.9e-10|9.8e-11|2.6e-12|4.8e-04|4.7e-07|6.5e-09|3.7e-12|
|ACM|9.6e-08|4.2e-05|8.8e-12|2.3e-04|3.6e-12|3.9e-09|3.1e-06|4.6e-05|2.1e-09|1.6e-10|5.5e-08|
|Cora|7.0e-03|1.8e-09|5.6e-12|1.0e-10|5.8e-07|7.9e-11|3.3e-04|7.6e-08|7.2e-04|7.5e-09|7.3e-09|

---

### Meta-Review · Area_Chair_5VpX · 2023-12-05

**Metareview:**

The paper introduces a new method for anomaly detection via a diffusion model that first adds and then removes noise to highlight anomalous parts of the input graph.

The method proposed in the paper is interesting and the reviewers and the AC really appreciated the effort in addressing the comments. Nevertheless the paper has few important shortcomings that have been highlighted in the discussion.

- the presentation could be improved, several questions are raised by

- Running time of the method is high so the method is not too practical

- the experimental results are nice but not too exciting

Overall, the paper has some merits but it is below the ICLR acceptance bar.

**Justification For Why Not Higher Score:**

- the presentation could be improved, several questions are raised by

- Running time of the method is high so the method is not too practical

- the experimental results are nice but not too exciting

**Justification For Why Not Lower Score:**

N / A

---

### Decision · Program_Chairs · 2024-01-16

Reject